# Impact of the South Asian monsoon outflow on atmospheric hydroperoxides in the upper troposphere

Bettina Hottmann[1], Sascha Hafermann[1], Laura Tomsche[1*†], Daniel Marno[1], Monica Martinez[1], Hartwig Harder[1], Andrea Pozzer[1], Marco Neumaier[2], Andreas Zahn[2], Birger Bohn[3], Greta Stratmann[4], Helmut Ziereis[4], Jos Lelieveld[1] and Horst Fischer[1]

[1]Atmospheric Chemistry Department, Max Planck Institute for Chemistry, Mainz, 55128, Germany
[2]Karlsruhe Institute of Technology, Karlsruhe, 76021, Germany
[3]Forschungszentrum Jülich GmbH, Jülich, 52425, Germany
[4]German Aerospace Center, Institute of Atmospheric Physics, Oberpfaffenhofen, 82234, Germany
[*]now at NASA Langley Research Center, Hampton, VA 23681, USA
[†]now at Universities Space Research Association, Columbia, MD 21046, USA
*Correspondence to*: Bettina Hottmann (Bettina.Hottmann@mpic.de) and Horst Fischer (Horst.Fischer@mpic.de)

**Abstract.** During the OMO (Oxidation Mechanism Observation) mission, trace gas measurements were performed onboard the HALO (High Altitude LOng range) research aircraft in summer 2015 in order to investigate the outflow of the south Asian summer monsoon and its influence on the composition of the Asian Monsoon Anticyclone (AMA) in the upper troposphere over the eastern Mediterranean and the Arabian Peninsula. This study focuses on *in situ* observations of hydrogen peroxide ($H_2O_2^{obs}$) and organic hydroperoxides ($ROOH^{obs}$), as well as their precursors and loss processes. Observations are compared to photostationary state calculations (PSS) of $H_2O_2^{PSS}$, and extended by a separation of $ROOH^{obs}$ into methyl hydroperoxide ($MHP^{PSS}$) and inferred unidentified hydroperoxide ($UHP^{PSS}$) mixing ratios using PSS calculations. Measurements are also contrasted to simulations with the general circulation ECHAM/MESSy for Atmospheric Chemistry (EMAC) model. We observed enhanced mixing ratios of $H_2O_2^{obs}$ (45%), $MHP^{PSS}$ (9%) and $UHP^{PSS}$ (136%) in the AMA relative to the northern hemispheric background. Highest concentrations for $H_2O_2^{obs}$ and $MHP^{PSS}$ of 211 $ppb_v$ and 152 $ppb_v$, respectively were found in the tropics outside the AMA, while for $UHP^{PSS}$, with 208 $ppt_v$ highest concentrations were found within the AMA. In general, the observed concentrations are higher than steady-state calculations and EMAC simulations by a factor of 3 and 2, respectively. Especially in the AMA, EMAC underestimates the $H_2O_2^{EMAC}$ (medians: 71 $ppt_v$ vs. 164 $ppt_v$) and $ROOH^{EMAC}$ (medians: 25 $ppt_v$ vs. 278 $ppt_v$) mixing ratios. Longitudinal gradients indicate a pool of hydroperoxides towards the center of the AMA, most likely associated with upwind convection over India. This indicates main contributions of atmospheric transport to the local budgets of hydroperoxides along the flight track, explaining strong deviations from steady-state calculations which only account for local photochemistry. Underestimation of $H_2O_2^{EMAC}$ by appr. a factor of 2 in the NH and the AMA and overestimation in the SH (factor 1.3) are most likely due to uncertainties in the scavenging efficiencies for individual hydroperoxides in deep convective transport to the upper troposphere, corroborated by a sensitivity study. It seems that the observed excess $UHP^{PSS}$ is excess MHP transported to the west from an upper tropospheric source related to convection in the summer monsoon over South-East Asia.

## 1 Introduction

The earth has an oxidizing atmosphere where OH functions as the main oxidizing agent (Levy, 1971). OH is formed by the photolysis of ozone ($\lambda$<320 nm) and subsequent reaction of the produced singlet D oxygen atom ($O^1D$) with water vapor.

The main sinks of OH are also the main sources of peroxy radicals ($HO_2$ and $RO_2$) in the reactions with CO, $CH_4$ and volatile organic compounds (VOCs) and the reaction with nitrogen dioxide ($NO_2$) to form nitric acid ($HNO_3$). At low $NO_x$ (NO+$NO_2$) concentrations, $HO_2$ reacts with itself to form $H_2O_2$ or with $RO_2$ to form organic hydroperoxides (ROOH). Since $HO_2$ and $RO_2$, especially $CH_3O_2$, react faster with NO than with $HO_2$, peroxides are mainly produced in areas with low NO and high OH mixing ratios (Lee et al., 2000). $H_2O_2$ is a strong oxidant in the aqueous phase, oxidizing for example $SO_2$ to $H_2SO_4$, and hence $H_2O_2$ partially contributes to acid rain formation (e.g. Hoffmann and Edwards, 1975; Penkett et al., 1979; Robbin Martin and Damschen, 1981; Calvert et al., 1985). The major photochemical sinks of hydroperoxides are photolysis, which recycles OH, and the reaction with OH forms $HO_2$. Physical loss of hydroperoxides due to dry and wet deposition establishes an ultimate loss mechanism of $HO_x$ radicals. Thus $H_2O_2$ and ROOH play a pivotal role to the $HO_x$-budget and modulate the oxidation capacity of the atmosphere (Lelieveld and Crutzen, 1990; Crutzen et al., 1999).

The global distribution of hydroperoxides is affected by transport, physical removal by dry deposition and rainout as well as net photochemical production processes. With increasing altitude, and thus decreasing water vapor concentration, the primary production of $HO_x$ decreases (Heikes et al., 1996) and leads to an increasing contribution of the photolysis of $H_2O_2$ and ROOH to the $HO_x$ budget (Jaeglé et al., 1997; Jaeglé et al., 2000; Faloona et al., 2000; Faloona et al., 2004). In more polluted areas, especially in the boundary layer, the $H_2O_2$ chemistry is more complex and leads to higher variabilities (Nunnermacker et al., 2008). Close to the surface dry deposition of $H_2O_2$ forms a strong sink resulting in decreasing concentrations with decreasing altitude. This often leads to a local maximum of $H_2O_2$ mixing ratios above the boundary layer at 2–5 km of altitude (Daum et al., 1990; Heikes, 1992; Weinstein-Lloyd et al., 1998; Snow, 2003; Snow et al., 2007; Klippel et al., 2011). A similar but weaker maximum at 2–5 km was found for methyl hydroperoxide (MHP) (Weinstein-Lloyd et al., 1998; Snow, 2003; Snow et al., 2007). Due to its lower deposition velocity associated with less efficient uptake by solid and aqueous surfaces, MHP is not as sensitive to deposition processes as $H_2O_2$ (Lind and Kok, 1986, 1994), yielding rather constant mixing ratios with altitude within the boundary layer. Further, the mixing ratios of both species generally decrease with increasing latitude in the free troposphere due to lower water vapor concentrations (Jacob and Klockow, 1992; Perros, 1993; Slemr and Tremmel, 1994; Snow, 2003; Snow et al., 2007; Klippel et al., 2011).

In spite of several *in situ* measurement campaigns of trace gases in the outflow of the Asian summer monsoon in the recent years, e.g. from the IAGOS-CARIBIC project (Ojha et al., 2016; Rauthe-Schöch et al., 2016), the IAGOS-MOZAIC project (Barret et al., 2016), the MINOS aircraft campaign (Lelieveld et al., 2002) and the PEM-WEST A mission (Heikes et al., 1996) our understanding of the physical and chemical processes within the Asian Monsoon Anticyclone (AMA) is limited. So far we know that the updrafts of the summer monsoon deep convection can effectively transport insoluble pollutants from the surface to the upper troposphere and there these polluted air masses can be transported over a long distance (Lawrence

and Lelieveld, 2010). Thus the Asian summer monsoon has a strong influence on the upper troposphere (UT) and the lower stratosphere (Randel et al., 2010; Gettelman et al., 2004) and it is important to study its physical and chemical properties in greater detail.

The focus of the OMO (Oxidation Mechanism Observation) campaign was to investigate photochemical processes in the AMA in the UT. During the mission, HALO probed a large variety of air masses, ranging from clean northern hemispheric (NH) background air above the western Mediterranean, southern hemispheric (SH) background air over the northern Indian Ocean and air masses affected by the South Asian summer monsoon in the AMA over the Arabian Peninsula. The main goals of the campaign were to analyze the influence of the AMA on the oxidizing power of the atmosphere and to determine the rates at which natural and human-made compounds are converted by oxidation processes in the atmosphere (Lelieveld et al., 2018).

The present study addresses the budgets of $H_2O_2$ and organic hydroperoxides. Since the measurements of the sum of all organic hydroperoxides do not differentiate between different species, we estimate the contribution from $MHP^{PSS}$ based on steady-state calculations. In former studies MHP was identified as the most abundant organic hydroperoxide in the free troposphere (Heikes et al., 1996; Jackson and Hewitt, 1996). Our goal was to investigate to what extent this is also the case for the outflow of the South Asian summer monsoon into the UT. In addition the *in situ* data were compared to results from the EMAC model (see 3.5) along the flight track for $H_2O_2$ and individual ROOH mixing ratios. $H_2O_2^{obs}$ mixing ratios were also evaluated with steady-state calculations based on measured $HO_x$ and photolysis frequency measurements onboard of HALO.

## 2 The OMO project

The OMO campaign took place from 21$^{st}$ of July to 27$^{th}$ of August 2015. During the campaign 17 flights with the HALO (High Altitude and LOng range) research aircraft were performed. The airports of Oberpfaffenhofen (Germany), Paphos (Cyprus), Gan (Maldives) and Bahrain served as bases for take-offs and landings. The flights were mainly performed over the Arabian Peninsula, the Eastern Mediterranean and the Northern Indian Ocean (11.3–80.2°E and 0.2°S–48.1°N). In Figure 1 the tracks of all OMO flights are shown. The aircraft reached altitudes up to 15 km which corresponds to 130 hPa to study the chemistry of the UT.

## 3 Methods

### 3.1 Hydroperoxide measurements

The hydroperoxide data ($H_2O_2^{obs}$ and total organic hydroperoxides $ROOH^{obs}$) during OMO were obtained using a modified commercial instrument (AEROLASER, model AL2021, Garmisch-Partenkirchen, Germany) called HYPHOP (HYdrogen Peroxide and Higher Organic Peroxides monitor). The HYPHOP-instrument was installed in a 19″ rack together with the IR-

laser absorption instrument TRISTAR (Tracer *In Situ* Tdlas for Atmospheric Research) mounted in the back of HALO. Air was sampled from the top of the aircraft fuselage through a forward-facing trace gas inlet (TGI) designed as a bypass, consisting of a ½" PFA (perfluoroalkoxy alkanes) tube inside the aircraft with an exit through a second TGI. From this bypass air was sampled at a flow rate of 2 slpm (standard liter per minute) through a ¼" PFA tube and directed to HYPHOP. To obtain constant pressure at the HYPHOP inlet a constant pressure inlet (CPI) consisting of a dual stage membrane pump (Vacuubrand MD1C VARIO SP, Wertheim, Germany) was used (Klippel et al., 2011).

HYPHOP relies on a dual enzyme detection method after transfer of gaseous hydroperoxides into a buffered solution (potassium hydrogen phthalate/NaOH, pH 6) in a glass stripping coil (Lazrus et al., 1985; Lazrus et al., 1986). This stripper also contains EDTA (ethylenediaminetetraacetic acid) to prevent the reaction of transition metal ions with the hydroperoxides. Additionally, formaldehyde (HCHO) is added to prevent the oxidation of dissolved $SO_2$ (in alkaline solutions $HSO_3^-$) by the hydroperoxides. Instead, HCHO and $HSO_3^-$ form hydroxymethyl sulfonate ($HOCH_2SO_3^-$). After the stripping coil the hydroperoxide containing solution is divided into two channels. Catalase is added to one channel in order to selectively destroy $H_2O_2$. This first channel thus measures only ROOH, while the second channel (without catalase) measures the sum of ROOH and $H_2O_2$. Since hydroperoxides cannot be detected by fluorescence directly, a second enzyme (horseradish peroxidase) and *p*-hydroxyphenylacetic acid (POPHA) are added to both channels. In a quantitative and selective reaction the enzyme catalyzes the oxidation of POPHA by hydroperoxides forming the fluorescent dye 6,6'-dihydroxy-3,3'-biphenyldiacetid acid. After excitation at 326 nm with a Cd lamp, the fluorescence at 400–420 nm is detected. To enlarge the fluorescence intensity sodium hydroxide is added.

In order to perform zero measurements, the sampled air is directed through a cylinder filled with hopcalite ($MnO_2$ and CuO) to eliminate $H_2O_2$, ROOH and Ozone. Since the efficiency of Hopkalit decreases with increased humidity, the air is dried beforehand with the help of orange gel ($SiO_2$ beads plus indicator).

To convert the detected signal into a concentration a 4-point calibration was performed before and after every flight. In the first two steps a liquid standard of $H_2O_2$ (1 µmol/L, freshly diluted from stock solution) followed by zero air is measured in both channels without catalase. Afterwards this is repeated with catalase in the ROOH channel for the last two steps. The sensitivity for both channels and the catalase efficiency are determined via this procedure. The concentration of the liquid standard is based on titration of the stock solution (10 mmol/L) with potassium permanganate.

To determine the stripping efficiency for $H_2O_2$, a gas phase standard based from a permeation source (Teflon tube filled with 30% $H_2O_2$ in a temperature-controlled glass flask) is used at a constant flow rate of approximately 40 sccm diluted with synthetic air and measured with the instrument. The permeation rate of the source is quantified by collecting the output of the source into cooled water. The addition of hydrochloric titanium tetrachloride yields the formation of the yellow $\eta^2$-peroxo complex $[Ti(\eta^2-O_2)Cl_4]^{2-}$ (Pilz and Johann, 1974) whose concentration is determined via a UV photometer. The stripping efficiency of MHP was assumed to be 60% and that of $H_2O_2$ 100% (AEROLASER, 2006; Lee et al., 2000).

The inlet efficiency was determined with the help of the permeation source which was measured with and without the CPI. In laboratory studies the inlet efficiency was determined to be 87% ± 3% decreasing during the campaign to 62.7% ± 0.8%, which is mainly due to the higher humidity.

The limits of detection (LOD) and precisions for $H_2O_2$ and MHP (assuming total ROOH[obs] to be only MHP), respectively, have been calculated for each flight from the reproducibility (1$\sigma$ standard deviation) of in-flight zero (650 values) and liquid calibration (100 values) measurements, taking into account the sensitivity, stripping and catalase efficiency. Values for the LOD are in the range of 8–53 ppt$_v$ for $H_2O_2$[obs] (median 23 ppt$_v$) and 9–52 ppt$_v$ for ROOH[obs] (median 23 ppt$_v$) respectively, assuming that ROOH[obs] is composed of MHP only. Please note, that due to the fact, that the exact composition of ROOH[obs] is

unknown, and the solubility of different ROOH species can be quite variable, a detection limit for ROOH[obs] cannot be given. Instead, we calculate an upper limit of the detection limit, assuming that all ROOH[obs] consists of MHP, the species with the smallest solubility. Precision values were determined from the reproducibility of standard measurements and are in the range of 0.2%@5.2 ppb$_v$ and 1.3%@5.9 ppb$_v$ for $H_2O_2$ and 0.3%@5.0 ppb$_v$ and 2.1%@6.0 ppb$_v$ for MHP. The time resolution (signal increase from 10% to 90%) of the instrument is 120 s. An ozone interference of 53 ppt$_v$ $H_2O_2$ per 100 ppb$_v$ $O_3$, which

was determined by $H_2O_2$ measurements in the stratosphere during the OMO-EU test campaign, was taken into account and corrected.

The total uncertainty calculated from statistical errors and uncertainties of liquid standard, inlet and stripping efficiency and ozone interference is 25% for $H_2O_2$ and 40% for MHP.

## 3.2 Other *in situ* measurements

For this study CO, $CH_4$, OH, $HO_2$, $O_3$, Acetone, NO, $NO_y$, $J_{H2O2}$ and $J_{MHP}$ data measured by other instruments have been used for data interpretation, steady-state calculations and interference corrections (see section 3.1). A complete list of all measured compounds can be found in Lelieveld et al., 2018. CO and $CH_4$ have been measured by the IR-quantum cascade laser absorption spectrometer TRISTAR (Schiller et al., 2008; Tadic et al., 2017). The measurements comprised an ambient air mode and in-flight calibrations. The latter were realized with secondary standards from pressurized bottles (6 L bottle,

Auer GmbH, Germany), which were calibrated against certified reference gases (Tomsche et al., 2019). With the help of the in-flight calibrations the *in situ* data are drift-corrected by interpolation between two calibrations (Tadic et al., 2017). The observed CO and $CH_4$ mixing ratios have a total uncertainty of 5.1% and 0.275%, respectively. The relatively high CO uncertainty reflects problems with the stability of the CO quantum cascade laser during the second half of OMO.

Laser induced fluorescence was the method utilized for $HO_x$ measurements (instrument name: HORUS, Faloona et al., 2004;

Martinez et al., 2010). The accuracies of the measurements are 17.1% for OH and 17.6% for $HO_2$. The limit of detection of the instrument does vary depending on altitude as this system has a sensitivity that depends on pressure. As altitude increases the LOD decreases from 0.1 ppt$_v$ to 0.02 ppt$_v$ for OH and 0.361 ppt$_v$ to 0.175 ppt$_v$ for $HO_2$.

FAIRO (Fast AIRborne Ozone instrument) is a light-weight (14.5 kg) and accurate 2-sensor device for measuring $O_3$. It combines two techniques, i.e. (a) a UV photometer that measures the light absorption by $O_3$ at a wavelength of $\lambda = 250$–

260 nm emitted by a UV-LED and (b) a chemiluminescence detector that monitors the chemiluminescence generated by $O_3$ on the surface of an organic dye adsorbed on dry silica gel. These techniques are simultaneously applied in order to combine the high measurement accuracy of the UV photometry with the high measurement frequency of the chemiluminescence detection. The UV photometer has a 1-$\sigma$ precision of 0.08 ppb$_v$ at a measurement frequency of 0.25 Hz (and a pressure of 1 bar) and an accuracy of 1.5% (determined by the uncertainty of the $O_3$ cross section). The chemiluminescence detector has

a precision of 0.05 ppb$_v$ at a measurement frequency of 12.5 Hz (Zahn et al., 2012). In post-processing the chemiluminescence detector data is calibrated using the UV photometer data.

Nitrogen oxide (NO) and total reactive nitrogen ($NO_y$) were measured using the AENEAS-atmospheric nitrogen oxides measuring system. The measurements were performed by a dual channel NO-chemiluminescence detector (CLD-SR 790, Eco Physics, Switzerland) in combination with a converter technique for the detection of total reactive nitrogen as NO. $NO_y$

comprises among others NO, $NO_2$, $HNO_3$, $NO_3$, $N_2O_5$, $HNO_2$, $HO_2NO_2$, PAN and organic nitrates. The individual $NO_y$ species were detected after conversion to NO using a gold tube maintained at about 300 °C with $H_2$ as a reducing agent (Ziereis et al., 2000). Ambient air was sampled using a standard HALO trace gas inlet equipped with a heated (~ 40 °C) PFA inlet line. The time resolution of the measurements was about 1 s. The overall uncertainty of the NO and $NO_y$ measurements depends on its ambient concentrations and is about 8% (6.5%) for volume mixing ratios of 0.5 nmol/mol (1 nmol/mol),

respectively (Stratmann et al., 2016).

VOCs (e.g. acetone) were measured with a homebuilt light-weight (~55 kg without rack) proton-transfer-reaction mass spectrometer which uses a commercial quadrupole mass analyzer (Pfeiffer, QMA 410, Germany). A modular V25 micro computer system (MPI-C, Mainz, Germany) is applied for instrument control and data acquisition. A custom-built inlet system comprises a platinum/quartz wool scrubber (Shimadzu, High Sensitivity Catalyst) held at 300 °C and components for

flow and pressure control. The instrument was calibrated between flights with a dynamically diluted gas standard containing approximately 500 ppb$_v$ of VOCs (Apel-Riemer Environmental Inc., USA). The accuracy for acetone is typically ±10% and the detection limit is ~60 ppt$_v$.

Photolysis frequencies were calculated from spectral actinic flux density spectra (280–650 nm) obtained from CCD spectroradiometer measurements on the top and bottom fuselage of the aircraft covering the upper and the lower hemisphere,

respectively (Bohn and Lohse, 2017). Recent recommendations of absorption cross sections and quantum yields were used in the calculations, as well as their temperature and pressure dependencies (if available) by taking into account measured static air temperatures and pressures. Radiometric uncertainties range around 5–6% under typical flight conditions. Additional uncertainties related to the molecular parameters are process specific. For $H_2O_2$ in particular, recommended absorption cross sections and their temperature dependencies were applied and unity quantum yields were assumed (Burkholder et al., 2015).

However, the recommended $H_2O_2$ absorption cross sections are confined to a wavelength range below 350 nm which is insufficient to capture atmospheric photolysis completely. Because measured cross sections decay exponentially over two orders of magnitude in the range 280–350 nm, this dependence was further extrapolated up to 370 nm where values drop well below $10^{-22}$ cm$^2$. Dependent on conditions this extrapolation increases atmospheric $H_2O_2$ photolysis frequencies by 10–

20%. For MHP the temperature dependence of the absorption cross sections is unknown. Therefore the recommended room temperature data were used under all conditions as well as unity quantum yields (Burkholder et al., 2015). Combined total uncertainties of 15% and 25% are estimated for $H_2O_2$ and MHP photolysis frequencies, respectively.

Latitude, longitude and altitude data as well as temperature and pressure were collected with the BAHAMAS (BAsic HALO Measurement And Sensor system) instrument. More detailed information about the installation of scientific instruments and mission flights can be found on http://www.halo.dlr.de/science/missions/omo/omo.html.

### 3.3 Photo-stationary state calculations

Since only the sum of organic hydroperoxides was measured we estimated the contribution of MHP using a photo-stationary-state (PSS) approximation relying on *in situ* measurements of $HO_2$, OH, CO, $CH_4$, NO, $J_{MHP}$ and $J_{H2O2}$ (see 3.2) and rate coefficient data from Atkinson et al., 2004 and Atkinson et al., 2006).

In the free troposphere the production rate P of $H_2O_2$ and MHP is due to the self-reaction of $HO_2$ and reaction of $CH_3O_2$ with $HO_2$, respectively, and can be calculated from Eq. 1 and 2.

$$P(H_2O_2) = k_{HO_2+HO_2} \cdot [HO_2]^2 \,, \tag{1}$$

$$P(MHP) = k_{CH_3O_2+HO_2} \cdot [CH_3O_2] \cdot [HO_2] \,, \tag{2}$$

Photochemical loss rates L of $H_2O_2$ and MHP are due to photolysis and reaction with OH according to Eq. 3 and 4.

$$L(H_2O_2) = \left(k_{H_2O_2+OH} \cdot [OH] + J_{H_2O_2}\right) \cdot [H_2O_2] \,, \tag{3}$$

$$L(MHP) = \left(k_{MHP+OH} \cdot [OH] + J_{MHP}\right) \cdot [MHP] \,, \tag{4}$$

For steady-state conditions the production and loss reactions are at equilibrium and the $MHP^{PSS}$ to $H_2O_2^{obs}$ ratio can be calculated from Eq. 5.

$$\frac{[MHP]^{PSS}}{[H_2O_2]^{obs}} = \frac{k_{CH_3O_2+HO_2} \cdot [CH_3O_2] \cdot [HO_2]}{k_{HO_2+HO_2} \cdot [HO_2]^2} \cdot \frac{k_{H_2O_2+OH} \cdot [OH] + J_{H_2O_2}}{k_{MHP+OH} \cdot [OH] + J_{MHP}} \,, \tag{5}$$

Because individual peroxy radicals were not measured, the $CH_3O_2$ to $HO_2$ ratio must be estimated from their production and loss terms. This ratio can be deduced as written in Eq. 6.

$$\frac{[CH_3O_2]}{[HO_2]} = \frac{L(HO_2) \cdot P(CH_3O_2)}{P(HO_2) \cdot L(CH_3O_2)} \,, \tag{6}$$

Dominant loss processes for $HO_2$ and $CH_3O_2$ are reactions with NO and the production of $H_2O_2$ and MHP, respectively, neglecting the production of peroxy nitrates due to low $NO_2$ concentrations in the UT (Eq. 7 and 8).

$$L(HO_2) = k_{CH_3O_2+HO_2} \cdot [CH_3O_2] \cdot [HO_2] + k_{HO_2+NO} \cdot [HO_2] \cdot [NO] + k_{HO_2+HO_2} \cdot [HO_2]^2 \,, \tag{7}$$

$$L(CH_3O_2) = k_{CH_3O_2+HO_2} \cdot [CH_3O_2] \cdot [HO_2] + k_{CH_3O_2+NO} \cdot [CH_3O_2] \cdot [NO] \,, \tag{8}$$

The first terms on the right side of both equations are identical. The second terms are dominated by the rate coefficients of the reactions with NO and the NO concentration. For the calculations of the rate coefficients the mean temperature of 259.18 K, the mean altitude of 10,992.8 m and the mean pressure of 22,932.9 Pa were used. The resulting values are shown in Eq. 9–11. As the relative humidity is very low in the upper troposphere the water dependence in eq. 11 was neglected.

$$k_{HO_2+NO} = 3.45 \cdot 10^{-12} \cdot \exp^{\frac{270}{T}} = 9.78 \cdot 10^{-12} \frac{cm^3}{molecule \cdot s} \; , \tag{9}$$

$$k_{CH_3O_2+NO} = 2.3 \cdot 10^{-12} \cdot \exp^{\frac{360}{T}} = 9.22 \cdot 10^{-12} \frac{cm^3}{molecule \cdot s} \; , \tag{10}$$

$$k_{HO_2+HO_2} = 2.2 \cdot 10^{-13} \cdot \exp^{\frac{600}{T}} + 1.9 \cdot 10^{-33} \cdot [N_2] \cdot \exp^{\frac{980}{T}} = 2.64 \cdot 10^{-12} \frac{cm^3}{molecule \cdot s} \; , \tag{11}$$

This indicates that the reaction of $HO_2$ with NO is more than a factor of 3 faster than the self-reaction. The measured NO concentration is an order of magnitude larger than measured $HO_2$, so that reaction with NO is the dominant process for both

$HO_2$ and $CH_3O_2$ resulting in similar loss rates for both radicals in the UT. Thus, the ratio of $CH_3O_2$ to $HO_2$ is dominated by their production rates (Eq. 12).

$$\frac{[CH_3O_2]}{[HO_2]} = \frac{P(CH_3O_2)}{P(HO_2)} = \frac{k_{CH_4+OH} \cdot [CH_4] \cdot [OH]}{k_{CO+OH} \cdot [CO] \cdot [OH]} \; , \tag{12}$$

The combination of Eq. 5 and 12 results in Eq. 13 which was used to calculate the MHP$^{PSS}$ concentrations based on the observed mixing ratios during OMO.

$$[MHP]^{PSS} = \frac{k_{CH_3O_2+HO_2}}{k_{HO_2+HO_2}} \cdot \frac{k_{H_2O_2+OH} \cdot [OH]^{obs} + J_{H_2O_2}{}^{obs}}{k_{MHP+OH} \cdot [OH]^{obs} + J_{MHP}{}^{obs}} \cdot \frac{k_{CH_4+OH} \cdot [CH_4]^{obs}}{k_{CO+OH} \cdot [CO]^{obs}} \cdot [H_2O_2]^{obs} \; , \tag{13}$$

Please note that other sources of $HO_2$ and $CH_3O_2$, in particular the photolysis of formaldehyde (HCHO) and acetaldehyde, respectively have been neglected. This is justified by the generally low mixing ratios of these species at high altitudes. Measurements of HCHO with the TRISTAR instrument yielded values below the detection limit of 30 pptv, and although acetaldehyde was not measured, we assume that its mixing ratio is within a factor of two of those for HCHO.

The total uncertainty of MHP$^{PSS}$ from the calculation according to equation 13 can be deduced from error propagation taking into account uncertainties in OH$^{obs}$ (17.1%), $J_{H2O2}{}^{obs}$ (15%), $J_{MHP}{}^{obs}$ (25%), $CH_4{}^{obs}$ (0.275%), CO$^{obs}$ (5.1%), $H_2O_2{}^{obs}$ (25%) and rate constants, to be of the order of 45% (1σ).

To estimate the contribution of MHP to the total organic hydroperoxides the calculated concentration of MHP$^{PSS}$ was subtracted from the measured sum of all organic hydroperoxides ROOH$^{obs}$. This leads to a concentration of unidentified

organic hydroperoxides (UHP$^{PSS}$) (Eq. 14). Please note that MHP$^{PSS}$ only accounts for local $CH_4$ oxidation production and not for transport phenomena.

$$[UHP]^{PSS} = [ROOH]^{obs} - [MHP]^{PSS} \; , \tag{14}$$

### 3.4 $H_2O_2$ calculation

In order to classify the measured $H_2O_2{}^{obs}$, $HO_2{}^{obs}$ and OH$^{obs}$ data we calculated $H_2O_2{}^{PSS}$ from measured HO$_x$ and $J_{H2O2}{}^{obs}$. For

the calculation Eq. 15 was used which is based on Eq. 1 and Eq. 3.

$$[H_2O_2]^{PSS} = \frac{[HO_2]^{obs\,2} \cdot k_{HO_2+HO_2}}{[OH]^{obs} \cdot k_{H_2O_2+OH} + J_{H_2O_2}{}^{obs}} \; , \tag{15}$$

A total uncertainty of 45% (1σ) due to uncertainties in OH$^{obs}$ (17.1%), $HO_2{}^{obs}$ (17.6%), $J_{H2O2}{}^{obs}$ (15%) and reaction rate constants, was calculated.

### 3.5 Other research tools

The EMAC (ECHAM/MESSy Atmospheric Chemistry) model comprises the 5[th] generation of the European Center HAMburg (ECHAM5; Roeckner et al., 2006; version 5.3.01) general circulation model and the Modular Earth Submodel System (MESSy; Jöckel et al., 2016; version 2.52, http://www.messy-interface.org/). For this study EMAC simulations (T42L90, 2.8° x 2.8° horizontal resolution, 90 vertical levels to 0.01 hPa, time resolution 12 min) were sampled along the OMO flights tracks. Detailed specifications and results have been published previously (Lelieveld et al., 2018; Tomsche et al., 2019).

Ten days back-trajectories were calculated along the flight path using FLEXPART to identify the air mass origin (Tomsche et al., 2019). Convective transport can be simulated in FLEXPART with the convection parameterization by Emanuel K. A. and Zivkovic-Rothman M., 1999. To represent moist convection realistically in models, the parametrization includes cloud microphysical processes, the physics of entrainment and mixing, as well as large scale control of ensemble convective activity. It builds on temperature and humidity fields to provide mass flux information (Stohl et al., 2005). The back trajectories in the present paper are calculated with the convective parametrization. Further the Lagrangian particle dispersion model FLEXPART produces so called centroid trajectories, based on the analysis of a cluster of trajectories. These trajectories are comparable to traditional trajectories, but include convection via the centroid of all particles per time step. As indicated by Tomsche et al. prominent source regions of AMA air masses are identified to be the Indo Gangetic Plain, Northeast India, Bangladesh and the Bay of Bengal. Additionally, Tomsche et al. used observations of methane to differentiate between air masses influenced by the AMA and background air. A comparison of vertical profiles indicated that the air inside the AMA showed significantly higher $CH_4$ concentrations than outside. Thus a threshold of $CH_4 \geq 1879.8$ $ppb_v$ was used to distinguish between air masses influenced by the AMA ($CH_4 \geq 1879.8$ $ppb_v$), the SH background ($CH_4 < 1820$ $ppb_v$) and the NH background ($1820$ $ppb_v \leq CH_4 < 1879.8$ $ppb_v$) (Tomsche et al., 2019).

## 4 Results and discussion

### 4.1 Data processing

Data were collected from a merged data set given as 60-second-means (calculated from the original data set obtained at higher resolutions) in order to get the same time resolution for all compounds. The given time is the middle of the block mean.

For the histograms the concentrations of all species shown were binned into samples with a width of 10 $ppt_v$, starting the plots with the lowest bin. To compare the simulations from EMAC with measured and PSS calculated data, the corresponding values (out of the 60-second-means) were used at the given times from EMAC.

**4.2 Case study: Flight 17 from Gan to Bahrain (10.08.2015)**

In a case study analyzing flight 17 from 10[th] of august 2015, the method used to determine the origin of the measured air masses and a quantification and comparison of measured and simulated mixing ratios of $H_2O_2^{obs}$, $MHP^{PSS}$ and $UHP^{PSS}$ is presented. During this flight we encountered the SH and NH as well as the AMA in the UT. The flight track is shown in Figure 2 (dotted line). Take-off was in Gan (Maldives) and landing in Bahrain on the Arabian Peninsula. Besides take-off and landing, the entire flight took place in the UT (<230 hPa). The calculated back trajectories show the origin of the air masses. At the beginning of the flight the measured air masses had their origin over the Indian Ocean and Indonesia. During the remaining flight the measured air stemmed from India. Tomsche et al. 2019 showed that the measured air in the AMA was affected by deep convection over India resulting in methane mixing ratios above the threshold. Figure 3 shows the time series for $H_2O_2^{obs}$ during the flight at the time steps given from the frequency of EMAC output (orange circles). The colored bar on top shows the origin of air masses, i.e. red for AMA, green for NH and blue for SH. The $H_2O_2^{obs}$ mixing ratios vary between 128–366 ppt$_v$. The modelled $H_2O_2^{EMAC}$ data are in the range of 110–799 ppt$_v$ (grey triangles). For the beginning of the flight model simulations agree rather well with the measurement data. At around 6:00 UTC the model calculated mixing ratios increase to more than 500 ppt$_v$, while the measured data decrease to 200 ppt$_v$ and lower. In this period a maximum difference between model and *in situ* data of 386 ppt$_v$ was found. One hour later the EMAC model data decrease to 416 ppt$_v$ and the *in situ* data increase to 214 ppt$_v$. During the following hour until around 8:00 UTC and thus at the higher altitude, both mixing ratios increase with the modelled data showing a much stronger increase up to approximately 800 ppt$_v$ while the *in situ* data increase only to 230 ppt$_v$. During the last period of the flight, simulated and measured data are again in good agreement. Here the mixing ratios from EMAC are in the range of 110–157 ppt$_v$ while the measured data are in the range of 128–203 ppt$_v$. This steep drop of $H_2O_2^{EMAC}$ might arise due to a change in the flight altitude, since between 8:01–8:06 UTC the aircraft changed from a flight level at 11,700 m to one aloft at 13,900 m.

In addition to EMAC simulations Figure 3 also shows the calculated $H_2O_2^{PSS}$ obtained from equation 15. Observed and PSS values for $H_2O_2$ mixing ratios agree very well, with a median deviation of 42 ppt$_v$, well within the combined uncertainties of measured data (25%) and PSS simulations (45%).

In Figure 4 the time series of $H_2O_2^{obs}$, $MHP^{PSS}$ and $UHP^{PSS}$ mixing ratios are shown (5 min means). In the beginning of the flight $MHP^{PSS}$ is the dominant organic hydroperoxide. The mixing ratios are in the range of 140–341 ppt$_v$ similar to those of $H_2O_2^{obs}$ (143–337 ppt$_v$). $UHP^{PSS}$ mixing ratios are in the range of 24–162 ppt$_v$ with a mean of 89 ppt$_v$. Later during the flight $UHP^{PSS}$ are the dominant hydroperoxides in air masses inside the AMA originating from India. Here we found $UHP^{PSS}$ mixing ratios up to 275 ppt$_v$. The $H_2O_2^{obs}$ mixing ratios show a similar temporal pattern and mixing ratio levels to those of $UHP^{PSS}$ over the Arabian Sea and the Arabian Peninsula, with values in the range of 140–243 ppt$_v$. $MHP^{PSS}$ mixing ratios are much lower (62–130 ppt$_v$, median 72 ppt$_v$) in this area. During this part of the flight the similarity in the time series of acetone and $UHP^{PSS}$ (Figure 5), indicate either similar source regions for both species, or the role of acetone as a precursor for the $UHP^{PSS}$ in AMA influenced air masses. From 6:20 UTC onwards an increase in both compounds is observed until

7:45 UTC, followed by a steep drop with a minimum at 8:17 UTC. The UHP$^{PSS}$ and acetone mixing ratios in this part of the flight are strongly correlated (Figure 6), with a slope of 0.19±0.02 (ppb$_v$/ppb$_v$) and an offset of (-0.003±0.02) ppb$_v$. The regression coefficient $R^2$ is very high (0.82). For H$_2$O$_2$$^{obs}$, MHP$^{PSS}$ and ROOH$^{obs}$ the correlation is not that strong with slopes of -0.02±0.02 (ppb$_v$/ppb$_v$), -0.07±0.01 (ppb$_v$/ppb$_v$) and 0.13±0.03 (ppb$_v$/ppb$_v$) respectively and offsets of (0.21±0.02) ppb$_v$.

(0.12±0.01) ppb$_v$ and (0.11±0.03) ppb$_v$ (Figure 6). The relation between ROOH mixing ratios and an air mass age tracer based on the ratio between [NO] to [NO$_y$] shows higher values of ROOH at smaller ratios representing older or more processed air masses (Figure 7), since highest ROOH mixing ratios (>200 ppt$_v$) are found at the lowest [NO]/[NO$_y$] ratios (all <0.19). Thus, most of the observed ROOH was measured in aged air masses transported within the anticyclone. The correlation with UHP$^{PSS}$ shows that this effect is mainly due to UHP$^{PSS}$. For H$_2$O$_2$$^{obs}$ there are also some higher mixing ratios

for high [NO] to [NO$_y$] mixing ratios and thus fresher air (Figure 7).

### 4.3 Results for the entire campaign

To extend the analysis to the entire campaign, Figure 8 shows all flight tracks in the UT during OMO. The color-code represents observed mixing ratios of H$_2$O$_2$$^{obs}$, MHP$^{PSS}$ and UHP$^{PSS}$ varying from low (purple) to high values (red). Histograms for the whole campaign of H$_2$O$_2$$^{obs}$ mixing ratios as well inferred MHP$^{PSS}$ and UHP$^{PSS}$ mixing ratios are

presented in Figure 9. Here only data from the UT (<300 hPa which corresponds to altitudes >9 km) were included in the analysis. Mixing ratios for all species were further differentiated by methane levels, such that data in air masses with CH$_4$ mixing ratios above the threshold of 1879.8 ppb$_v$ were classified as AMA influenced, while air masses with a CH$_4$ mixing ratios between 1820 ppb$_v$ and 1879.8 ppb$_v$ were classified as NH background and those with CH$_4$ <1820 ppb$_v$ as SH following Tomsche et al., 2019). The upper panel indicates that H$_2$O$_2$$^{obs}$ mixing ratios are most abundant at values of 70–

90 ppt$_v$ in NH background air masses (green), 130 ppt$_v$ to 270 ppt$_v$ with three notable peaks at 180–190 ppt$_v$, 210–220 ppt$_v$ and 250–270 ppt$_v$ in SH air masses (blue) and 150–170 ppt$_v$ in AMA influenced air masses (red). The medians are 115 ppt$_v$ for the NH background, 211 ppt$_v$ for the SH and 167 ppt$_v$ for the AMA, indicating an excess of 52 ppt$_v$ in AMA influenced air masses compared to the NH background, while in the SH the H$_2$O$_2$$^{obs}$ mixing ratio is twice as high as the NH background. For MHP$^{PSS}$ (Figure 9, middle panel) the frequency distribution in the NH background shows a maximum at 30–40 ppt$_v$

(green). For AMA influenced air a sharp maximum at 50–70 ppt$_v$ (red) is found. Air masses from the SH exhibit a rather flat distribution with a maximum at values of 40–50 ppt$_v$ and a median of 152 ppt$_v$ (blue). With median mixing ratios of 70 ppt$_v$ and 64 ppt$_v$ we found only slightly higher mixing ratios for AMA influenced air masses in comparison to the NH background. For UHP$^{PSS}$ (Figure 9, bottom panel) we again found a flat distribution of mixing ratios in the SH (blue) with a maximum at values of 140–150 ppt$_v$ and a median of 129 ppt$_v$. The maximum in the frequency distribution for NH

background conditions is found at 70–90 ppt$_v$ (green), while in AMA influenced air masses it was significantly higher with 210–230 ppt$_v$ (red). Thus, mixing ratios of UHP$^{PSS}$ are approximately 2–3 times higher in the AMA outflow than in the NH background. This is also represented in the medians of 210 ppt$_v$ in the AMA and 89 ppt$_v$ in the NH background.

In the analysis of flight 17 we found a strong correlation between $UHP^{PSS}$ and acetone (Figure 6) and an increase of $UHP^{PSS}$ at the highest air mass ages, represented by low $[NO]/[NO_y]$ (Figure 7). Extension of this analysis to all observations in the
upper troposphere obtained during OMO yields similar results for the relation between $UHP^{PSS}$, $ROOH^{obs}$ and $H_2O_2^{obs}$ and acetone (Figure 10). Enhanced mixing ratios of hydroperoxides are typically associated with enhanced acetone mixing ratios, especially for $UHP^{PSS}$. A simple calculation of the production of MHP out of the photolysis of acetone and the reaction of acetaldehyde (from EMAC) with OH shows that per day appr. 40 $ppt_v$ MHP can be formed within the AMA. The lifetime of MHP was calculated to be around 1.5 days. Thus not all of the $UHP^{PSS}$ in the AMA (median 210 $ppt_v$) can be accounted for
MHP that was chemically produced from VOCs in the AMA. The scatter plots of the hydroperoxides vs. $[NO]/[NO_y]$ for the whole data set, show no clear correlation with a large spread of hydroperoxides mixing ratios at the lowest $[NO]/[NO_y]$ ratios, representing the oldest, i.e. chemically most processed air masses (Figure 11).

### 4.3.1 $H_2O_2$ steady-state calculation

A scatter plot of the results from the $H_2O_2^{PSS}$ based on observed $HO_x$ data in the UT (eq. 15) is shown in Figure 12. The
black dotted line shows the 1:1 line, the green dashed lines represent the 2:1 and 1:2 relations. It is obvious that the comparison is affected by a rather large offset of approximately 350 $ppt_v$ in the observations that is not accounted for in the steady-state calculations. The regression coefficient $R^2$ is 0.26. Most of the $H_2O_2^{PSS}$ mixing ratios (75%) vary between 0 and 65 $ppt_v$ with a median value of 15 $ppt_v$, while the $H_2O_2^{obs}$ extend over a larger range mainly between 10–210 $ppt_v$ with a median of 150 $ppt_v$, and thus 10 times higher than for steady-state, indicating that more than 80% of all points in the
correlation are outside the range of uncertainty. This can also be seen in the histograms in Figure 13. Table 1 shows the statistical comparison of both data sets. The discrepancy between $H_2O_2^{obs}$ and $H_2O_2^{PSS}$ shows that the local PSS does not account all main contributions of $H_2O_2$ even though all chemical reactions are included. Thus transport phenomena like deep convection seem to play a key role (see 4.3.3).

### 4.3.2 Comparison to EMAC

Figure 14 shows histograms for the comparison between $H_2O_2^{obs}$, $MHP^{PSS}$ and $UHP^{PSS}$ with EMAC simulations. Median $H_2O_2^{EMAC}$ values are similar for NH background (66 $ppt_v$) and AMA (71 $ppt_v$) conditions (Table 2), while $H_2O_2^{obs}$ indicate an enhancement of +64 $ppt_v$ in the AMA relative to the NH background. For the SH the model simulated $H_2O_2^{EMAC}$ (272 $ppt_v$) mixing ratios are four times higher than in the NH background (66 $ppt_v$), while the $H_2O_2^{obs}$ only show a median increase by roughly a factor of 2 (100 $ppt_v$ to 211 $ppt_v$) (Table 2).
In general EMAC tends to strongly underestimate total hydroperoxide in all air masses by a factor of 5–10. $MHP^{EMAC}$ mixing ratios are mainly lower than 50 $ppt_v$ for background and AMA, while $MHP^{PSS}$ ranges from LOD–140 $ppt_v$. Again the model simulates highest $MHP^{EMAC}$ mixing ratios in the SH with values up to 502 $ppt_v$ compared to up to 346 $ppt_v$ in the $MHP^{PSS}$ calculations. Similar as for $H_2O_2^{EMAC}$, medians of $MHP^{EMAC}$ for NH background and AMA conditions are show very small differences (11 $ppt_v$ and 13 $ppt_v$ respectively, Table 2), while slightly higher differences were found for $UHP^{EMAC}$

in the AMA (64 $ppt_v$ and 70 $ppt_v$, respectively). In the simulations, southern hemispheric $MHP^{EMAC}$ mixing ratios are almost ten times higher than NH background values (116 $ppt_v$ and 11 $ppt_v$, respectively), compared to two to three times higher in the observations.

Data for $UHP^{EMAC}$ in the model are calculated from the sum of simulated ethyl hydroperoxide (EHP) and peroxyacetic acid, which are the only non-methyl organic hydroperoxides in the free troposphere according to the model with non-zero mixing ratios. $UHP^{EMAC}$ mixing ratios range from 1–238 $ppt_v$ in the NH background, 1–259 $ppt_v$ in the AMA and 1–132 $ppt_v$ in the SH. $UHP^{PSS}$ based on the observations indicate lowest mixing ratios in the NH background (LOD–261 $ppt_v$), while in the AMA and the SH the ranges are quite similar (80–311 $ppt_v$ and LOD–334 $ppt_v$). A comparison of median values emphasizes the large difference between model simulations and observation based estimates. In the NH background, the median $UHP^{PSS}$ mixing ratio from the observations is 70 $ppt_v$ higher than EMAC simulations (78 $ppt_v$ and 8 $ppt_v$ respectively). In the AMA the difference is even larger, with about 200 $ppt_v$ higher $UHP^{PSS}$ levels compared to the EMAC simulations. The smallest difference with 89 $ppt_v$ was found for the SH (Table 2).

### 4.3.3 Longitudinal gradients

So far discussions of different air masses have been based on measurements of methane, subdividing the observations in NH background, AMA and SH data. Tomsche et al., 2019) have shown that longitudinal gradients are found in the AMA over the Arabian Peninsula. Observations in the west are often near the edge of the anticyclone, while observations towards the east are closer to its center. In Figure 16 observations, steady-state calculations and EMAC simulations for upper tropospheric (9–15 km) $H_2O_2$ are displayed as a function of longitude from west to east (20–30 °N, 36–60 °E, according to the red box in Figure 15). To identify gradients, the data are subdivided into bins of 2° longitude. The observations (orange) show roughly a 100% increase of $H_2O_2^{obs}$ from west to east (90 $ppt_v$ to 175 $ppt_v$), similar to simulation with EMAC (black), although absolute mixing ratio levels in $H_2O_2^{EMAC}$ are smaller (61 $ppt_v$ to 121 $ppt_v$). Contrary to these observed gradients, $H_2O_2^{PSS}$ based on HORUS data (blue) do not vary with longitude, except for the last two bins. The steady-state calculations are based exclusively on observed concentrations of $HO_2$ and OH radicals and thus yield only the net photochemical production, while the EMAC simulations and the observations will also account for vertical and horizontal advection from up-wind source regions. Previous studies show inconsistent results. Snow et al. (2007) and Barth et al. (2016) for example both show that $H_2O_2$ is depleted in convective outflow compared to background upper troposphere. In contrast, other studies found that deep convection can be a source of $H_2O_2$ in the upper troposphere (e.g. Jaeglé et al., 1997; Prather and Jacob, 1997; Mari et al., 2003; Bozem et al., 2017). Similarly, convection over India during the summer monsoon is a potential source of excess $H_2O_2$ in the upper troposphere. With a photochemical lifetime of several days, this excess in $H_2O_2$ reaches the western AMA, giving rise to the observed and model simulated longitudinal gradients. Since the steady-state calculations do not account for transport this can explain the rather large deviation of 144–164 $ppt_v$ (between 51° and 57°) with the observations. Differences between observation and EMAC simulation could potentially arise due to uncertainties in the scavenging efficiency for $H_2O_2$, as the chemistry does not seem to be a dominant cause of uncertainty.

Similar longitudinal gradients are also observed for measured total organic hydroperoxides (ROOH$^{obs}$, green asterisks in Figure 17), inferred UHP$^{PSS}$ (black) as well as total ROOH$^{EMAC}$ (blue). Steady-state calculations of MHP$^{PSS}$ (pink) and simulations of MHP$^{EMAC}$ (yellow) show either no, or only weak longitudinal gradients. Assuming that MHP is also enhanced in the outflow of deep convection (Mari et al., 2000; Barth et al., 2016) at least part of the enhancement in ROOH$^{obs}$ (and thus inferred UHP$^{PSS}$) could be due to advected MHP.

## 4.4 Discussion

To our knowledge we present the first observations of $H_2O_2$ and ROOH mixing ratios in the Asian Monsoon Anticyclone. Previous studies have been mainly focused on the northern hemispheric upper troposphere. Several aircraft campaigns including peroxide measurements were performed over North America. They are summarized in Snow et al., 2007): The SONEX campaign took place in fall 1997 in the UT and yielded mean values of 120 ppt$_v$ for $H_2O_2$ and 50 ppt$_v$ for MHP (medians: 80 ppt$_v$ and 30 ppt$_v$, respectively). The TOPSE campaign in winter/spring 2000 probed the middle troposphere yielding median $H_2O_2$ and MHP mixing ratios of 150 ppt$_v$ for both species. During the INTEX-NA campaign in summer 2004 observed median mixing ratios at altitudes of 6–10 km were about 400 ppt$_v$ for $H_2O_2$, and 200 ppt$_v$ for MHP. A comparison with our results (Table 2) shows that we found similar mixing ratios as in SONEX in the northern hemispheric background of 115 ppt$_v$ and 64 ppt$_v$ for $H_2O_2$ and MHP, respectively. Mixing ratios for both species reported for TOPSE and INTEX-NA are slightly higher than ours, which may be related to the lower altitude range of 6–10 km (in comparison to >9 km for OMO) in these studies. Previous observations have shown that $H_2O_2$ and MHP show highest mixing ratios at altitudes between 2–5 km followed by a sharp decrease towards higher altitudes (see e.g. Daum et al., 1990; Heikes, 1992; Weinstein-Lloyd et al., 1998; Snow, 2003; Snow et al., 2007; Klippel et al., 2011).

Heikes et al. associated enhanced $H_2O_2$ mixing ratios above 5 km in the North Pacific of the Asian coast (30 °N) with outflow from the typhoon Mireille (Heikes et al., 1996). These observations were made close to the source region for the AMA influenced air masses described here (see back trajectories in the case study of flight 17, Figure 2 or Tomsche et al., 2019). For MHP Heikes et al. (1996) found mixing ratios of 250–500 ppt$_v$ in the southern longitudinal section above 5 km, similar to median mixing ratios of 152 ppt$_v$ for MHP in SH air masses in the UT found in this study.

Although the mixing ratios observed during this study are similar to previous observations in the upper troposphere, one striking result is that a state-of-the-art global circulation model (EMAC) and a local steady-state calculation constrained by measured radical levels significantly underestimate $H_2O_2$ mixing ratios in particular in the AMA. The general tendency is that the steady-state model produces the lowest values, with EMAC falling in between steady-state and observations (e.g. Figure 16). A comparison of the EMAC simulations for the two radicals that affect $H_2O_2$ most strongly (OH and $HO_2$) yields a rather good agreement. A scatter plot between modelled and observed $HO_2$ yields a slope of 0.72±0.01 (ppt$_v$/ppt$_v$) and an offset of (4.30±0.09) ppt$_v$ with a regression coefficient $R^2$ of 0.58 (Figure 18 left). The OH data show more scatter with a tendency for EMAC to overestimate the mixing ratios (slope: 1.7±0.2 (ppt$_v$/ppt$_v$); offset: (-0.1±0.1) ppt$_v$; regression coefficient $R^2$: 0.09, Figure 18 right). Although there is rather good agreement between EMAC simulations and observations

for all the species that affect the local photochemical budget of $H_2O_2$, EMAC significantly exceeds PSS calculation for $H_2O_2$. This is an indication that an additional $H_2O_2$ source is accounted for in the global model and that the local photo-stationary-state assumption is not fulfilled. The additional source is attributed to transport associated with deep convection over India, yielding in an upwind source of $H_2O_2$ that is significant throughout the western part of the AMA. In the AMA, clouds are absent, so that gas phase photochemical processes may determine the lifetime of $H_2O_2$. Based on observed $OH^{obs}$ levels and photolysis frequencies during OMO the $H_2O_2$ lifetime in the upper troposphere is of the order of several days, sufficiently long for the excess $H_2O_2$ to reach the western parts of the AMA, producing the observed longitudinal $H_2O_2$ gradient observed in both observations and EMAC simulations (Figure 16). The total amount of $H_2O_2$ injected into the UT by convective outflow depends on the scavenging efficiency (Mari et al., 2000; Barth et al., 2016; Bozem et al., 2017). Differences between $H_2O_2^{obs}$ and $H_2O_2^{EMAC}$ are most likely due to an overestimation of scavenging in the model as also pointed out by Klippel et al., 2011).

To investigate this assumption we performed a sensitivity study with the wet scavenging for all soluble species being switched-off globally. The result is shown in Figure 19. The $H_2O_2^{EMAC}$ mixing ratios significantly increase with longitude by a factor of 3–4 and thus to the level of $H_2O_2^{obs}$. Please note that significant enhancements in $MHP^{EMAC}$ and $ROOH^{EMAC}$ were not found in the sensitivity study with switched-off scavenging, indicating that the strong underestimation by the model of these species is not due to an overestimation of wet removal in convective clouds. Instead, we found that EMAC underestimates ROOH in all air masses and not only in the AMA. The reasons for this underestimation are unknown. In a previous comparison of MHP observations and EMAC simulations over Europe for July 2007, Klippel et al. (2011) also reported a factor of 10 difference in the upper troposphere, while a comparison during the fall season (October 2006) yielded a rather good agreement (within a factor of 2).

There is a rather large uncertainty regarding the scavenging efficiency of MHP in deep convection (Barth et al., 2016). For the Trace A campaign Mari et al., 2000) found observed (modelled) enhancement ratios of postconvective to preconvective mixing ratios of 11 (9.5) for MHP and 1.9 (1.2) for $H_2O_2$. Such efficient transport in the Indian Summer Monsoon would yield a strong source of upper tropospheric MHP explaining the large enhancement of $ROOH^{obs}$ in the AMA described here. Please note that large enhancements of $MHP^{EMAC}$ and $ROOH^{EMAC}$ were not found in the sensitivity study with switched-off scavenging, indicating that the strong underestimation by the model of those species is not due to an overestimation of wet removal in convective clouds. It seems that a large part of the $UHP^{PSS}$ is actually MHP advected throughout the AMA after deep convective transport over India. In the EMAC simulations the transport of MHP is less efficient and thus $MHP^{EMAC}$ is lower than $MHP^{PSS}$ and $UHP^{PSS}$. Please note that EMAC has a general tendency to overestimate CO in the UT, especially for the NH background, while it tends to underestimate CH4 (Tomsche et al., 2019). The deviations in general are not significant, with the exception of CH4 in the AMA (Table 1 in Tomsche et al., 2019). As discussed in Tomsche et al. 2019 the CH4 mixing ratio in the AMA depends of the co-location of convection and underlying methane sources. The model resolution of 2.8° x 2.8° is not sufficient to resolve small-scale variations in both convection and CH4 source distribution. The H2O2 mixing ratio over the Indian sub-continent is not expected to show large spatial variations, since latitudinal

gradients are generally small (see e.g. Klippel et al., 2011). Therefore, we do not expect that the model resolution will have a strong influence on the deviation between H2O2$^{obs}$ and H2O2$^{EMAC}$. Another uncertainty arises from missing information on the absolute mixing ratios of H2O2, MHP and higher organic hydroperoxides in the inflow region of deep convection over India, since observations of these species in the boundary layer over India are not available. Note that the amount of hydrogenperoxide and organic hydroperoxides transported to the upper troposphere depends on the scavenging efficiency

and on the mixing ratios of the individual species in the inflow region (Barth et al., 2016, Bozem et al., 2017). Thus, an underestimation of hydroperoxides in the upper troposphere after convective injection can be either due to an underestimation of the scavenging efficiency for individual species, an underestimation of their mixing ratio in the inflow region or a combination of both, and might differ for individual hydroperoxides. Due to a lack of observations in the inflow and outflow region of convection over India this question cannot be resolved in this study.

## 5 Conclusion

Hydrogen peroxide and organic hydroperoxides were measured during the OMO campaign in the upper troposphere in NH background air over the western Mediterranean, the Asian Summer Monsoon Anticyclone over the Arabian Peninsula and the SH over the Maldives and the Indian Ocean in summer 2015. The observed mixing ratios for background conditions in the NH and SH are in line with previous studies described in the literature. A case study (of flight 17) revealed enhanced

$H_2O_2$$^{obs}$ and ROOH$^{obs}$ mixing ratios in the AMA relative to the NH background. Similar results are found for other flights throughout the campaign. The atmospheric chemistry-general circulation model EMAC slightly underestimates $H_2O_2$$^{EMAC}$ in the NH background (medians: 66 pptv vs. 100 pptv), significantly underestimates it in the AMA (medians: 71 pptv vs. 164 pptv), and overestimates it in the SH (medians: 272 pptv vs. 211 pptv). Steady-state calculations for $H_2O_2$$^{PSS}$ and MHP$^{PSS}$ based on observed precursors yield much lower values compared to $H_2O_2$$^{obs}$ and MHP$^{PSS}$ by roughly a factor of 3, in

particular in the AMA, resulting in a large contribution of an unidentified organic hydroperoxide (UHP$^{PSS}$) in air masses affected by the AMA. A comparison between EMAC simulations and HO$_x$ levels shows a good agreement indicating that deviations between $H_2O_2$$^{EMAC}$ and $H_2O_2$$^{PSS}$ levels are due to transport. Convective injection of $H_2O_2$ (and ROOH) into the upper troposphere over India most likely forms a pool of hydroperoxides in the upper troposphere that subsequently influences the western AMA, giving rise to a significant longitudinal gradient of $H_2O_2$ and ROOH mixing ratios, with

increasing values towards the center of the AMA. It is likely that next to an unidentified organic hydroperoxide (e.g. PAA) at least part of UHP$^{PSS}$ is due to additional MHP from an up-wind source. A sensitivity study using EMAC with no scavenging tends to reproduce the observed longitudinal gradients in $H_2O_2$, although it does not increase the level of ROOH. The reasons for this different behavior are unclear.

**Data availability**

The data are available from the HALO database (https://halo-db.pa.op.dlr.de/, last access: 7 November 2019).

**Author contributions**

BH and SH were responsible for $H_2O_2$ and ROOH measurements and data. BH conducted further data analysis and wrote the original draft of the paper in close cooperation with HF. $CH_4$ and CO data were provided by LT, $HO_x$ data by DM, MM and

HH, $O_3$ and acetone data by MN and AZ, photolysis frequencies by BB and NO and $NO_y$ data by HZ and GS. AP was responsible for the EMAC model simulations. JL was the principal investigator of the OMO mission. All authors were involved in the review and editing of the paper.

**Competing interests**

The authors declare that they have no conflict of interest.

**Acknowledgements**

We would like to thank all of the participants of the OMO mission, the German Aerospace Center (DLR), and EDT Offshore Ltd in Cyprus for their cooperation during the mission. We further thank Rainer Königstedt for installing the TRIHOP

instrument and Uwe Parchatka for supporting the measurements of CO and $CH_4$.

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

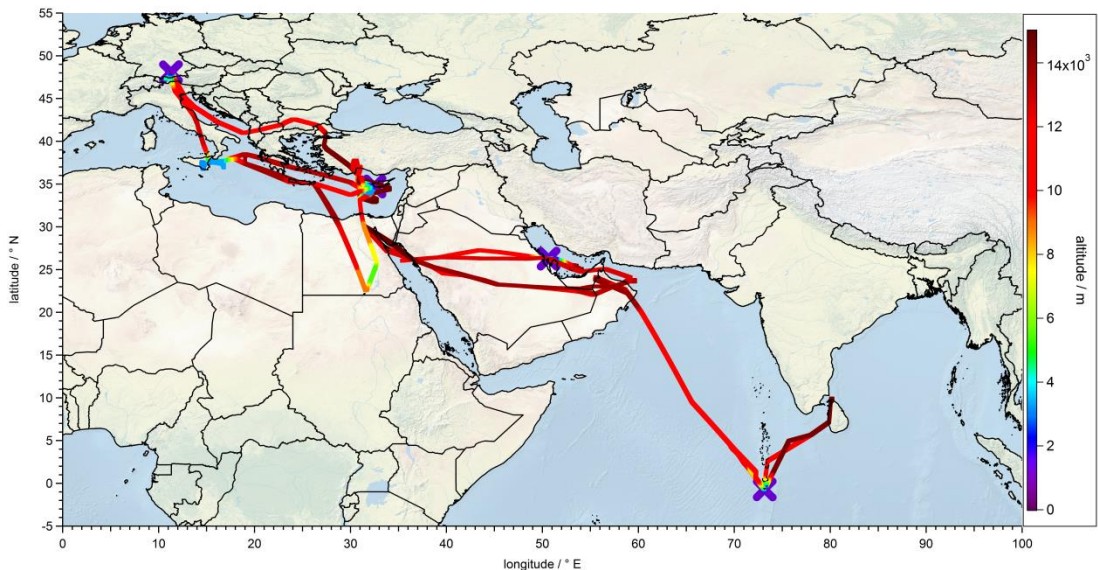

**Figure 1: Flight tracks (colors indicate altitude) and airports (purple crosses) used during the OMO campaign.**


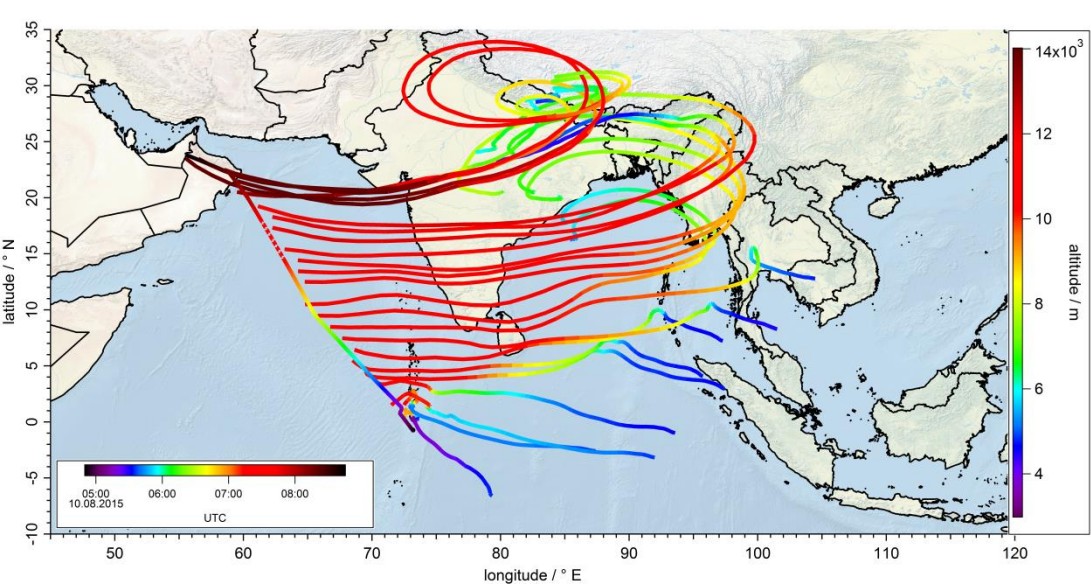

**Figure 2: Track of flight 17 (black dotted line) and calculated 10-day-back trajectories (lines colored as a function of altitude) to show the origin of sampled air masses during the flight.**

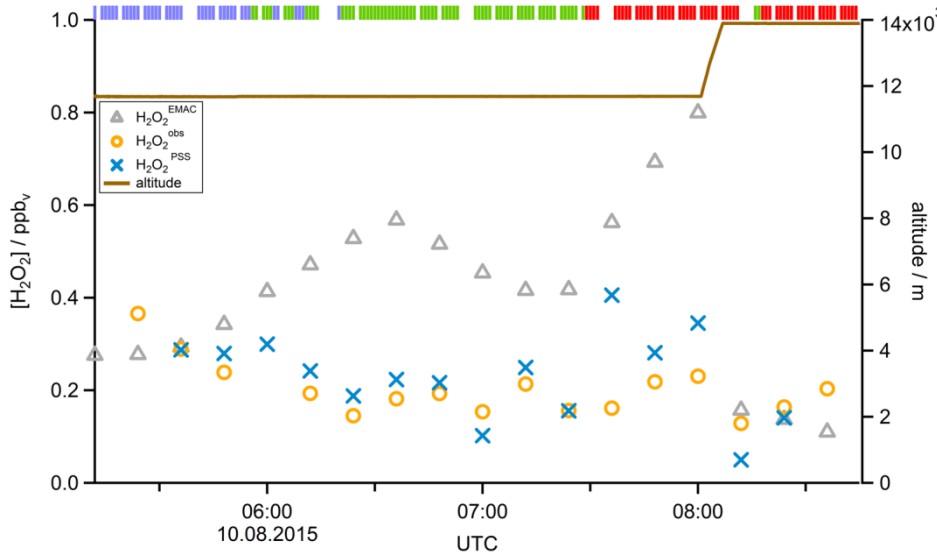

**Figure 3:** Time series of $H_2O_2^{obs}$ (orange circles), $H_2O_2^{PSS}$ (blue crosses) and $H_2O_2^{EMAC}$ (grey triangles) mixing ratios for flight 17. The brown line shows the altitude, the colored bar on top indicates the origin of air masses according to the methane mixing ratio classification: for SH blue, NH green and AMA red.

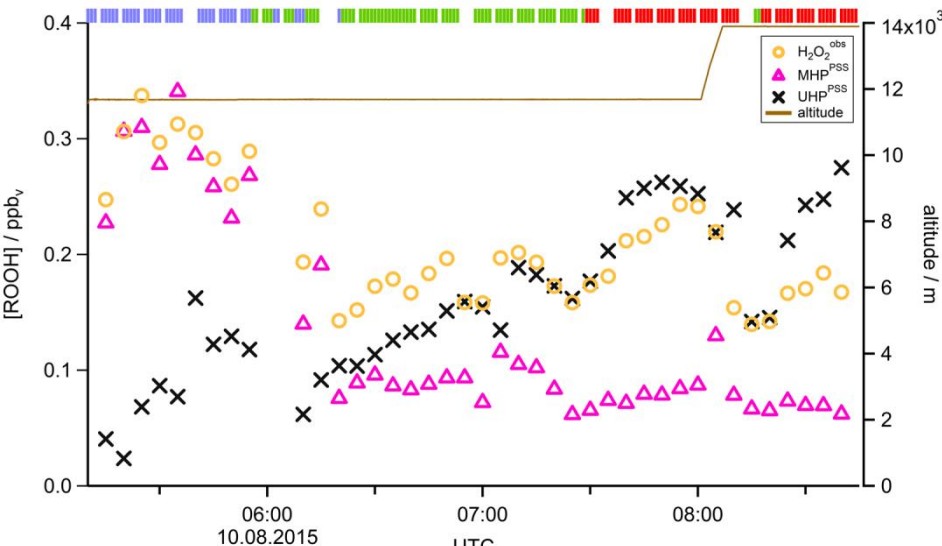

**Figure 4:** Time series of hydroperoxide mixing ratios during flight 17. The mixing ratios of $H_2O_2^{obs}$ (orange circles), $MHP^{PSS}$ (pink triangles) and $UHP^{PSS}$ (black crosses) are shown. The brown line shows the altitude, the colored bar on top indicates the origin of air masses according to the methane mixing ratio classification: for SH blue, NH green and AMA red.

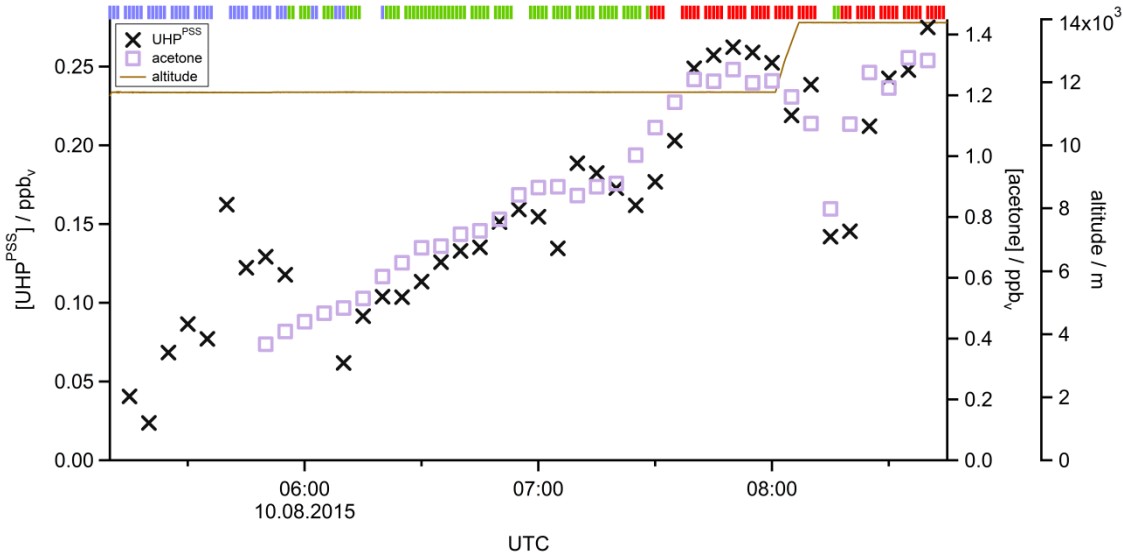

**Figure 5: Time series of UHP$^{PSS}$ (black crosses) and acetone (purple squares) mixing ratios during flight 17. The brown line shows the altitude, the colored bar on top indicates the origin of air masses according to the methane mixing ratio classification: for SH**
**blue, NH green and AMA red.**

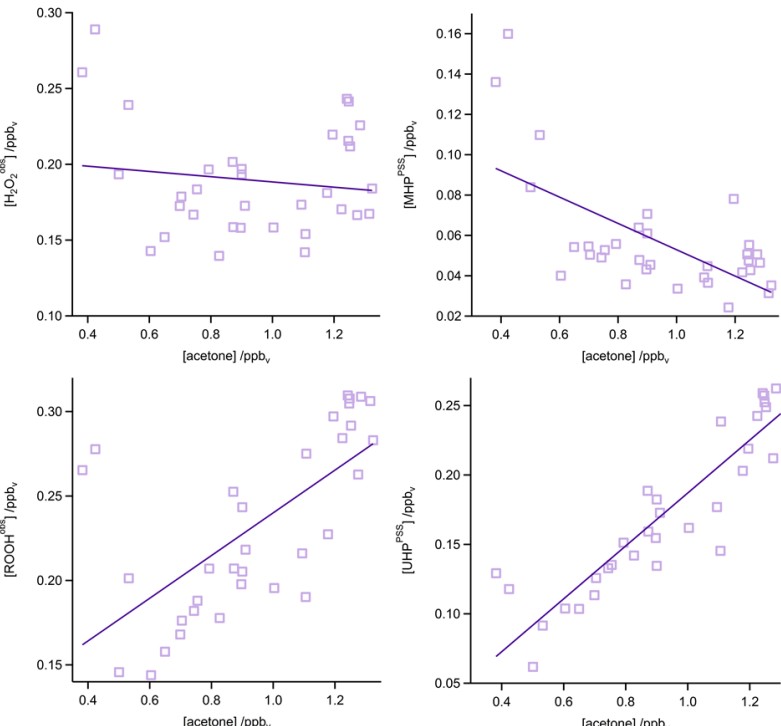

**Figure 6: Scatter plots of measured acetone and H$_2$O$_2$$^{obs}$ (top left), MHP$^{PSS}$ (top right), ROOH$^{obs}$ (bottom left) and UHP$^{PSS}$ (bottom right) during flight 17. The dark purple lines represent the least orthogonal distance fits with regression coefficients R$^2$ of 0.02, 0.41, 0.43 and 0.82.**

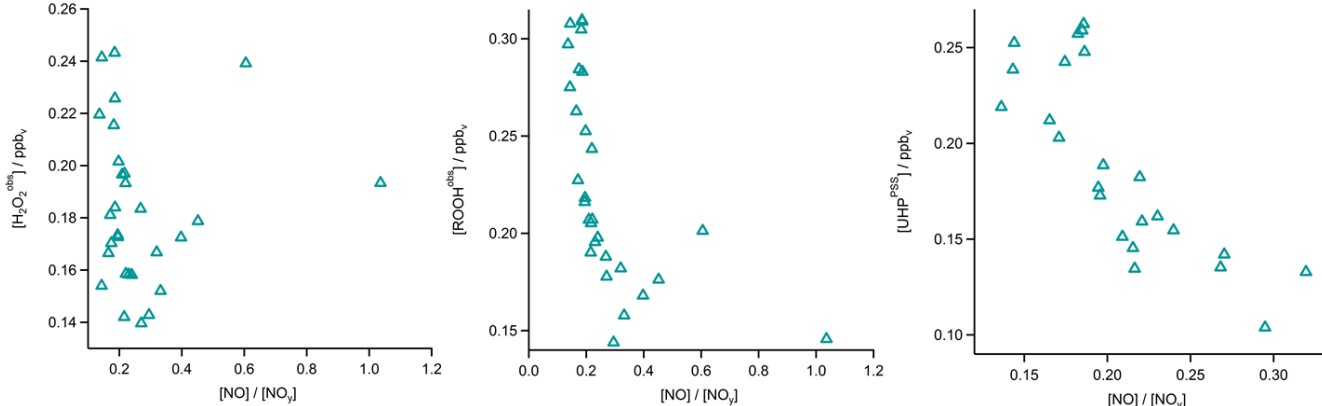


**Figure 7: Scatter plots of H$_2$O$_2$$^{obs}$ (left), ROOH$^{obs}$ (middle) and UHP$^{PSS}$ (right) and NO/NO$_y$ ratio during flight 17.**

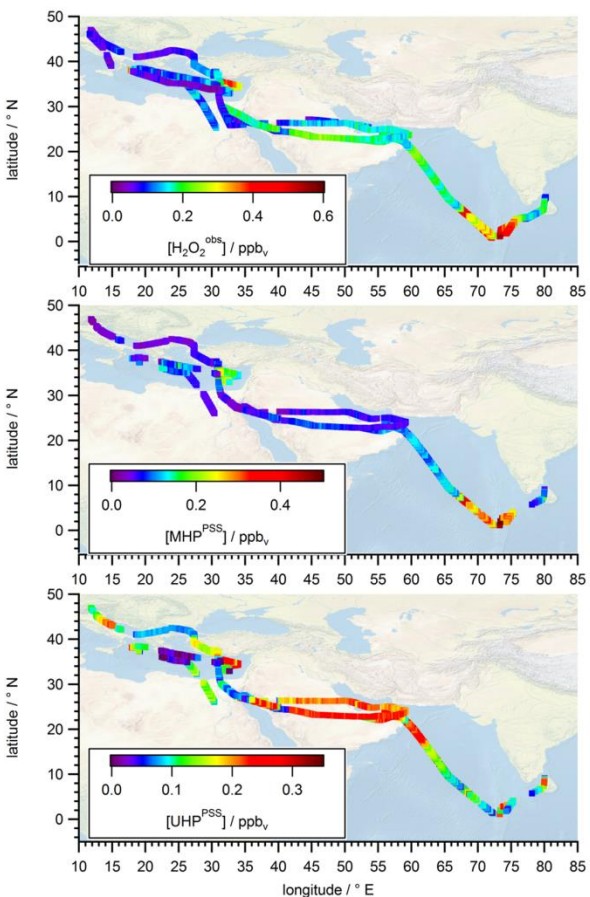

**Figure 8: All flight positions in the upper troposphere (p<300 hPa) during OMO as a function of (a) H$_2$O$_2$$^{obs}$ on top, (b) MHP$^{PSS}$ in the middle and (c) UHP$^{PSS}$ at the bottom.**

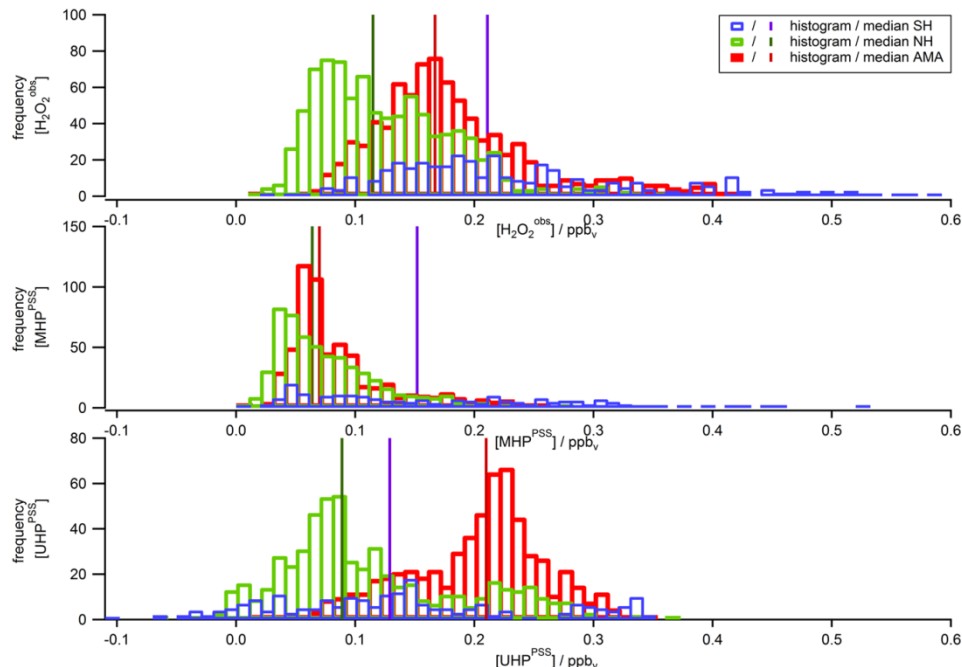


**Figure 9: Histograms of $H_2O_2^{obs}$ (top), $MHP^{PSS}$ (middle) and $UHP^{PSS}$ (bottom) mixing ratios during the OMO campaign for NH background (green), SH (blue) and AMA (red) air masses.**

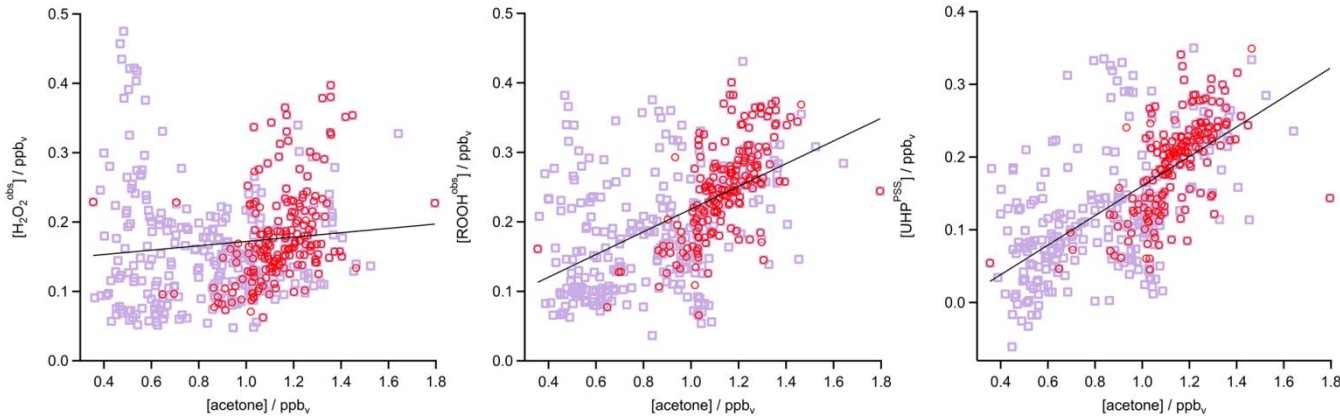


**Figure 10: Scatter plots of acetone and $H_2O_2^{obs}$ (left), $ROOH^{obs}$ (middle) and $UHP^{PSS}$ (right) in the UT (purple squares) and especially in the AMA (red circles). The black lines represent the least orthogonal distance fit with linear regression coefficients $R^2$ of 0.01 ($H_2O_2^{obs}$), 0.27 ($ROOH^{obs}$) and 0.41 ($UHP^{PSS}$).**

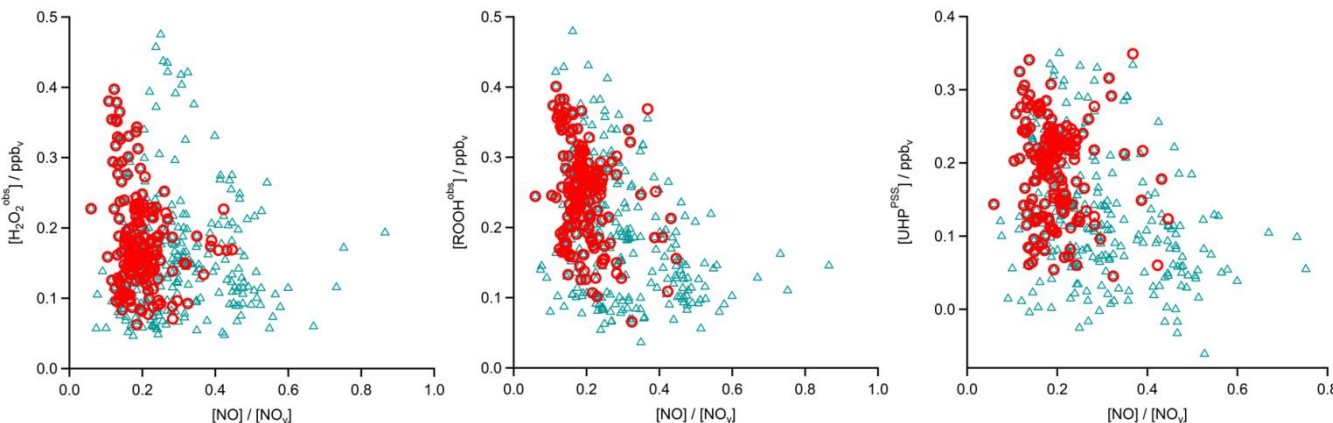

**Figure 11: Scatter plots of NO/NO$_y$ and H$_2$O$_2^{obs}$ (left), ROOH$^{obs}$ (middle) and UHP$^{PSS}$ (right) in the UT (blue triangles) and especially in the AMA (red circles).**

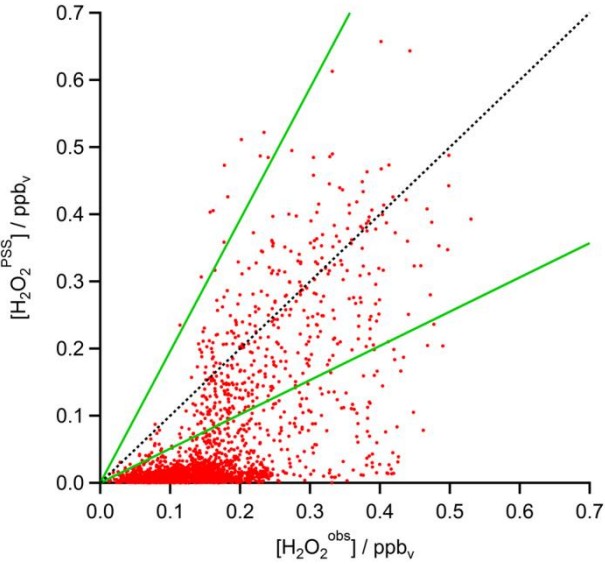


**Figure 12: Scatter plot of H$_2$O$_2^{obs}$ and H$_2$O$_2^{PSS}$ mixing ratios (red) with the 1:1 (black), 1:2 and 2:1 (both green) lines.**

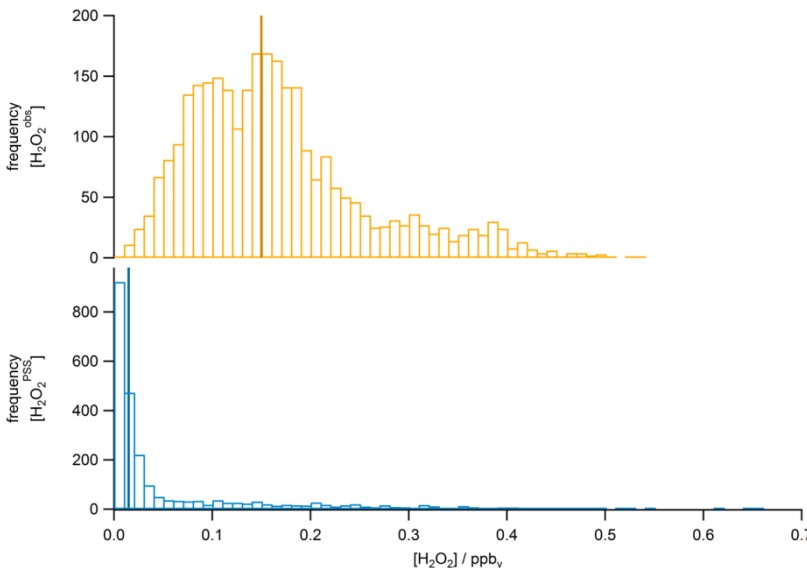

**Figure 13: Histograms of H$_2$O$_2^{obs}$ (top) and H$_2$O$_2^{PSS}$ (bottom) mixing ratios (bars) and the associated medians (lines).**

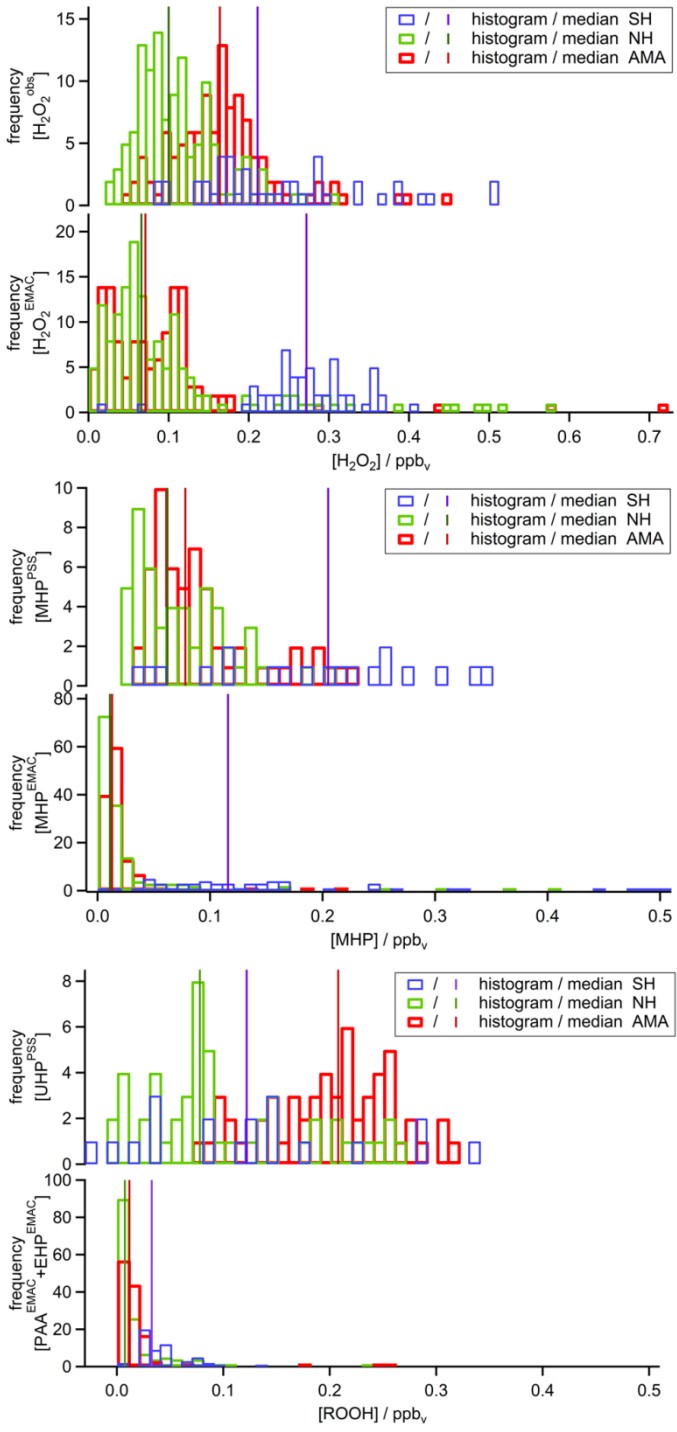

**Figure 14:** Histograms of $H_2O_2^{obs}$ and $H_2O_2^{EMAC}$ (top), $MHP^{PSS}$ and $MHP^{EMAC}$ (middle) and $UHP^{PSS}$ and $UHP^{EMAC}$ (given as $PAA^{EMAC}$ and $EHP^{EMAC}$, bottom) mixing ratios during the OMO campaign for NH background (green), SH (blue) and AMA (red) air masses.

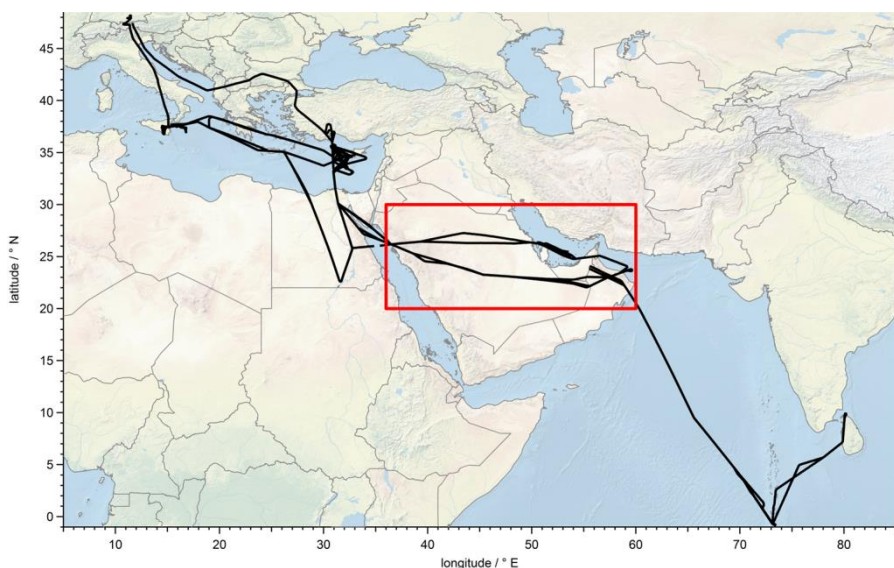

**Figure 15: Location of measurements used for the longitudinal gradient study (red box) out of all flight tracks (black).**

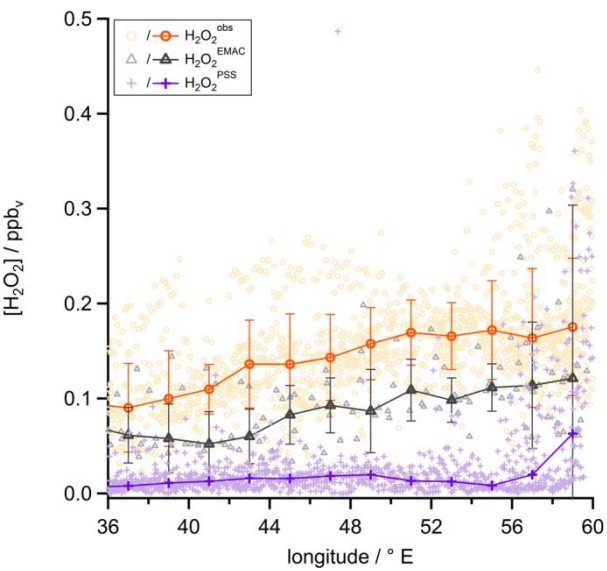


**Figure 16: Longitudinal trends of $H_2O_2^{obs}$ mixing ratios (orange circles), $H_2O_2^{EMAC}$ (black triangles) and $H_2O_2^{PSS}$ (purple crosses). The data are shown in the light colors while the darker ones represent the medians.**

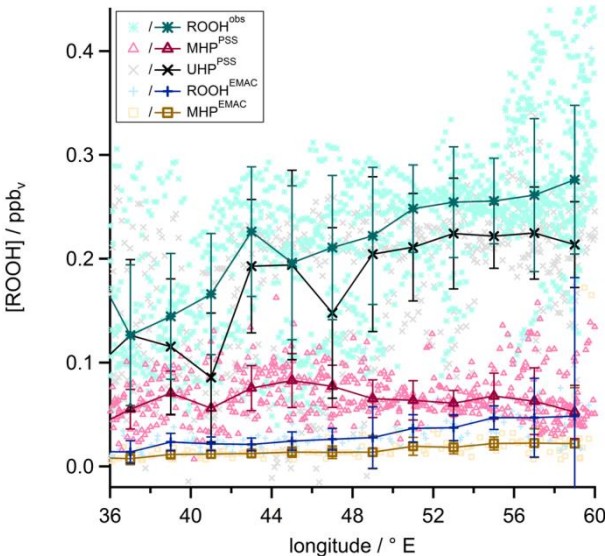

**Figure 17: Longitudinal trends of ROOH$^{obs}$ mixing ratios (green asterisks), ROOH$^{EMAC}$ (blue plus signs) and mixing ratios for MHP$^{PSS}$ (pink triangles) and UHP$^{PSS}$ (black crosses) as well as MHP$^{EMAC}$ (yellow squares). The data are shown in the light colors while the darker ones represent the medians.**

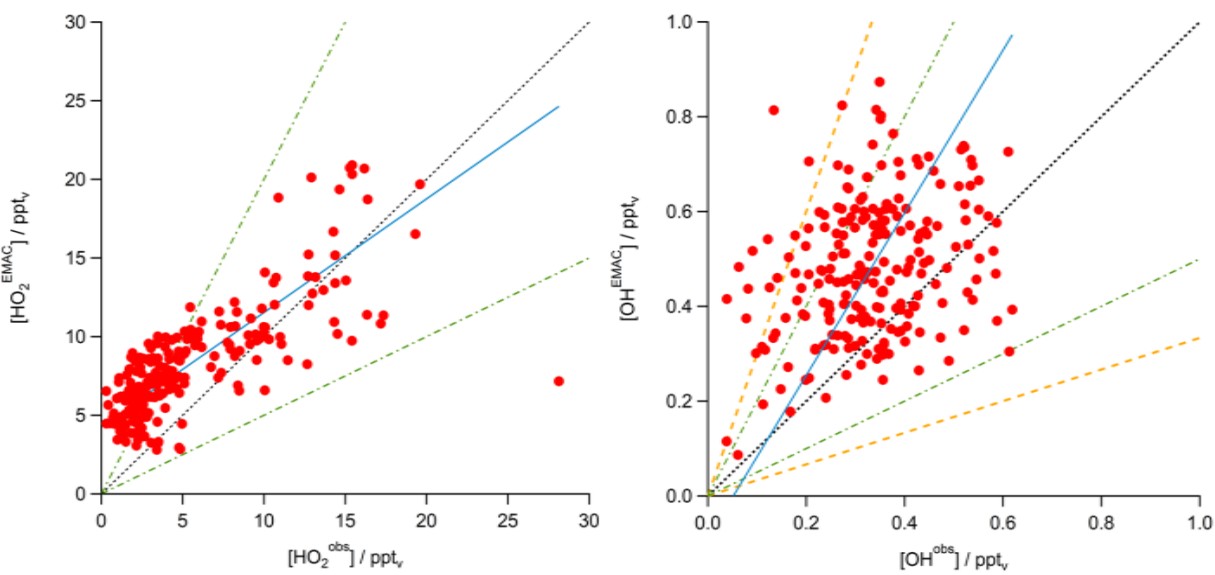

**Figure 18: Scatter plot of HO$_2$$^{obs}$ and HO$_2$$^{EMAC}$ data (left) and OH$^{obs}$ and OH$^{EMAC}$ data (right) (both red) with the 1:1 (black), 1:2 and 2:1 (both green) lines. The blue line shows the calculated least orthogonal distance fit.**

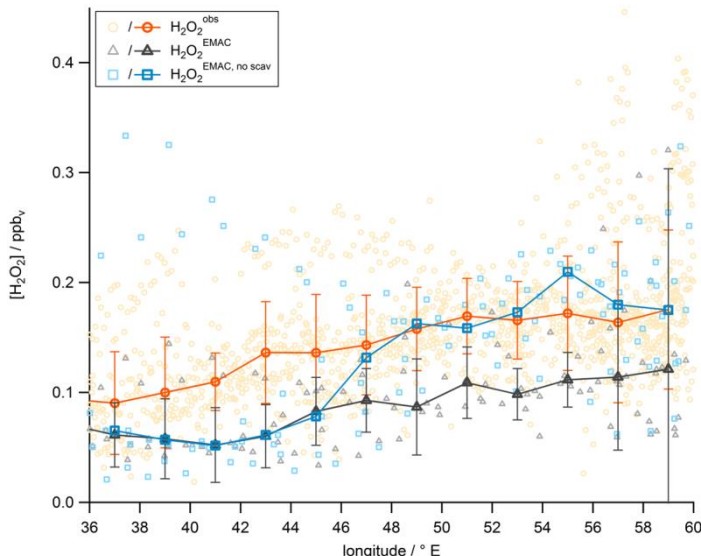

**Figure 19: Longitudinal trends of $H_2O_2{}^{obs}$ mixing ratios (orange circles), $H_2O_2{}^{EMAC}$ (black triangles) and the sensitivity study without scavenging in $H_2O_2{}^{EMAC}$ (blue circles). The data are shown in the light colors while the darker ones represent the medians.**

**Table 1: Comparison of $H_2O_2$ mixing ratios in the upper troposphere from observations and PSS calculations.**

| method | median | $[H_2O_2]/ppt_v$ |
|--------|--------|------------------|
| PSS | median | 15 |
| | range | LOD–657 |
| | avg±sdev | 61±101 |
| obs | median | 150 |
| | range | LOD–530 |
| | avg±sdev | 165±91 |


**Table 2: Comparison of H$_2$O$_2$, MHP and UHP mixing ratios in the upper troposphere from EMAC, measurements and PSS calculations.**

| region | median | [H$_2$O$_2$]/ppt$_v$ | | [MHP]/ppt$_v$ | | [UHP]/ppt$_v$ | | [ROOH] / ppt$_v$ | |
|---|---|---|---|---|---|---|---|---|---|
| | | EMAC | obs | EMAC | PSS | EMAC | PSS | EMAC | obs |
| NH | median | 66 | 100 | 11 | 64 | 8 | 78 | 18 | 135 |
| | range | 6–576 | 20–301 | 2–408 | 21–202 | 1–238 | LOD–261 | 3–458 | 18–439 |
| | avg±sdev | 102±110 | 110±53 | 28±58 | 75±42 | 18±31 | 103±77 | 46±79 | 151±82 |
| AMA | median | 71 | 164 | 13 | 70 | 12 | 208 | 23 | 244 |
| | range | 8–714 | 46–446 | 2–216 | 37–220 | 1–259 | 80–311 | 2–445 | 86–364 |
| | avg±sdev | 84±92 | 167±69 | 18±28 | 92±49 | 18±34 | 199±59 | 34±58 | 245±59 |
| SH | median | 272 | 211 | 116 | 152 | 33 | 122 | 174 | 236 |
| | range | 15–409 | 85–510 | 2–502 | 40–346 | 1–132 | LOD–334 | 2–547 | 9–446 |
| | avg±sdev | 272±68 | 238±105 | 155±125 | 191±95 | 42±24 | 125±102 | 197±129 | 232±84 |