# Peer review of "Impact of the South Asian monsoon outflow on atmospheric hydroperoxides in the upper troposphere"

_Atmospheric Chemistry and Physics, 2020_

## Referee Comment (RC1) · Anonymous Referee #1 · 28 Mar 2020

This study presents observations of hydroperoxides during an aircraft campaign investigating the outflow of the south Asian summer monsoon and how if affects the composition of the Asian Monsoon Anticyclone (AMA). The observations of H2O2 and ROOH are enhanced in the AMA, which the authors suggest is due to convective transport of these species. The authors compare these observations to steady state calculations constrained by observed OH, HO2, and photolysis frequencies as well as to the results of the EMAC global model.

I have three major concerns regarding this study, which I summarize below, followed by more minor comments.

[Figure]

Major concerns:

1) The inferred UHP is taken as the difference between measured ROOH and PSS MHP. The basis for this is not clear to me, as it assumes that MHP is accurately simulated by the PSS calculations. Given that the PSS H2O2 calculation underestimates observed H2O2 by a significant amount (up to a factor of 10!), there is no reason to believe that the PSS MHP doesn't suffer from the same problem. I found the use of UHP to be very confusing as it sometimes referred to as PSS UHP or calculated UHP or observed UHP. Given that what is measured is H2O2 and ROOH, I suggest that the authors only use these two quantities throughout the manuscript and compare them to PSS H2O2, PSS MHP, EMAC H2O2, EMAC ROOH, thus removing any use of UHP.

2) Throughout the manuscript (including the abstract and conclusions) the comparisons between observations and model results (both PSS and EMAC) are described in very vague and qualitative terms (such as "observed concentrations are higher than model calculations", "the model underestimates H2O2", etc...). This lacks rigor and leaves the reader unsure about the magnitude of the misrepresentation of the models. I strongly encourage the authors to be more quantitative in their comparisons throughout the manuscript, using statistical measures, which could include mean bias, normalized mean bias, FAC2, RMSE, etc... The figures and tables should include these statistical measures.

3) The authors suggest that deviation between EMAC and observed H2O2 and ROOH are due to uncertainties in the scavenging efficiencies of these species in the model. As described in line 430 "..sensitivity study with EMAC excluding scavenging". This sentence seems to suggest that scavenging of all species is turned off, which seems like a rather brute force method as lack of scavenging of other species could in turn affect the photochemical evolution of H2O2 and ROOH. A simulation in which scavenging of only H2O2 and ROOH is turned off seems more appropriate. Also, is scavenging turned off only for the AMA region or the entire globe? If it is for the entire globe, then the difference between the simulations might not be related to processes associated

with convection over India. It seems that the link between convection and peroxides could be investigated more carefully with EMAC, including a more targeted sensitivity simulation, and also examining correlations of modeled peroxides and NO/NOy ratio as well as acetone. Also, the authors do not discuss how well convection is represented in the model in the first place. For example how well does EMAC reproduce observations of species that are not scavenged (such as CO or some VOCs)?

Minor Comments

Sections 3.3, 3.4: Can the authors indicate the lifetimes of H2O2 and MHP during flight conditions? This will be useful to assess the validity of the PSS assumption. In particular, the validity of PSS will also depend on time of day/SZA. For what conditions do the authors apply PSS?

Line 258. The authors mention that the data were averaged in 60 second intervals. It wasn't clear from the description what the frequency of the measurements were, in particular for H2O2, ROOH and the species used to calculate PSS.

Figure 2. Can the authors indicate the number of days for which the back trajectories were calculated?

Figures 3. It is unclear why the authors show the data on the timescale of the EMAC model, given that in the text (line 263) the authors say that the model was interpolated in time and space along the flight track. Given that the observations are likely available at a higher time resolution as can be seen in Figure 4, it might make more sense to show the observations at their original time resolution instead of the much coarser 10-15 minute resolution of EMAC.

Figures 6&7 (as well as Figures 10&11) and lines 295-3050. The authors discuss the correlation of UHP with acetone and NO/NOy, without mentioning the correlation with the other peroxides. First, given that UHP is inferred based on the PSS calculation, it would make more sense to use observed ROOH. Also, nased on looking at Figure 4, it

seems that H2O2 might also be correlated with acetone and NO/NOy. Correlations (or lack thereof) with H2O2 should be discussed in the text.

Line 336 "The deviations from unity in the slope are within the combined uncertainties of measured and steady-state estimations of H2O2 (51%, 1 $\sigma$)" Looking at Figure 12, it looks like most points are outside the +/- 51% range. Can the authors be more quantitative and state the number of points outside the uncertainties?

Section 4.3.2 In the comparison to EMAC, the authors tend to focus on the range of modeled and observed values, which is not very informative. It would be more useful to discuss means or medians and provide statistical measures of the model/observations mismatch (such as mean bias, mean normalized bias, mean normalized gross error, etc...). Also to put the comparison of EMAC to peroxides in perspective, it would be useful if the authors could discuss the comparisons to other tracers (acetone, O3, H2O, CO, NOx, NOy, aerosols, etc...), which might shed light on whether the mismatch is an issue related to emissions, transport, scavenging, or chemistry.

Line 379. Many other studies before Bozem et al. (2017) have shown the role of deep convection as a source of peroxides in the upper troposphere, including Prather and Jacob (1997), Jaeglé et al. (1997), Mari et al. (2002) among others.

Lines 386-390. The authors fail to mention the very large underestimate of EMAC ROOH compared to observations.

References:

Jaeglé, L., et al.: Observed OH and HO2 in the upper troposphere suggest a major source from convective injection of peroxides, Geophys. Res. Lett., 24, 3181–3184, https://doi.org/10.1029/97GL03004, 1997.

Mari, C., et al., On the relative role of convection, chemistry, and transport over the South Pacific Convergence Zone during PEM‐Tropics B: A case study, J. Geophys. Res., 107, 8232, doi:10.1029/2001JD001466, 2002.

Prather, M. J. and Jacob, D. J.: A persistent imbalance in HOx and NOx photochemistry of the upper troposphere driven by deep convection, Geophys. Res. Lett., 24, 3189–3192, https://doi.org/10.1029/97GL03027, 1997.

---

## Referee Comment (RC2) · Anonymous Referee #2 · 29 Mar 2020

The paper describes aircraft measurements of hydroperoxide compounds and supporting observations taken during the Oxidation Mechanism Observation (OMO) mission. These measurements are analyzed alongside photochemical steady state calculations, trajectory modeling, and global model simulations to understand the source regions of the air sampled in the upper troposphere. The major findings are that hydroperoxide mixing ratios are enhanced in the Asian Monsoon Anticyclone (AMA) compared to the background Northern Hemisphere (NH) mixing ratios, but highest hydrogen peroxide (H2O2) and methyl hydrogen peroxide (CH3OOH) mixing ratios were found in the background Southern Hemisphere (SH). The authors attribute the high mixing ratios in the AMA to upwind convection, using a sensitivity simulation with the global model EMAC

to support this claim.

It is interesting to discover the higher-than-expected hydroperoxide mixing ratios in the Asian Monsoon Anticyclone, and then learn what caused these high mixing ratios. I find that the study provided hints as to the cause of the high mixing ratios, but did not provide complete attribution. The analysis would benefit from conducting box model chemistry calculations to fully understand the processes affecting hydroperoxide mixing ratios. In addition, there are a number of items that need further work as detailed below.

Specific Science Comments

1. The abstract should be more quantitative in their claim about enhancements in the AMA versus NH background. Line 22 states that observations show enhanced mixing ratios for $H_2O_2$, MHP ($CH_3OOH$), and UHP (unidentified hydroperoxides) in the AMA relative to the NH background. However, Figure 4 shows perhaps a small enhancement (10-20%) of $H_2O_2$, which is within the uncertainty (25%) of the measurements and no enhancement of MHP. There is only substantial enhancement (78%) of UHP. More convincing are the histograms in Figure 9 that show median values of $H_2O_2$ in the AMA to be 55% higher than those in the NH background, but MHP median values are quite similar between AMA and NH background. Again, the UHP median value is clearly enhanced in the AMA.

2. Introduction. Consider adding more information on the flow patterns of the Asian monsoon. A good resource for this information is Lawrence and Lelieveld ACP (2010).

3. Section 3.1 describes the hydroperoxide measurements. The method measures total peroxides which is the sum of $H_2O_2$, $CH_3OOH$, and other organic peroxides. The method uses a catalase to destroy $H_2O_2$ allowing the ability to infer $H_2O_2$ (i.e., total peroxides minus ROOH gives $H_2O_2$, where ROOH represents the sum of organic peroxides). ROOH is assumed to be mainly $CH_3OOH$. To determine $CH_3OOH$, a photostationary-state chemistry approximation is used based on measurements of OH, $HO_2$, CO, $CH_4$, NO, and photolysis rates. It is unclear why it is valid to estimate

CH3OOH from photostationary steady-state when the chemical lifetimes of CH3OOH and H2O2 are a few days (as stated on line 381, page 13).

It would be better to describe the measurement technique as measuring total peroxides, inferring H2O2, and estimating MHP_PSS with the remaining ROOH being called unidentified hydroperoxides (UHP). Then, when the authors suggest that most of the UHP is CH3OOH, then they can use MHP without further notation.

4. Lines 295-305. The correlation between UHP and acetone is very strong, but there's no explanation on what the cause and effect may be. I suggest further analysis on this result. Even stating acetone photolysis produces methyl peroxy radical which can react with HO2 to form CH3OOH is good, but more interesting would be box model calculations.

5. Section 4.3. I think assuming photostationary steady-state for H2O2 and CH3OOH interferes with the comparisons described in Section 4.3.

a. While, it is useful to point out the large discrepancy between observed H2O2 and H2O2 estimated by photostationary steady-state, it needs an explanation of why there is such a discrepancy (Lines 333-340).

b. In Section 4.3.2, I do not find it useful to compare the ranges of the observations and EMAC results. For example, EMAC clearly underpredicts CH3OOH mixing ratios most of the time, but the range of EMAC results overlaps with the observations. It may be better to discuss medians or simply describe that most EMAC CH3OOH is < 50 pptv for NH background and AMA air, while most observations range from level of detection to 120 pptv.

c. The photostationary steady-state assumption interferes with the comparisons of the MHP and UHP observations with EMAC. If H2O2 and CH3OOH are not in photostationary steady-state, then it would make more sense to compare total organic peroxides between observations and EMAC results. The differences between model results and

observations need an explanation of why they are different.

d. Shouldn't a conclusion be that assuming photostationary steady-state can be inappropriate?

6. Lines 379-385. This discussion, to me, is not well supported and contains a lot of suppositions. This manuscript is relying on one paper (Bozem et al., 2017) to say $H_2O_2$ is enhanced in convective outflow regions compared to the background upper troposphere. Yet Bozem et al. (2017) found an unusually high $H_2O_2$ mixing ratio in convective outflow (1.25 ppbv) which is not found in other studies. Snow et al. (2007) and Barth et al. (2016) both show that $H_2O_2$ is depleted in convective outflow compared to background upper troposphere. We do not know what $H_2O_2$ mixing ratios are like in convective outflow near the convection that transports constituents into the AMA, but we can make use of the array of literature from past studies to guide us for what to expect and what further analysis is needed. This leads to my next comment.

7. There will always be some variability in peroxides scavenging efficiencies and uncertainty in these scavenging efficiencies due to the complex processes associated with convection and chemistry. However, there were also 100s km (multiple days) of transit between the convection in northern India and the measurements over the Arabian Sea. What chemistry occurred during this transit? If it is true that the observations are reflecting chemistry in convective outflow, then one would also expect other volatile organic compounds, CO, and $CH_4$ to have been lofted in the convection. I would recommend conducting a number of box model calculations (e.g. Pickering et al., JGR, 1992; Apel et al., ACP, 2012, etc) to learn what chemical transformations are affecting the peroxides. Further, this box model can more definitively provide information on what unidentified hydroperoxides are.

8. Lines 405-409. I do not think the typhoon Mireille case is suitable to compare to this paper's results. Typhoon Mireille occurred over the Western Pacific ingesting air from Oceania (Preston et al., JGR, 2019) in the early 1990s and not the South Asia region

of the mid 2010s.

9. I am surprised there is no mention of past literature on peroxides, peroxy radicals and convection and how these current results compare to those previous findings. Some papers to discuss are Jaeglé et al., GRL, 1997, Prather and Jacob, GRL, 1997, Crawford et al., JGR, 1999).

Organization, Clarity, Technical Comments

1. Line 140 should include a list of all species measured. It should state OH and HO2 instead of HOx, and include NO.

2. Lines 192-195. Shouldn't pressure and temperature measurements also be listed? These state parameters must be needed for calculating rate constants and air density.

3. Line 209. It would be good to explain why the CH3O2 to HO2 ratio is needed. It would also be good to define P(HO2) and P(CH3O2).

4. Line 217. Why is a scale height needed? Why not use pressure measurements from the aircraft along with temperature to get air density that can then be translated to N2?

5. Line 221. Isn't there a H2O dependence to the self-reaction of HO2? See, for example, JPL (2015). https://jpldataeval.jpl.nasa.gov/

6. Line 229. Can you show or quantify the contribution of CH4+OH and CO+OH to the total CH3O2 and HO2 production, respectively? If the hypothesis is that the higher AMA mixing ratios are due to convective transport, then HCHO, CH3OOH, and other VOCs will be elevated compared to background mixing ratios and their chemistry may be more important than assumed here.

7. Lines 245-248. What grid spacing is used for EMAC? Perhaps a short summary of the configuration could be given in supplemental material. Further, how do the authors analyzed EMAC output to provide comparisons to aircraft observations? Are the model

results interpolated in time and space to the aircraft location? Or are values from the nearest grid point and model output time (which, I assume, is every hour)?

8. Lines 249-255. What is the procedure when the back-trajectory encounters convection? Does FLEXPART have a means to represent convective transport? Or are the trajectories stop when convection is encountered?

9. Line 253-254. Since methane and its use for identifying AMA air via a threshold value is discussed on these lines, it would be useful to combine the first paragraph of section 4.1 with this information. Or move lines 252-255 to section 4.1.

10. Line 262. It is not clear which species concentrations are binned into 10 pptv segments. Please specify which species.

11. Line 265. "Case study: flight 17" is not very descriptive to the general audience. Consider using a heading that mentions the date and location of the flight.

12. Line 266. There should also be a short description of the flight, again mentioning date, but perhaps adding weather conditions (cloudy anywhere?) and location of the anticyclone, etc. You might want to add this description to section 2.

13. Line 272. –> Tomsche et al. (2019) and on line 273, remove (Tomsche et al. 2019).

14. Line 274. State what EMAC time step is. I imagine this is the frequency of model output.

15. Line 279-280. Clarify that it is EMAC model data.

16. Line 282. –> last period of the flight at the higher altitude,

17. Line 293. I think it would be better to say "temporal pattern" rather than "evolution" as there is no following an air parcel in time in the figure.

18. Line 318. –> is found. Air masses . . . .

19. Line 333. –> photostationary steady-state

20. Line372. It should be degrees E (not O), and please mention the red box in the figure.

21. Line 373. The observations are in orange (not yellow).

Figures and Table

1. Consider putting some figures into one with panels.

2. Figure 2. Please add number of days for back trajectories to caption. It would be helpful to mark each hour (text box of time) along the flight track so one can connect the map to the time series.

3. Figure 3 and others. It would help to say "EMAC modelled" for clarity. And "data-constrained calculated" should be "photostationary steady state calculated". At least be consistent from figure to figure and figure to text with nomenclature.

4. Figure 4. Purple triangles look red in my version.

5. Figures 5, 6 and 7 could be combined. The lilac colored squares should be darker.

6. Figure 8. Please note in the figure caption that you are showing data only at <300 hPa.

7. Figures 10, 11, and 12 could be combined. The lilac colored squares should be darker.

8. Figure 12 figure caption should say "photostationary steady state calculated".

9. Figure 14 axes labels and legends need to be larger. Please explain what vertical lines are in the caption.

10. Figure 15. Please explain what the red box is in the caption.

11. Figure 16. The photostationary steady state calculated markers are purple not blue.

12. Figure 17. The MHP mixing ratios look more like pink than purple.

13. Table 1. Are these values from all flights? Please say so in the caption. "calc." is not a good heading. I suggest PSS estimate.
* * *

---

## Author Comment (AC1) · 20 May 2020

**Anonymous Referee #1.**

(black: RC, red: AC, blue: changed in manuscript)

This study presents observations of hydroperoxides during an aircraft campaign investigating the outflow of the south Asian summer monsoon and how it affects the composition of the Asian Monsoon Anticyclone (AMA). The observations of H2O2 and ROOH are enhanced in the AMA, which the authors suggest is due to convective transport of these species. The authors compare these observations to steady state calculations constrained by observed OH, HO2, and photolysis frequencies as well as to the results of the EMAC global model. I have three major concerns regarding this study, which I summarize below, followed by more minor comments.

Major concerns:

*1) The inferred UHP is taken as the difference between measured ROOH and PSS MHP. The basis for this is not clear to me, as it assumes that MHP is accurately simulated by the PSS calculations. Given that the PSS H2O2 calculation underestimates observed H2O2 by a significant amount (up to a factor of 10!), there is no reason to believe that the PSS MHP doesn't suffer from the same problem. I found the use of UHP to be very confusing as it sometimes referred to as PSS UHP or calculated UHP or observed UHP. Given that what is measured is H2O2 and ROOH, I suggest that the authors only use these two quantities throughout the manuscript and compare them to PSS H2O2, PSS MHP, EMAC H2O2, EMAC ROOH, thus removing any use of UHP.*

We are sorry for the confusion. As mentioned in the experimental section, the ROOH measurement is unspecific and due to the different solubilities of hydroperoxides only qualitative. In order to estimate the amount of MHP, which according to previous measurements is expected to be the dominant (if not the only) ROOH component, we calculated the amounts of MHP from a photo-stationary state calculation (as well as H2O2). The difference between ROOH and PSS MHP is unexplained or unaccounted for, thus we named it UHP. As discussed in the paper UHP can be due to an unidentified hydroperoxide (e.g. PAA), additional MHP due to advection or a combination of both. Since no specific ROOH measurements were made nor does the 3D-model indicate substantial amount of hydroperoxides other than MHP, we cannot finally decide on the nature (or composition) of the UHP. In the revised manuscript, we will more clearly define UHP and follow the referee in using PSS H2O2, PSS MHP, EMAC H2O2, EMAC ROOH in addition to EMAC MHP. Nevertheless we would still like to use UHP in the sense of an unaccounted for hydroperoxide either as additional MHP (exceeding PSS MHP) and/or a significant contribution by an organic hydroperoxide not simulated in EMAC (e.g. PAA).

Section 4.3.2 changed to:

Figure 14 shows histograms for the comparison between H2O2, MHP and UHP observations with EMAC simulations. Hydrogen peroxide simulations from EMAC cover a broader range of mixing ratios for both NH background (6 pptv up to 576 pptv) and AMA (8–714 pptv) compared to observations (NH background: 20–301 pptv; AMA 46–446 pptv). For the SH model simulations and observations indicate almost identical ranges of 15–409 pptv and 85–510 pptv, respectively. Median EMAC-H2O2 values are similar for NH background (66 pptv) and AMA (71 pptv) conditions (Table 2), while observations indicate an enhancement of +64 pptv in the AMA relative to the NH background. For the SH the model simulated H2O2 mixing ratios are four times higher than in the NH background (272 pptv), while the observations only show a median increase by 47 pptv to 211 pptv (Table 2).

EMAC mainly simulates MHP mixing ratios lower than 50 pptv for background and AMA, while PSS-MHP ranges from LOD–140 pptv. Again the model simulates highest MHP mixing ratios in the SH with values up to 502 pptv compared to up to 346 pptv in the PSS-MHP calculations. Similar as for H2O2, medians of EMAC-MHP for NH background and monsoon conditions are almost identical (11 pptv and 13 pptv respectively, Table 2) in the model simulations, while the observations show a small difference towards higher mixing ratios in the AMA (64 pptv and 70 pptv, respectively). In the simulations, southern hemispheric EMAC-MHP mixing ratios are almost ten times higher than NH background values, compared to two to three times higher ones in the observations.

Data for UHP in the model are calculated from the sum of simulated ethyl hydroperoxide (EHP) and peroxyacetic acid (PAA), which are the only non-methyl organic hydroperoxides in the free troposphere according to the model. EMAC-UHP mixing ratios range from 1–238 pptv in the NH background, 1–259 pptv in the AMA and 1–132 pptv in the SH. PSS-UHP based on the observations indicate lowest mixing ratios in the NH background (LOD–261 pptv), while in the AMA and the SH the ranges are quite similar (80–311 pptv and LOD–334 pptv). A comparison of median values emphasizes the large difference between model simulations and observation-based estimates. In the NH background, the median PSS-UHP mixing ratio from the observations is 70 pptv higher than EMAC simulations (78 pptv and 8 pptv respectively). In the AMA the difference is even larger, with about 200 pptv higher PSS-UHP levels compared to the EMAC simulations. The smallest difference with a factor of four was found for the SH (Table 2).

*2) Throughout the manuscript (including the abstract and conclusions) the comparisons between observations and model results (both PSS and EMAC) are described in very vague and qualitative terms (such as "observed concentrations are higher than model calculations", "the model underestimates H2O2", etc. . .). This lacks rigor and leaves the reader unsure about the magnitude of the misrepresentation of the models. I strongly encourage the authors to be more quantitative in their comparisons throughout the manuscript, using statistical measures, which could include mean bias, normalized mean bias, FAC2, RMSE, etc. . . The figures and tables should include these statistical measures.*

We added the ranges, averages and standard deviations for the in-situ and PSS comparison (Table 1). In addition we added more statistical information in table 2 (former table 1). The abstract as well as conclusion were changed so that they are more quantitative.

Abstract changed to:

L22: We observed enhanced mixing ratios of H2O2 (45%), MHP (9%) and UHP (136%) in the AMA relative to the northern hemispheric background. Highest concentrations for H2O2 and MHP of 211 ppbv and 152 ppbv, respectively were found in the tropics outside the AMA, while for UHP, with 208 pptv, highest concentrations were found within the AMA. In general, the observed concentrations are higher than steady-state calculations and EMAC simulations. Especially in the AMA, EMAC underestimates the H2O2 (medians: 71 pptv vs. 164 pptv) and ROOH (medians: 25 pptv vs. 278 pptv) mixing ratios.

Conclusion changed to:

L471: The atmospheric chemistry-general circulation model EMAC slightly underestimates H2O2 in the NH background (medians: 66 pptv vs. 100 pptv), but significantly underestimates it in the AMA (medians: 71 pptv vs. 164 pptv), and overestimates it in the SH (medians: 272 pptv vs. 211 pptv). Steady-state calculations for H2O2 and MHP based on observed precursors yield much lower values, in particular in the AMA, resulting in a large contribution of an unidentified organic hydroperoxide (UHP) in air masses affected by the Asian summer monsoon.

New table added:

Table 1: Comparison of H2O2 mixing ratios in the upper troposphere from measurements and PSS calculations.

| Data set | statistical measures | $[H_2O_2]$/$ppt_v$ |
|---|---|---|
| PSS | median
range
avg±sdev | 15
LOD–657
61±101 |
| HYPHOP | median
range
avg±sdev | 150
LOD–530
165±91 |

And thus former Table 1 Is now Table 2.

Section 4.3.1 starting with line 357 changed to:

The regression coefficient is quite high (0.83), even though most of the calculated steady-state mixing ratios (75%) are in the range between 0 and 65 pptv with a median value of 15 pptv, while the measured mixing ratios extend over a larger range mainly between 10–210 pptv with a median of 150 pptv and thus 10 times higher than for steady-state, which can also be seen in the histograms in Figure 13. Table 1 shows the statistical comparison of both data sets.

Table 2 changed to:

| region | median | $[H_2O_2]$/$ppt_v$ | | $[MHP]$/$ppt_v$ | | $[UHP]$/$ppt_v$ | |
|---|---|---|---|---|---|---|---|
| | | EMAC | HYPHOP | EMAC | PSS | EMAC | PSS |

| NH background | median | 66 | 100 | 11 | 64 | 8 | 78 |
| | range | 6–576 | 20–301 | 2–408 | 21–202 | 1–238 | LOD–261 |
| | avg±sdev | 102±110 | 110±53 | 28±58 | 75±42 | 18±31 | 103±77 |
| monsoon | median | 71 | 164 | 13 | 70 | 12 | 208 |
| | range | 8–714 | 46–446 | 2–216 | 37–220 | 1–259 | 80–311 |
| | avg±sdev | 84±92 | 167±69 | 18±28 | 92±49 | 18±34 | 199±59 |
| SH background | median | 272 | 211 | 116 | 152 | 33 | 122 |
| | range | 15–409 | 85–510 | 2–502 | 40–346 | 1–132 | LOD–334 |
| | avg±sdev | 272±68 | 238±105 | 155±125 | 191±95 | 42±24 | 125±102 |

Title of Fig. 13 changed to: "Figure 13: Histograms of measured (top) and calculated (bottom) H2O2 mixing ratios (bars) and the associated medians (lines)."

*3) The authors suggest that deviation between EMAC and observed H2O2 and ROOH are due to uncertainties in the scavenging efficiencies of these species in the model. As described in line 430 "..sensitivity study with EMAC excluding scavenging". This sentence seems to suggest that scavenging of all species is turned off, which seems like a rather brute force method as lack of scavenging of other species could in turn affect the photochemical evolution of H2O2 and ROOH. A simulation in which scavenging of only H2O2 and ROOH is turned off seems more appropriate. Also, is scavenging turned off only for the AMA region or the entire globe? If it is for the entire globe, then the difference between the simulations might not be related to processes associated with convection over India. It seems that the link between convection and peroxides could be investigated more carefully with EMAC, including a more targeted sensitivity simulation, and also examining correlations of modeled peroxides and NO/NOy ratio as well as acetone. Also, the authors do not discuss how well convection is represented in the model in the first place. For example how well does EMAC reproduce observations of species that are not scavenged (such as CO or some VOCs)?*

- CO in the upper troposphere is lower in EMAC due to weaker convective transport (possibly caused by the low resolution of the global model) as described in Tost et al. (2016) and Tomsche et al. (2019).

Tost, H., Jöckel, P., and Lelieveld, J.: Influence of different convection parameterisations in a GCM, Atmos. Chem. Phys., 6, 5475–5493, https://doi.org/10.5194/acp-6-5475-2006, 2006

-As described in Klippel et al., scavenging of all soluble species was turned off in the model, not only for the Indian sub-continent. Since the purpose of the sensitivity study is to calculate the maximum amount of H2O2 that can be transported into the AMA, we believe that this is justified. Simultaneous wash-out of other soluble species (HNO3, organic acids etc) will not directly affect H2O2 but might change HOx levels, thus affect H2O2 through secondary chemistry. Also convective addition of H2O2 in other regions (e.g. the West African Monsoon) will hardly affect H2O2 levels in the AMA due to the limited lifetime of H2O2.

*Minor Comments*

*Sections 3.3, 3.4: Can the authors indicate the lifetimes of H2O2 and MHP during flight conditions? This will be useful to assess the validity of the PSS assumption. In particular, the validity of PSS will also depend on time of day/SZA. For what conditions do the authors apply PSS?*

The lifetime of $H_2O_2$ is around 4 days and MHP around 1 day. Major losses are the reaction with OH and photolysis by sunlight. No special conditions were applied to the calculations as the aim was to estimate local photochemistry. Measurements were only performed during daytime.

*Line 258. The authors mention that the data were averaged in 60 second intervals. It wasn't clear from the description what the frequency of the measurements were, in particular for H2O2, ROOH and the species used to calculate PSS.*

Data are from merged data sets which were calculated from the original data that were recorded with a higher resolution. For $H_2O_2$ and ROOH we record 1 value per second, TRISTAR ($CH_4$ and CO) appr. 1 per second, OH and $HO_2$ appr. 4 per minute, NO and $NO_y$ 1 per second, J values 1 value per 1 or 2 seconds.

Changed to:

L272: Data were collected from a merged data set given as 60-second-means (calculated from the original data set obtained at higher resolutions) in order to get the same time resolution for all compounds.

*Figure 2. Can the authors indicate the number of days for which the back trajectories were calculated?*

Ten day back trajectories were used.

The title of the figure is changed to:

Figure 2: Track of flight 17 (black dotted line) and calculated 10-day-back trajectories (lines colored as a function of altitude) to show the origin of sampled air masses during the flight.

*Figures 3. It is unclear why the authors show the data on the timescale of the EMAC model, given that in the text (line 263) the authors say that the model was interpolated in time and space along the flight track. Given that the observations are likely available at a higher time resolution as can be seen in Figure 4, it might make more sense to show the observations at their original time resolution instead of the much coarser 10-15 minute resolution of EMAC.*

EMAC results are obtained at a temporal resolution of 12 min and spatial resolution of 2.8° x 2.8°, and interpolated in space along the flight track. Therefore, any interpolation to higher time resolution of this dataset would not provide any added value.

Changed to:

L253: For this study EMAC simulations were performed for the OMO flight tracks in 2.8°x2.8° grids with a time resolution of 12 minutes.

*Figures 6&7 (as well as Figures 10&11) and lines 295-3050. The authors discuss the correlation of UHP with acetone and NO/NOy, without mentioning the correlation with the other peroxides. First, given that*

*UHP is inferred based on the PSS calculation, it would make more sense to use observed ROOH. Also, nased on looking at Figure 4, it seems that H2O2 might also be correlated with acetone and NO/NOy. Correlations (or lack thereof) with H2O2 should be discussed in the text.*

Following the suggestion, we have provided the requested relations. Please note that a positive correlation not necessarily indicates a relation to a precursor. It can also indicate a co-location of sources.

Section changed to:

L311: The PSS-UHP and acetone mixing ratios in this part of the flight are strongly correlated (Figure 6), with a slope of 0.19±0.02 (ppbv/ppbv) and an offset of (-0.003±0.02) ppbv. The regression coefficient R2 is very high (0.99). For H2O2 and ROOH the correlation is not that strong with slopes of -0.02±0.02 (ppbv/ppbv) and 0.13±0.03 (ppbv/ppbv), respectively, and offsets of (0.21±0.02) ppbv and (0.11±0.03) ppbv (Figure 6). The relation between ROOH mixing ratios and an air mass age tracer based on the ratio between [NO] to [NOy] shows higher values of ROOH at smaller ratios representing older or more processed air masses (Figure 7), since highest ROOH mixing ratios (>200 pptv) are found at the lowest [NO]/[NOy] ratios (all <0.19). Thus, most of the observed ROOH was measured in aged air masses transported within the anticyclone. The correlation with PSS-UHP shows that this effect is mainly due to PSS-UHP. For H2O2 there are also some higher mixing ratios for high [NO] to [NOy] mixing ratios and thus fresher air (Figure 7).

Section changed to:

L344: In the analysis of flight 17 we found a strong correlation between PSS-UHP and acetone (Figure 6) and an increase of PSS-UHP at the oldest air mass ages, represented by low [NO]/[NOy] (Figure 7). Extension of this analysis to all observations in the upper troposphere obtained during OMO yields similar results for the relation between PSS-UHP, in situ ROOH and in situ H2O2 and acetone (Figure 10). Enhanced mixing ratios of hydroperoxides are typically associated with enhanced acetone mixing ratios, especially for PSS-UHP. A simple calculation of the production of MHP out of the photolysis of acetone and the reaction of acetaldehyde (from EMAC) with OH shows that per day appr. 40 pptv MHP can be formed within the AMA. The lifetime of MHP was calculated to be around 1.5 days. Thus not all of the PSS-UHP in the AMA (median 210 pptv) can be accounted for by MHP that was chemically produced from VOCs in the AMA. The scatter plots of the hydroperoxides vs. [NO]/[NOy] for the whole data set, show no clear correlation with a large spread of hydroperoxides mixing ratios at the lowest [NO]/[NOy] ratios, representing the oldest, i.e. chemically most processed air masses (Figure 11).

Figures 6 and 7 changed to:

[Figure]

**Figure 6: Scatter plots of measured acetone and in situ H2O2 (left), in situ ROOH (middle) and PSS-UHP (right) during flight 17. The black lines represent the least orthogonal distance fits with regression coefficients of 0.99, 0.98 and 0.99.**

[Figure]

**Figure 7: Scatter plots of in situ H2O2 (left), in situ ROOH (middle) and PSS-UHP (right) and NO/NOy ratio during flight 17.**

Figures 10 and 11 changed to:

[Figure]

**Figure 10: Scatter plots of in situ acetone and in situ H2O2 (left), in situ ROOH (middle) and PSS-UHP (right) in the UT (purple squares) and especially in the AMA (red circles). The black lines represent the least orthogonal distance fit with linear regression coefficients of 0.96 (H2O2), 0.97 (ROOH) and 0.96 (UHP).**

[Figure]

**Figure 11: Scatter plots of NO/NOy and in situ H2O2 (left), in situ ROOH (middle) and PSS-UHP (right) in the UT (blue triangles) and especially in the AMA (red circles).**

*Line 336 "The deviations from unity in the slope are within the combined uncertainties of measured and steady-state estimations of H2O2 (51%, 1 σ)" Looking at Figure 12, it looks like most points are outside the +/- 51% range. Can the authors be more quantitative and state the number of points outside the uncertainties?*

82% of points are outside the range of the uncertainties of ± 51%.

[Figure]

*Section 4.3.2 In the comparison to EMAC, the authors tend to focus on the range of modeled and observed values, which is not very informative. It would be more useful to discuss means or medians and provide statistical measures of the model/observations mismatch (such as mean bias, mean normalized bias, mean normalized gross error, etc. . .). Also to put the comparison of EMAC to peroxides in perspective, it would be useful if the authors could discuss the comparisons to other tracers (acetone, O3, H2O, CO, NOx, NOy, aerosols, etc. . .), which might shed light on whether the mismatch is an issue related to emissions, transport, scavenging, or chemistry.*

Table 2 was changed (see above). In addition we want to refer to Lelieveld et al., 2018 where a detailed comparison of observed and modelled data is shown.

*Line 379. Many other studies before Bozem et al. (2017) have shown the role of deep convection as a source of peroxides in the upper troposphere, including Prather and Jacob (1997), Jaeglé et al. (1997), Mari et al. (2002) among others.*

Line 404 (former 379) was changed to

In contrast, other studies found that deep convection can be a source of H2O2 in the upper troposphere (e.g. Jaeglé et al., 1997; Prather and Jacob, 1997; Mari et al., 2003; Bozem et al., 2017).

*Lines 386-390. The authors fail to mention the very large underestimate of EMAC ROOH compared to observations.*

Here we focus on the qualitative analysis of the longitudinal gradients and thus on the trends along the longitude. Including the additional information, the difference between in situ data and the EMAC model is discussed quite elaborately now.

*References:*

*Jaeglé, L., et al.: Observed OH and HO2 in the upper troposphere suggest a major source from convective injection of peroxides, Geophys. Res. Lett., 24, 3181–3184, https://doi.org/10.1029/97GL03004, 1997. Mari, C., et al., On the relative role of convection, chemistry, and transport over the South Pacific Convergence Zone during PEMâAˇ RTropics B: A case study, J. Geophys. ˇ Res., 107, 8232, doi:10.1029/2001JD001466, 2002. C4 ACPD Interactive comment Printer-friendly version Discussion paper Prather, M. J. and Jacob, D. J.: A persistent imbalance in HOx and NOx photochemistry of the upper troposphere driven by deep convection, Geophys. Res. Lett., 24, 3189– 3192, https://doi.org/10.1029/97GL03027, 1997. Interactive comment on Atmos. Chem. Phys. Discuss., https://doi.org/10.5194/acp-2020-93, 2020.*

---

## Author Comment (AC2) · 20 May 2020

**Please note the colour code**
**(black: RC, red: AC, blue: changes in manuscript)**

The paper describes aircraft measurements of hydroperoxide compounds and supporting observations taken during the Oxidation Mechanism Observation (OMO) mission. These measurements are analyzed alongside photochemical steady state calculations, trajectory modeling, and global model simulations to understand the source regions of the air sampled in the upper troposphere. The major findings are that hydroperoxide mixing ratios are enhanced in the Asian Monsoon Anticyclone (AMA) compared to the background Northern Hemisphere (NH) mixing ratios, but highest hydrogen peroxide ($H_2O_2$) and methyl hydrogen peroxide ($CH_3OOH$) mixing ratios were found in the background Southern Hemisphere (SH). The authors attribute the high mixing ratios in the AMA to upwind convection, using a sensitivity simulation with the global model EMAC to support this claim.

It is interesting to discover the higher-than-expected hydroperoxide mixing ratios in the Asian Monsoon Anticyclone, and then learn what caused these high mixing ratios. I find that the study provided hints as to the cause of the high mixing ratios, but did not provide complete attribution. The analysis would benefit from conducting box model chemistry calculations to fully understand the processes affecting hydroperoxide mixing ratios. In addition, there are a number of items that need further work as detailed below.

*Specific Science Comments*
*1. The abstract should be more quantitative in their claim about enhancements in the AMA versus NH background. Line 22 states that observations show enhanced mixing ratios for $H_2O_2$, MHP ($CH_3OOH$), and UHP (unidentified hydroperoxides) in the AMA relative to the NH background. However, Figure 4 shows perhaps a small enhancement (10-20%) of $H_2O_2$, which is within the uncertainty (25%) of the measurements and no enhancement of MHP. There is only substantial enhancement (78%) of UHP. More convincing are the histograms in Figure 9 that show median values of $H_2O_2$ in the AMA to be 55% higher than those in the NH background, but MHP median values are quite similar between AMA and NH background. Again, the UHP median value is clearly enhanced in the AMA.*

Abstract changed to:
Line 22: We observed enhanced mixing ratios of $H_2O_2$ (45%), MHP (9%) and UHP (136%) in the AMA relative to the northern hemispheric background. Highest concentrations for $H_2O_2$ and MHP of 211 ppbv and 152 ppbv, respectively were found in the tropics outside the AMA, while for UHP, with 208 pptv highest concentrations were found within the AMA. In general, the observed concentrations are higher than steady-state calculations and EMAC simulations. Especially in the AMA, EMAC underestimates the $H_2O_2$ (medians: 71 pptv vs. 164 pptv) and ROOH (medians: 25 pptv vs. 278 pptv) mixing ratios.

*2. Introduction. Consider adding more information on the flow patterns of the Asian monsoon. A good resource for this information is Lawrence and Lelieveld ACP (2010).*

Introduction changed to:
Line 66: So far we know that the updrafts of the summer monsoon deep convection can effectively transport insoluble pollutants from the surface to the upper troposphere and there these polluted air masses can be transported over a long distance (Lawrence and Lelieveld, 2010). Thus the Asian summer monsoon has a strong influence on the upper troposphere (UT)

and the lower stratosphere (Randel et al., 2010; Gettelman et al., 2004) and it is important to study its physical and chemical properties in greater detail.

*3. Section 3.1 describes the hydroperoxide measurements. The method measures total peroxides which is the sum of H2O2, CH3OOH, and other organic peroxides. The method uses a catalase to destroy H2O2 allowing the ability to infer H2O2 (i.e. total peroxides minus ROOH gives H2O2, where ROOH represents the sum of organic peroxides). ROOH is assumed to be mainly CH3OOH. To determine CH3OOH, a photostationary-state chemistry approximation is used based on measurements of OH, HO2, CO, CH4, NO, and photolysis rates. It is unclear why it is valid to estimate CH3OOH from photostationary steady-state when the chemical lifetimes of CH3OOH and H2O2 are a few days (as stated on line 381, page 13). It would be better to describe the measurement technique as measuring total peroxides, inferring H2O2, and estimating MHP_PSS with the remaining ROOH being called unidentified hydroperoxides (UHP). Then, when the authors suggest that most of the UHP is CH3OOH, then they can use MHP without further notation.*

As mentioned in the experimental section, the ROOH measurement is unspecific and due to the different solubilities of hydroperoxides qualitative. In order to estimate the amount of MHP, which according to previous measurements is expected to be the dominant (if not the only) ROOH component, we calculated the amount of MHP from a photo-stationary state calculation (as well as H2O2). The difference between ROOH and PSS MHP is unexplained or unaccounted for, thus we named it UHP. As discussed in the paper UHP can be due to an unidentified hydroperoxide (e.g. PAA), additional MHP due to advection or a combination of both. Since no specific ROOH measurements were made nor does the 3D-model indicate substantial amount of hydroperoxides other than MHP, we cannot finally decide on the nature (or composition) of the UHP.

*4. Lines 295-305. The correlation between UHP and acetone is very strong, but there's no explanation on what the cause and effect may be. I suggest further analysis on this result. Even stating acetone photolysis produces methyl peroxy radical which can react with HO2 to form CH3OOH is good, but more interesting would be box model calculations.*

We calculated the production of MHP from acetone and acetaldehyde. For the AMA appr. 40 ppt$_v$ MHP per day can be formed through this reaction.

Section changed to.
L347: Enhanced mixing ratios of hydroperoxides are typically associated with enhanced acetone mixing ratios, especially for PSS-UHP. Our calculation of the production of MHP from the photolysis of acetone and the reaction of acetaldehyde (from EMAC) with OH shows that per day appr. 40 pptv MHP can be formed within the AMA. The lifetime of MHP was calculated to be around 1.5 days. Thus not all of the PSS-UHP in the AMA (median 210 pptv) can be accounted for MHP that was chemically produced from VOCs in the AMA.

*5. Section 4.3. I think assuming photostationary steady-state for H2O2 and CH3OOH interferes with the comparisons described in Section 4.3.*
*a. While, it is useful to point out the large discrepancy between observed H2O2 and H2O2 estimated by photostationary steady-state, it needs an explanation of why there is such a discrepancy (Lines 333-340).*

The discrepancy is mainly due to transport phenomena especially deep convection over India.

We added to Section 4.3.1.:
L363: The discrepancy between in situ and PSS-H2O2 shows that the local PSS does not account all main contributions of H2O2 even though all chemical reactions are included. Thus transport phenomena like deep convection seem to play a key role (see 4.3.3).

*b. In Section 4.3.2, I do not find it useful to compare the ranges of the observations and EMAC results. For example, EMAC clearly underpredicts CH3OOH mixing ratios most of the time, but the range of EMAC results overlaps with the observations. It may be better to discuss medians or simply describe that most EMAC CH3OOH is < 50 pptv for NH background and AMA air, while most observations range from level of detection to 120 pptv.*

Section changed to:
Line 375: EMAC mainly simulates MHP mixing ratios lower than 50 pptv for background and AMA, while PSS-MHP ranges from LOD–140 pptv.

*c. The photostationary steady-state assumption interferes with the comparisons of the MHP and UHP observations with EMAC. If H2O2 and CH3OOH are not in photostationary steady-state, then it would make more sense to compare total organic peroxides between observations and EMAC results. The differences between model results and observations need an explanation of why they are different.*

In sum we compare total observed peroxides with EMAC-the mixing ratios are just splitted into PSS-MHP from local CH4 oxidation and other organic hydroperoxides. If this is MHP from other chemical sources (e.g. from acetone photolysis or reaction of acetaldehyde and OH), transported MHP or if it is another hydroperoxide like PAA cannot be verified.
Reasons for the differences between the model and observations are given in Lines 432–440:
"Although there is rather good agreement between EMAC simulations and observations for all the species that affect the local photochemical budget of H2O2, EMAC significantly exceeds PSS calculation for H2O2. This is an indication that an additional H2O2 source is accounted for in the global model and that the local photo-stationary-state assumption is not fulfilled. The additional source is attributed to transport associated with deep convection over India, yielding in an upwind source of H2O2 that is significant throughout the western part of the AMA. In the AMA, clouds are absent, so that gas phase photochemical processes may determine the lifetime of H2O2. Based on observed OH levels and photolysis frequencies during OMO the H2O2 lifetime in the upper troposphere is of the order of several days, sufficiently long for the excess H2O2 to reach the western parts of the AMA, producing the observed longitudinal H2O2 gradient observed in both observations and EMAC simulations (Figure 16)."

Sections changed to:
Line 454: Differences between H2O2 observations and EMAC simulations are most likely due to an overestimation of scavenging in the model as also pointed out by Klippel et al., 2011). To investigate this assumption we performed a sensitivity study with EMAC excluding scavenging. The result is shown in Figure 19. The H2O2 mixing ratios significantly increase with longitude by a factor of 3–4 and thus to the level of observed H2O2.

Line 458: There is a rather large uncertainty regarding the scavenging efficiency of MHP in deep convection (Barth et al., 2016). For the Trace A campaign Mari et al. (2000) found observed (modelled) enhancement ratios of post-convective to pre-convective mixing ratios of 11 (9.5) for MHP and 1.9 (1.2) for H2O2. Such efficient transport in the Indian Summer Monsoon would yield a strong source of upper tropospheric MHP explaining the large enhancement of ROOH in the AMA described here. It seems that a large part of the PSS-UHP is actually MHP advected

throughout the AMA after deep convective transport over India. In the EMAC simulations the transport of MHP is less efficient and thus EMAC-MHP is lower than PSS-MHP and PSS-UHP.
*d. Shouldn't a conclusion be that assuming photostationary steady-state can be inappropriate?*

Yes we would like to stress this in the conclusions. Local photochemistry does not explain the high mixing ratios that we found. Thus transport must play a substantial role especially since other chemical formation of MHP is insufficient. We assume that deep convection is the reason for the higher concentrations in the upper troposphere.

*6. Lines 379-385. This discussion, to me, is not well supported and contains a lot of suppositions. This manuscript is relying on one paper (Bozem et al., 2017) to say H2O2 is enhanced in convective outflow regions compared to the background upper troposphere. Yet Bozem et al. (2017) found an unusually high H2O2 mixing ratio in convective outflow (1.25 ppbv) which is not found in other studies. Snow et al. (2007) and Barth et al. (2016) both show that H2O2 is depleted in convective outflow compared to background upper troposphere. We do not know what H2O2 mixing ratios are like in convective outflow near the convection that transports constituents into the AMA, but we can make use of the array of literature from past studies to guide us for what to expect and what further analysis is needed. This leads to my next comment.*

We now show the inconsistent results from both sides.

Section changed to:
L403: Previous studies present results that are difficult to reconcile. Snow et al. (2007) and Barth et al. (2016) for example both show that H2O2 is depleted in convective outflow compared to background upper troposphere. In contrast, other studies found that deep convection can be a source of H2O2 in the upper troposphere (e.g. Jaeglé et al., 1997; Prather and Jacob, 1997; Mari et al., 2003; Bozem et al., 2017).

*7. There will always be some variability in peroxides scavenging efficiencies and uncertainty in these scavenging efficiencies due to the complex processes associated with convection and chemistry. However, there were also 100s km (multiple days) of transit between the convection in northern India and the measurements over the Arabian Sea. What chemistry occurred during this transit? If it is true that the observations are reflecting chemistry in convective outflow then one would also expect other volatile organic compounds, CO, and CH4 to have been lofted in the convection. I would recommend conducting a number of box model calculations (e.g. Pickering et al., JGR, 1992; Apel et al., ACP, 2012, etc) to learn what chemical transformations are affecting the peroxides. Further, this box model can more definitively provide information on what unidentified hydroperoxides are.*

We do not see any chance to get more information on the organic hydroperoxides since PAN and acetaldehyde were not measured during the campaign. Calculations based on EMAC acetaldehyde and observed acetone are added (see above).

*8. Lines 405-409. I do not think the typhoon Mireille case is suitable to compare to this paper's results. Typhoon Mireille occurred over the Western Pacific ingesting air from Oceania (Preston et al., JGR, 2019) in the early 1990s and not the South Asia region of the mid 2010s.*

We do not want to exactly compare the Typhoon study with our study. We just want to mention that such phenomena are already known from previous studies.

*9. I am surprised there is no mention of past literature on peroxides, peroxy radicals and convection and how these current results compare to those previous findings. Some papers to*

*discuss are Jaeglé et al., GRL, 1997, Prather and Jacob, GRL, 1997, Crawford et al., JGR, 1999).*

Some of these studies are now mentioned (see 6.).

*Organization, Clarity, Technical Comments*
*1. Line 140 should include a list of all species measured. It should state OH and HO2 instead of HOx, and include NO.*

Section changed to:
L144: For this study CO, CH4, OH, HO2, O3, Acetone, NO, NOy, JH2O2 and JMHP data measured by other instruments have been used for data interpretation, steady-state calculations and interference corrections (see section 3.1). A complete list of all measured compounds can be found in Lelieveld et al., 2018.

*2. Lines 192-195. Shouldn't pressure and temperature measurements also be listed? These state parameters must be needed for calculating rate constants and air density.*

Section changed to:
L196: Latitude, longitude and altitude data as well as temperature and pressure were collected with the BAHAMAS (BAsic HALO Measurement And Sensor system) instrument. More detailed information about the installation of scientific instruments and mission flights can be found on http://www.halo.dlr.de/science/missions/omo/omo.html.

*3. Line 209. It would be good to explain why the CH3O2 to HO2 ratio is needed. It would also be good to define P(HO2) and P(CH3O2).*

P(CH3O2) and P(HO2) are defined in eq. 12. As it is analogous to P(H2O2) and P(MHP) we thought this is enough explanation. If not-we can change it.

*4. Line 217. Why is a scale height needed? Why not use pressure measurements from the aircraft along with temperature to get air density that can then be translated to N2?*

Section changed to:

L220: For the calculations of the rate coefficients the mean temperature of 259.18 K, the mean altitude of 10,992.8 m and the mean pressure of 22,932.9 Pa were used.

*5. Line 221. Isn't there a H2O dependence to the self-reaction of HO2? See, for example, JPL (2015). https://jpldataeval.jpl.nasa.gov/*

Yes there is a [$H_2O$] dependence. But in this case it was neglected because of the low RH in the upper troposphere. An example: For 100 ppm $H_2O$ a factor of {1+1.4x10$^{-21}$*[H2O]*exp(2200/T)} and thus 1.004 needs to be included to the calculation. (http://iupac.pole-ether.fr/htdocs/datasheets/pdf/HOx14_2HO2_(M).pdf)

Section changed to:
L222: As the relative humidity is very low in the upper troposphere the water dependence in eq. 11 was neglected.

*6. Line 229. Can you show or quantify the contribution of CH4+OH and CO+OH to the total CH3O2 and HO2 production, respectively? If the hypothesis is that the higher AMA mixing ratios*

*are due to convective transport, then HCHO, CH3OOH, and other VOCs will be elevated compared to background mixing ratios and their chemistry may be more important than assumed here.*

Section changed to:
Line 234: This is justified by the generally low mixing ratios of these species at high altitudes. Measurements of HCHO with the TRISTAR instrument yielded values below the detection limit of 30 pptv, and although acetaldehyde was not measured, we assume that its mixing ratio is within a factor of two of those of HCHO.

*7. Lines 245-248. What grid spacing is used for EMAC? Perhaps a short summary of the configuration could be given in supplemental material. Further, how do the authors analyzed EMAC output to provide comparisons to aircraft observations? Are the model results interpolated in time and space to the aircraft location? Or are values from the nearest grid point and model output time (which, I assume, is every hour)?*

The EMAC simulation was made for the flight track of the aircraft. EMAC data were compared to measurements and calculations for the same time and thus location in 2.8°x2.8° grids of the aircraft. Therefore the corresponding values out of the 1 minute means were used. EMAC offers one value every 12 minutes.

Line 253: For this study EMAC simulations were performed for the OMO flight tracks in 2.8°x2.8° grids with a time resolution of 12 minutes. Detailed specifications and results have been published previously (Lelieveld et al., 2018; Tomsche et al., 2019).

Line 276: To compare the simulations from EMAC with measured and PSS calculated data, the corresponding values (out of the 60-second-means) were used at the given times from EMAC.

*8. Lines 249-255. What is the procedure when the back-trajectory encounters convection? Does FLEXPART have a means to represent convective transport? Or are the trajectories stop when convection is encountered?*

Convective transport can be simulated in FLEXPART with the convection parameterization by Emanuel and Zivkovic-Rothman (1999). To represent moist convection realistically in models, the parametrization includes cloud microphysical processes, the physics of entrainment and mixing, as well as large scale control of ensemble convective activity. It builds on temperature and humidity fields to provide mass flux information (Stohl et al., 2005). The back trajectories in the present paper are calculated with the convective parametrization. Further the Lagrangian particle dispersion model FLEXPART produces so called centroid trajectories, which found on cluster analysis. These trajectories are comparable to traditional trajectories, but include convection via the centroid of all particles per time step.

Section changed to:
L256: Ten-day back-trajectories were calculated along the flight path using FLEXPART to identify the air mass origin (Tomsche et al., 2019). Convective transport can be simulated in FLEXPART with the convection parameterization by Emanuel and Zivkovic-Rothman (1999). To represent moist convection realistically in models, the parametrization includes cloud microphysical processes, the physics of entrainment and mixing, as well as large scale control of ensemble convective activity. It builds on temperature and humidity fields to provide mass flux information (Stohl et al., 2005). The back trajectories in the present paper are calculated with the convective parametrization. Further, the Lagrangian particle dispersion model FLEXPART produces so called centroid trajectories, which found on cluster analysis. These trajectories are

comparable to traditional trajectories, but include convection via the centroid of all particles per time step.

References:

Emanuel, K. A. and Zivkovic-Rothman, M. (1999). Development and Evaluation of a Convection Scheme for Use in Climate Models. Journal of the Atmospheric Sciences, 56(11):17661782.

Stohl, A., Forster, C., Frank, A., Seibert, P., and Wotawa, G.: Technical note: The Lagrangian particle dispersion model FLEXPART version 6.2, Atmos. Chem. Phys., 5, 2461–2474, https://doi.org/10.5194/acp-5-2461-2005, 2005.

*9. Line 253-254. Since methane and its use for identifying AMA air via a threshold value is discussed on these lines, it would be useful to combine the first paragraph of section 4.1 with this information. Or move lines 252-255 to section 4.1.*

Sentences shifted to 3.5

Line 267: Thus a threshold of CH4≥1879.8 ppbv was used to distinguish between air masses influenced by the monsoon (CH4≥1879.8 ppbv), the SH background (CH4<1820 ppbv) and the NH background (1820 ppbv≤CH4<1879.8 ppbv) (Tomsche et al., 2019).

*10. Line 262. It is not clear which species concentrations are binned into 10 pptv segments. Please specify which species.*

Section changed to:
L275: For the histograms the concentrations of all species shown were binned into samples with a width of 10 pptv, starting the plots with the lowest bin.

*11. Line 265. "Case study: flight 17" is not very descriptive to the general audience. Consider using a heading that mentions the date and location of the flight.*

Header changed to:
L278: 4.2 Case study: Flight 17 from Gan to Bahrain (10.08.2015)

*12. Line 266. There should also be a short description of the flight, again mentioning date, but perhaps adding weather conditions (cloudy anywhere?) and location of the anticyclone, etc. You might want to add this description to section 2.*

Yes there were some clouds but as the focus is on the upper troposphere and the convection took place before and somewhere else it was not mentioned.

Section changed to:
L279: In a case study analyzing flight 17 from 10th of august 2015, the method used to determine the origin of the measured air masses and a quantification and comparison of measured and simulated mixing ratios of H2O2, PSS-MHP and PSS-UHP is presented.

L286: Figure 3 shows the time series for measured H2O2 during the flight at the time steps given from the frequency of EMAC output (orange circles).

*13. Line 272. –> Tomsche et al. (2019) and on line 273, remove (Tomsche et al. 2019).*

Section changed to:
L285: Tomsche et al. 2019 showed that the measured air in the AMA was affected by deep convection over India resulting in methane mixing ratios above the threshold.

*14. Line 274. State what EMAC time step is. I imagine this is the frequency of model output.*

Section changed to:
L286: Figure 3 shows the time series for measured H2O2 during the flight at the time steps given from the frequency of EMAC output (orange circles).

*15. Line 279-280. Clarify that it is EMAC model data.*

Section changed to:
L292: One hour later the EMAC model data decrease to 416 pptv and the in situ data increase to 214 pptv.

*16. Line 282. –> last period of the flight at the higher altitude,*

Section changed to:
L293: During the following hour until around 8:00 UTC and thus at higher altitude, both mixing ratios increase with the modelled data showing a much stronger increase up to approximately 800 pptv while the in situ data increase only to 230 pptv.

*17. Line 293. I think it would be better to say "temporal pattern" rather than "evolution" as there is no following an air parcel in time in the figure.*

*Section changed to:*
*L306:* The *in situ* $H_2O_2$ mixing ratios show a similar temporal pattern and mixing ratio levels to those of PSS-UHP over the Arabian Sea and the Arabian Peninsula, with values in the range of 140–243 ppt$_v$.

*18. Line 318. –> is found. Air masses*

Section changed to:
L334: For PSS-MHP (Figure 9, middle panel) the frequency distribution in the NH background shows a maximum at 30–40 pptv (green). For AMA influenced air a sharp maximum at 50–70 pptv (red) is found. Air masses from the SH exhibit a rather flat distribution with a maximum at values of 40–50 pptv and a median of 152 pptv (blue).

*19. Line 333. –> photostationary steady-state*
Section changed to:
L355: A scatter plot of the results from the H2O2 photostationary steady-state calculation based on observed HOx data in the UT (eq. 15) is shown in Figure 12.

*20. Line372. It should be degrees E (not O), and please mention the red box in the figure.*

Section changed to:

L395: In Figure 16 observations, steady-state calculations and EMAC simulations for upper tropospheric (9–15 km) H2O2 are displayed as a function of longitude from west to east (20–30 °N, 36–60 °E, according to the red box in Figure 15).

*21. Line 373. The observations are in orange (not yellow).*

Section changed to:
L297: The observations (orange) show roughly a 100% increase of in situ H2O2 from west to east (90 pptv to 175 pptv), similar to simulation with EMAC (black), although absolute mixing ratio levels in EMAC-H2O2 are smaller (61 pptv to 121 pptv).

*Figures and Table*
*1. Consider putting some figures into one with panels.*

As the other referee asked for more figures we want to decide this at the end.

*2. Figure 2. Please add number of days for back trajectories to caption. It would be helpful to mark each hour (text box of time) along the flight track so one can connect the map to the time series.*

Figure changed to:

[Figure]

[Figure]

*3. Figure 3 and others. It would help to say "EMAC modelled" for clarity. And "dataconstrained calculated" should be "photostationary steady state calculated". At least be consistent from figure to figure and figure to text with nomenclature.*

Changed to:
Figure 3: Time series of measured (orange circles), PSS calculated (blue crosses) and modelled (grey triangles) H2O2 mixing ratios for flight 17. The brown line shows the altitude, the colored bar on top indicates the origin of air masses according to the methane mixing ratio classification: for SH blue, NH green and monsoon red.

Figure 4: Time series of hydroperoxide mixing ratios during flight 17. The mixing ratios of in situ H2O2 (orange circles), PSS-MHP (purple triangles) and PSS-UHP (black crosses) are shown.

The brown line shows the altitude, the colored bar on top indicates the origin of air masses according to the methane mixing ratio classification: for SH blue, NH green and monsoon red.

Figure 5: Time series of PSS-UHP (black crosses) and in situ acetone (green circles) mixing ratios during flight 17. The brown line shows the altitude, the colored bar on top indicates the origin of air masses according to the methane mixing ratio classification: for SH blue, NH green and monsoon red.

Figure 6: Scatter plots of measured acetone and in situ $H_2O_2$ (left), in situ ROOH (middle) and PSS-UHP (right) during flight 17. The black lines represent the least orthogonal distance fits with regression coefficients of 0.99, 0.98 and 0.99.

Figure 7: Scatter plots of in situ $H_2O_2$ (left), in situ ROOH (middle) and PSS-UHP (right) and NO/NOy ratio during flight 17.

Figure 8: All flight positions in the upper troposphere (p<300 hPa) during OMO as a function of (a) in situ $H_2O_2$ on top, (b) PSS-MHP in the middle and (c) PSS-UHP at the bottom.

Figure 9: Histograms of in situ $H_2O_2$ (top), PSS-MHP (middle) and PSS-UHP (bottom) mixing ratios during the OMO campaign for NH background (green), SH (blue) and monsoon (red) air masses.

Figure 10: Scatter plots of in situ acetone and in situ $H_2O_2$ (left), in situ ROOH (middle) and PSS-UHP (right) in the UT (purple squares) and especially in the AMA (red circles). The black lines represent the least orthogonal distance fit with linear regression coefficients of 0.96 ($H_2O_2$), 0.97 (ROOH) and 0.96 (UHP).

Figure 11: Scatter plots of NO/NOy and in situ $H_2O_2$ (left), in situ ROOH (middle) and PSS-UHP (right) in the UT (blue triangles) and especially in the AMA (red circles).

Figure 12: Scatter plot of in situ and PSS calculated $H_2O_2$ mixing ratios (red) with the 1:1 (black), 1:2 and 2:1 (both green) lines.

Figure 13: Histograms of in situ (top) and PSS (bottom) $H_2O_2$ mixing ratios (bars) and the associated medians (lines).

Figure 14: Histograms of in situ and EMAC $H_2O_2$ (top), PSS and EMAC-MHP (middle) and PSS and EMAC-UHP (bottom) mixing ratios during the OMO campaign for NH background (green), SH (blue) and AMA (red) air masses.

Figure 16: Longitudinal trends of in situ $H_2O_2$ mixing ratios (orange circles), EMAC-$H_2O_2$ (black triangles) and PSS $H_2O_2$ (purple crosses). The data are shown in the light colors while the darker ones represent the medians.

Figure 17: Longitudinal trends of in situ ROOH mixing ratios (green asterisks), EMAC-ROOH (blue plus signs) and PSS mixing ratios for MHP (pink triangles) and UHP (black crosses) as well as EMAC-MHP (yellow squares). The data are shown in the light colors while the darker ones represent the medians.

Figure 18: Scatter plot of in situ and EMAC HO2 data (left) and the OH data (right) (both red) with the 1:1 (black), 1:2 and 2:1 (both green) lines. The blue line shows the calculated least orthogonal distance fit.

Figure 19: Longitudinal trends of in situ H2O2 mixing ratios (orange circles), EMAC (black triangles) and the sensitivity study without scavenging in EMAC (blue circles). The data are shown in the light colors while the darker ones represent the medians.

*4. Figure 4. Purple triangles look red in my version.*

Might be due to the printer used.

*5. Figures 5, 6 and 7 could be combined. The lilac colored squares should be darker.*

As figures 6 and 7 are now made of 3 scatter plots we do not combine these figures.

*6. Figure 8. Please note in the figure caption that you are showing data only at <300 hPa.*

Caption changed to:
Figure 8: All flight positions in the upper troposphere (p<300 hPa) during OMO as a function of (a) in situ H2O2 on top, (b) PSS-MHP in the middle and (c) PSS-UHP at the bottom.

*7. Figures 10, 11, and 12 could be combined. The lilac colored squares should be darker.*

As figures 10 and 11 are now made of 3 scatter plots we do not combine these figures.

*8. Figure 12 figure caption should say "photostationary steady state calculated".*

Caption changed to:
Figure 12: Scatter plot of in situ and PSS calculated H2O2 mixing ratios (red) with the 1:1 (black), 1:2 and 2:1 (both green) lines.

*9. Figure 14 axes labels and legends need to be larger. Please explain what vertical lines are in the caption.*

Vertical lines are explained in the legend-these are medians. Font size now increased.

[Figure]

**Figure 1: Histograms of observed and modelled H$_2$O$_2$ (top), MHP (middle) and UHP (bottom) mixing ratios during the OMO campaign for NH background (green), SH (blue) and AMA (red) air masses.**

10. Figure 15. Please explain what the red box is in the caption.

Caption changed to:

Figure 15: Location of measurements used for the longitudinal gradient study (red box) out of all flight tracks (black).

*11. Figure 16. The photostationary steady state calculated markers are purple not blue.*

Caption changed to:
Figure 16: Longitudinal trends of in situ H2O2 mixing ratios (orange circles), EMAC-H2O2 (black triangles) and PSS H2O2 (purple crosses). The data are shown in the light colors while the darker ones represent the medians.

*12. Figure 17. The MHP mixing ratios look more like pink than purple.*

Caption changed to:
Figure 17: Longitudinal trends of in situ ROOH mixing ratios (green asterisks), EMAC-ROOH (blue plus signs) and PSS mixing ratios for MHP (pink triangles) and UHP (black crosses) as well as EMAC-MHP (yellow squares). The data are shown in the light colors while the darker ones represent the medians.

*13. Table 1. Are these values from all flights? Please say so in the caption. "calc." is not a good heading. I suggest PSS estimate.*

Table changed to:

**Table 2: Comparison of $H_2O_2$, MHP and UHP mixing ratios in the upper troposphere from EMAC, measurements and PSS calculations.**

| region | median | $[H_2O_2]/ppt_v$ | | $[MHP]/ppt_v$ | | $[UHP]/ppt_v$ | |
|---|---|---|---|---|---|---|---|
| | | EMAC | HYPHOP | EMAC | PSS | EMAC | PSS |
| NH background | median | 66 | 100 | 11 | 64 | 8 | 78 |
| | range | 6–576 | 20–301 | 2–408 | 21–202 | 1–238 | LOD–261 |
| | avg±sdev | 102±110 | 110±53 | 28±58 | 75±42 | 18±31 | 103±77 |
| monsoon | median | 71 | 164 | 13 | 70 | 12 | 208 |
| | range | 8–714 | 46–446 | 2–216 | 37–220 | 1–259 | 80–311 |
| | avg±sdev | 84±92 | 167±69 | 18±28 | 92±49 | 18±34 | 199±59 |
| SH background | median | 272 | 211 | 116 | 152 | 33 | 122 |
| | range | 15–409 | 85–510 | 2–502 | 40–346 | 1–132 | LOD–334 |
| | avg±sdev | 272±68 | 238±105 | 155±125 | 191±95 | 42±24 | 125±102 |

---

## Author Comment (AC3) · 20 May 2020

Please note that in the revised draft Greta Stratmann from German Aerospace Center, Institute of Atmospheric Physics, Oberpfaffenhofen, 82234, Germany was added to the Co-Author list. She together with HZ was responsible for the NO and NOy measurements.

---

## Referee Report (RR1)

I thank the reviewers for considering and incorporating my suggested changes. I have a few minor comments that can be addressed as technical corrections.

1. I am glad to see that the authors have modified the text to denote H2O2 and MHP calculated from the photostationary steady-state method as PSS-H2O2 and PSS-MHP. I see that in the Results section. It would also be good to use that nomenclature in section 3.3, namely with equations 13 and 14 as well as in the text:
  The total uncertainty of MHP from the calculation → The total uncertainty of PSS-MHP
  the calculated concentration of MHP was subtracted → the PSS-MHP was subtracted

2. To be more explicit about my comment on CH3O2 and HO2 ratios and production, I suggest the following. The line "Similarly the CH3O2 to HO2 ratio can be deduced from Eq. 6." just before equation 6 should say something like, "Because individual peroxy radicals were not measured, the CH3O2 to HO2 ratio must be estimated from their production and loss terms. This ratio can be deduced as written in equation 6."
I agree with the authors that P(CH3O2) and P(HO2) do not need to be written explicitly when they are in equation 12.

3. I was surprised to find out that HCHO had values below the detection limit of 30 pptv. In the AMA I would expect much higher mixing ratios because of the convective lofting. Perhaps the added sentences in section 3.3 refer to the NH background values. If so, I suggest saying, "HCHO in the *background NH* was below the detection limit of 30 pptv,"

4. My suggestion of putting some figures together is no longer needed now that the scatter plots are multi-paneled based on the other reviewer's suggestion of additional plots.

Figure 3 caption should be revised with the update to the flight track colors.

Figure 5: I still see the triangles as more red than purple, and I'm looking at the computer screen. Since it is not exactly the same red as the monsoon category at the top, perhaps it is a dark red or purple-red combination. Below I pasted the figure and then added a text box with the colors MSWord thinks are purple, red, and dark red.

[Figure]

Purple
Red
Dark red
Lilac

Figure 6. Here the acetone mixing ratios are shown as lilac squares and not green circles as stated in the caption.

Table 1. region → method

Table 2. It still would be good to clarify if the values reported are from all flights or a subset or flights.

---

## Referee Report (RR2)

The authors have addressed some of my concerns, but I still have significant remaining concerns and some additional issues.

1) Lack of consistent notation for what is calculated, inferred, and observed in the peroxides. The authors go back and forth between various notations in the text and figures. PSS-H2O2, H2O2 PSS, [H2O2]$_{PSS}$. Similarly UHP is referred to as UHP or PSS-UHP, and MHP is referred to as MHP or PSS-MHP. Observed H2O2 is referred to as H2O2 HYPHOP or [H2O2]$_{HYPHOP}$, or in situ H2O2 or H2O2. EMAP H2O2 is referred to as H2O2 EMAC or modelled H2O2 (Figure 3). This is highly confusing! Please pick one notation and be consistent throughout the manuscript, carefully going over the figure legends and captions. For example, on Figure 4, H2O2 should be labeled as H2O2 HYPHOP to be consistent with Figure 3. Also, MHP should be MHP PSS (or [MHP]$_{PSS}$). I suggest that throughout the text UHP be labeled as [UHP]$_{PSS}$ as it isn't measured. For the figures to be consistent with the text, H2O2 PSS should be labeled [H2O2]$_{PSS}$.

2) While the authors have changed the manuscript in several areas to include a more quantitative comparison between PSS, EMAC and observations, there are still several areas that remain vague and qualitative, and could thus be improved (see in list of minor comments below).

3) In response to my concern about the authors' discussion of the deviation between EMAC and observed H2O2 and ROOH in terms of scavenging efficiencies, the authors added information in their reply that is not included in the revised manuscript. For example, the authors state that "CO in the upper troposphere is lower in EMAC due to weaker convective transport (possibly caused by the low resolution of the global model) as described in Tost et al. (2016) and Tomsche et al. (2019)." This is relevant to their study but not included in the manuscript, where the authors attribute the difference between observed and EMAC H2O2 solely to excessive scavenging. Also, when looking at Tomsche et al. (2019) (their Table 1), it seems that EMAC is overestimating CO in AMA but underestimating CH4, so I am confused. Is this an issue with emissions of CO and CH4? In their response, the authors also say that scavenging of all soluble species is turned off in their sensitivity study and that this is done globally. This is not  clear from the text, which remained unchanged as "To investigate this assumption we performed a sensitivity study with EMAC excluding scavenging." The authors should update the text to be provide more detail to the reader. Finally the authors do not address whether the EMAC very large underestimate in ROOH can be explained by excessive wet removal during transport. It seems that their sensitivity study should be able to answer this question in a straightforward way.

Various other comments:

- Abstract: The deviation from steady-state is mentioned twice in the abstract but not quantified "In general, the observed concentrations are higher than steady-state calculations and EMAC simulations. " "…explaining strong deviations to steady-state calculations which only account for local photochemistry. " The authors should clearly state the magnitude of the deviation from PSS for H2O2.

- Abstract line 29: "strong deviations FROM steady-state calculations "

- Abstract line 30: The expression "Deviations to EMAC simulations" does not mean anything.

- Abstract line 30-33: "Deviations to EMAC simulations are most likely due to uncertainties in the scavenging efficiencies for individual hydroperoxides in deep convective transport to the upper troposphere, corroborated by a sensitivity study." And text lines 410-411 "Differences between observation and EMAC simulation could potentially arise due to uncertainties in the scavenging efficiency for H2O2, as the chemistry does not seem to be a dominant cause of uncertainty." From the response to the reviewers it seems that another explanation is the too weak convective transport in the model due to its relatively coarse resolution. The authors quoted an underestimate in CO. This should be noted in the abstract and text.

- Line 78-80. The authors should state what is actually measured in this paragraph. They only mention that the measurement of ROOH is unspecific without mentioning what is actually measured…

- Line 95. "The hydroperoxide data…" is very vague. It would be helpful to the reader if the authors would actually state what species are measured. Please clarify.

- Lines 132-140. Given that the ROOH measurement is non-specific, I find it confusing that the LOD and uncertainties are given for MHP. Shouldn't they be given for ROOH, which is actually what is measured? Please clarify the text.

- Section 3.3 For clarity [MHP] in all the equations should be replaced with $[MHP]_{pss}$. This would make things consistent with section 3.4, where $[H2O2]_{pss}$ is used. Also, it is unclear if [H2O2] in equation (13) is the observed ($[H2O2]_{obs}$) or PSS value ($[H2O2]_{pss}$). Please clarify the text.

- Lines 254-255. This statement is misleading as it suggests that the EMAC simulations were done only on the OMO flight tracks. I assume that the authors mean to say that the EMAC results were extracted along the OMO flight tracks. Also, what is the vertical resolution of the simulation?

- Line 262. The meaning of "which found on cluster analysis" is unclear. Please reword.

- Figure 6. The regression coefficients listed in the figure legend (0.99, 0.98, 0.99) are not consistent with what is shown in the Figure and what is discussed in the text (line 313-314) stating that the correlation for H2O2 and ROOH are not as strong. Also please specify in the figure legend whether the values are r or $r^2$. Same comment for Figure 10, where the quoted correlations (0.96, 0.97, 0.96) seem too high. Also for Figure 10 is the black line for the entire dataset of only AMA? This should be specified in the figure caption.

- Lines 358-360. I already pointed out in my first review that the statement "The deviations from unity in the slope are within the combined uncertainties of measured and steady-state estimations of H2O2 (51%)" is quite wrong. The authors concurred by saying in their reply that 82% of the points are OUTSIDE the range of uncertainties, but haven't updated the text to correct their misleading statement.

- Section 4.3.2, lines 368-370. I agree with the second reviewer in that comparing the ranges of observed and modeled values is not informative. I suggest that the authors remove the discussion of ranges here as they have added the discussion of medians, which is a more relevant metric.

- Section 4.3.2. The authors go back and forth between referring to AMA and monsoon in the text, Table 2 and Figure 14. Please use one consistent notation.

- Lines 372-374. "For the SH the model simulated H2O2 mixing ratios is four times higher than in the NH background (272 pptv), while the observations only show a median increase by 47 pptv to 211 pptv (Table 2)." This seems incorrect as the observed increase is 111 pptv. It would actually be more relevant to compare the relative increase in both (factor of 2 increase in observed H2O2 between NH and SH background.

- Table 2 and Section 4.3.2. For the discussion of the comparisons of MHP and UHP, it would be more relevant to compare observed ROOH and EMAC MHP+EHP+PAA as the comparison between EMAC and PSS MHP is the comparison between two models, and the comparison for UHP is highly indirect. I suggest that the authors add a column for ROOH in Table 2 and discuss it in the text. The comparison between observed and EMAC ROOH is shown in Figure 17 and shows a very large underestimate of EMAC (factor of 5-10?). A similar suggesting was made by the other reviewer and not addressed by the authors.

- Section 4.3.3 Longitudinal gradient. The EMAC model is a factor of 5-10 lower than observed ROOH, but for H2O2 is it only a factor of ~2 lower compared to observations as Figures 16 and 17 show. The authors do not discuss potential reasons for the more extreme degree of disagreement in ROOH, which I suggest that they add to the text.

- Lines 455-457 and Figure 19. It would be useful to include a panel comparison modeled and observed ROOH in Figure 19. It seems that removing scavenging in EMAC results in between agreement for H2O2, this begs the question as to whether it also leads to improved agreement with ROOH. Please address this.

- Lines 470-475 "Steady-state calculations for H2O2 and MHP based on observed precursors yield much lower values, in particular in the AMA" Lower values relative to what? Also, this is misleading as MHP is not measured while only H2O2 is measured. Finally, the authors are again very qualitative, please give a quantitative comparison between observed and PSS H2O2.

- Lines 477-480 "Convective injection of H2O2 (and MHP) into the upper troposphere over India most likely forms a pool of hydroperoxides in the upper troposphere that subsequently influences the western AMA, giving rise to a significant longitudinal gradient of H2O2 and MHP mixing ratios, with increasing values towards the center of the AMA. It is likely that at least a large part of UHP is due to additional MHP from an up-wind source." I suggest replacing MHP with ROOH as ROOH is the only quantity that is measured. Also, the conclusions should reflect the fact that EMAC significantly underestimates ROOH, but that the underestimate in H2O2 is not quite as large. The conclusion lacks any mention of the sensitivity study results.

---

## Editor Decision (ED1)

Several comments by referee #2 have not or only very (too) briefly been addressed in the revised manuscript. They are cited below (re-ordered by topic). In addition, I have several additional comments that need to be addressed before the manuscript can be accepted for publication.

**Previous comments by Referee #2:**

- 3) […] In their response, the authors also say that scavenging of all soluble species is turned off in their sensitivity study and that this is done globally. This is not clear from the text, which remained unchanged as "To investigate this assumption we performed a sensitivity study with EMAC excluding scavenging." The authors should update the text to be provide more detail to the reader. Finally the authors do not address whether the EMAC very large underestimate in ROOH can be explained by excessive wet removal during transport. It seems that their sensitivity study should be able to answer this question in a straightforward way.

Authors' response:
*Please note that large enhancements of MHP and larger hydroperoxides were not found in the sensitivity study with switched-off scavenging, indicating that the strong underestimation by the model of those species is not due to an overestimation of wet removal in convective clouds.*
*We also change the text on line 455:*
To investigate this assumption we performed a sensitivity study with the wet scavenging for all soluble species being switched-off globally.

- Lines 477-480 "Convective injection of H2O2 (and MHP) into the upper troposphere over India most likely forms a pool of hydroperoxides in the upper troposphere that subsequently influences the western AMA, giving rise to a significant longitudinal gradient of H2O2 and MHP mixing ratios, with increasing values towards the center of the AMA. It is likely that at least a large part of UHP is due to additional MHP from an up-wind source." I suggest replacing MHP with ROOH as ROOH is the only quantity that is measured. Also, the conclusions should reflect the fact that EMAC significantly underestimates ROOH, but that the underestimate in H2O2 is not quite as large. The conclusion lacks any mention of the sensitivity study results.

Authors' response:
We changed the text to:
Convective injection of $H_2O_2$ (and potentially MHP) into the upper troposphere over India most likely forms a pool of hydroperoxides in the upper troposphere that subsequently influences the western AMA, giving rise to a significant longitudinal gradient of $H_2O_2$ and ROOH mixing ratios, with increasing values towards the center of the AMA. It is likely that at least a large part of $UHP_{PSS}$ is due to additional MHP from an up-wind source. A sensitivity study using EMAC with no scavenging tends to reproduce the observed longitudinal gradients in $H_2O_2$, although it does not increase the level of ROOH. The reasons for this different behavior are unclear.

Editor comment 1: Both of these referee comments address the scavenging (or the lack of it) of hydroperoxides. Your response is very brief. Given that EMAC only considers three hydroperoxides, could it be that the properties relevant for scavenging (solubility) are not characteristic for the

majority of the hydroperoxides? Or if it is not scavenging, what other sources processes could occur?

**Previous comment by Referee #2:**
Abstract line 30-33: "Deviations to EMAC simulations are most likely due to uncertainties in the scavenging efficiencies for individual hydroperoxides in deep convective transport to the upper troposphere, corroborated by a sensitivity study." And text lines 410-411 "Differences between observation and EMAC simulation could potentially arise due to uncertainties in the scavenging efficiency for H2O2, as the chemistry does not seem to be a dominant cause of uncertainty." From the response to the reviewers it seems that another explanation is the too weak convective transport in the model due to its relatively coarse resolution. The authors quoted an underestimate in CO. This should be noted in the abstract and text.

Authors' response:
We do not state that convection is too weak. Instead as mentioned above the low resolution of the model leads to an underestimation of species with local source in the inflow region of convection (e.g. CH4).

Editor comment 2: Please add this information to the manuscript

**Previous comment by Referee #2:**
Lines 132-140. Given that the ROOH measurement is non-specific, I find it confusing that the LOD and uncertainties are given for MHP. Shouldn't they be given for ROOH, which is actually what is measured? Please clarify the text.
Due to the fact, that the exact composition of $ROOH_{obs}$ is unknown, and the solubility of different ROOH species can be quite variable, a detection limit for $ROOH_{obs}$ cannot be given. Instead, we calculate an upper limit of the detection limit, assuming that all $ROOH_{obs}$ consists of MHP, the species with the smallest solubility.

Authors' response:
We changed the text to (page 5, line 136): and 9 – 52 pptv for ROOH (median 23 pptv) assuming that $ROOH_{obs}$ is composed of MHP only.

Editor comment 3: Add this information on the solubility to the manuscript.

**Previous comment by Referee #2:**
Lines 358-360. I already pointed out in my first review that the statement "The deviations from unity in the slope are within the combined uncertainties of measured and steady-state estimations of H2O2 (51%)" is quite wrong. The authors concurred by saying in their reply that 82% of the points are OUTSIDE the range of uncertainties, but haven't updated the text to correct their misleading statement.

Authors' response:
See above. Changed to:
The regression coefficient $R^2$ is 0.26 with most of the $H_2O_{2PSS}$ mixing ratios (75%) varying between 0 and 65 pptv with a median value of 15 pptv, while the $H_2O_{2obs}$ extend over a larger range mainly between 10–210 pptv with a median of 150 pptv and thus 10 times higher than for steady-state, which can also be seen in the histograms in Figure 13.

Editor comment 4: Please add the information to the text that > 80% of all points are outside the range of uncertainty.

**Previous comment by Referee #2:**
Table 2 and Section 4.3.2. […] I suggest that the authors add a column for ROOH in Table 2 and discuss it in the text. The comparison between observed and EMAC ROOH is shown in Figure 17 and shows a very large underestimate of EMAC (factor of 5-10?). A similar suggesting was made by the other reviewer and not addressed by the authors.
Authors' response:
We added a sentence to Section 4.3.2: In general EMAC tends to strongly underestimate measured total hydroperoxide in all air masses by a factor of 5 -10.

Editor comment 5: Given that both referees suggested to add a column to Table 2 reporting the ROOH values from EMAC and observations, your response is not sufficient. It seems that based on your discussion, you should be able to add these values.

**Previous comment by Referee #2:**
Lines 455-457 and Figure 19. It would be useful to include a panel comparison modeled and observed ROOH in Figure 19. It seems that removing scavenging in EMAC results in between agreement for H2O2, this begs the question as to whether it also leads to improved agreement with ROOH. Please address this.
Authors' response:
See our comment above.

Editor comment 6: Your response to the referee comment is too brief. I assume that the results for ROOH(EMAC) are not exactly identical with and without scavenging. I second the referee's suggestion to contrast the small predicted effect of ROOH scavenging to the much larger one by H2O2 by adding a panel for ROOH in Figure 19. Could you estimate which scavenging coefficient would be needed for ROOH to reach a better model/observation agreement?

**Previous comment by Referee #2:**
Lines 477-480 "Convective injection of H2O2 (and MHP) into the upper troposphere over India most likely forms a pool of hydroperoxides in the upper troposphere that subsequently influences the western AMA, giving rise to a significant longitudinal gradient of H2O2 and MHP mixing ratios, with increasing values towards the center of the AMA. It is likely that at least a large part of UHP is due to additional MHP from an up-wind source." I suggest replacing MHP with ROOH as ROOH is the only quantity that is measured. Also, the conclusions should reflect the fact that EMAC significantly underestimates ROOH, but that the underestimate in H2O2 is not quite as large. The conclusion lacks any mention of the sensitivity study results.
Authors' response:
We changed the text to:
Convective injection of $H_2O_2$ (and potentially MHP) into the upper troposphere over India most likely forms a pool of hydroperoxides in the upper troposphere that subsequently influences the western AMA, giving rise to a significant longitudinal gradient of $H_2O_2$ and ROOH mixing ratios, with increasing values towards the center of the AMA. It is likely that at least a large part of UHP$_{PSS}$ is due to additional MHP from an up-wind source. A sensitivity study using EMAC with no scavenging tends

to reproduce the observed longitudinal gradients in $H_2O_2$, although it does not increase the level of ROOH. The reasons for this different behavior are unclear.

Editor comment 7: Your response is unclear and confusing. The referee had suggested to replace MHP in this context by ROOH. Your response implies that the large underestimate of ROOH is exclusively due to MHP. Why do you exclude the possibility that other ROOH are underestimated?

**Additional editor comments**

8) I appreciate that you added indices (PSS, EMAC, obs) to the species names. However, thy do not seem correct at all places and sometimes they are even confusing. Some examples are listed below. Please check the complete manuscript for their use.

   - l. 23: 'We observed enhanced mixing ratios of […] MHP(PSS), UHP(PSS)' – is consistent with the previous sentence ('Observations are compared to photostationary calculations…')

   - l. 212 and Equation 5: Shouldn't it be H2O2(PSS) and not H2O2(obs)?

9) l. 30 and l. 31: 'Deviations' is very qualitative. Please quantify the value and state whether there is a consistent trend (over- or underestimate) between model results and observations.

10) l. 134 and l. 138: Clarify 'to be mainly MHP' or 'MHP only'?

11) l. 234/5: Should this read
'The combination of Eq. 5 and 12 **results** in Eq. 13 which was used to calculate the MHP$^{PSS}$ concentrations **based on** the observed mixing ratios during OMO.'?

12) l. 244: 'To estimate the composition of the organic hydroperoxides…' should be replaced by 'To estimate the contribution of MHP to the total organic hydroperoxides…' (or similar) because you estimate does not yield any further information on the composition of ROOH.

13) Equation 15: 1) The index obs should be before exponent, i.e. $[HO_2]^{obs\ 2}$
2) Specify the reaction denoted by $k_{OH}$ (i.e. similar to Eq.-13 etc $k_{OH+\ldots}$

14) l. 323-325: I do not understand this sentence. Why does the fact that you see a correlation between UHP)(PSS) and ROOH(obs) imply that this is 'mainly due to UHP(PSS)'? How does the correlation of MHP(PSS) look like?

15) l. 371/2:: The text here is confusing. Should it read
'For the SH the model simulated H2O2$^{EMAC}$ mixing ratios **(272 pptv)** are four times higher than in the NH background **(66 pptv),**`

16) l. 373: Unless I misunderstand this sentence or the table, shouldn't it be '(**100** pptv to 211 pptv) (Table 2).'?

17) l. 374: 'measured' should be removed here

18) l. 378 and 379: For both EMAC and observations, the NH background and AMA values show about a 10% difference. I suggest rewording the sentence in this regard rather than saying 'almost identical' versus 'small difference', respectively.

19) l. 379-381: Please add the predicted values to this sentence so it is easier to follow the data in the table.

20) l. 388-390: These sentences do not read well. You mix relative differences ('factor of four') with absolute differences ('200 ppt'). Either use only the most meaningful difference or report both relative and absolute differences.

21) l. 400: should 'blue' be 'purple'?

22) l. 409: Where can one see the 'rather large deviation of 150 pptv'? The average difference between the PSS values and observations in Figure 16 look less than that.

23) l. 414: 'Assuming that MHP is also enhanced in the outflow of deep convention…' – Is this assumption based on literature (if so, add reference(s)) or on observations in the current study?

24) l. 426: Table 2 shows a value of 100 pptv, not 115 pptv.

25) l. 456: Add 'EMAC' to H2O2.

26) l. 478 and l. 483: This is contradictory: You define UHP as all organic hydroperoxides except MHP. The sentence in l. 478 seems consistent with this definition 'a large contribution of an unidentified organic hydroperoxide (UHPPSS)' whereas the later text 'a large part of UHPPSS is due to additional MHP from an up-wind source' is inconsistent. Please make sure that you use the definitions of ROOH, UHP, MHP consistently.

---

## Author Response (AR2)

**Anonymous Referee #1.**
(black: RC, red: AC, blue: changed in manuscript)

Thank you very much for your comments. We herewith answer point-by-point to your referee report:

1. I am glad to see that the authors have modified the text to denote H2O2 and MHP calculated from the photostationary steady-state method as PSS-H2O2 and PSS-MHP. I see that in the Results section. It would also be good to use that nomenclature in section 3.3, namely with equations 13 and 14 as well as in the text:
The total uncertainty of MHP from the calculation → The total uncertainty of PSS-MHP
the calculated concentration of MHP was subtracted → the PSS-MHP was subtracted

Following your suggestion and a similar one from a second reviewer we have changed the nomenclature throughout the text, tables, formulas and figures to:

$H_2O_2^{obs}$, $H_2O_2^{PSS}$, $H_2O_2^{EMAC}$, $ROOH^{obs}$, $ROOH^{EMAC}$, $MHP^{PSS}$, $MHP^{EMAC}$, $UHP^{PSS}$

according to this nomenclature we change the text to:

The total uncertainty of MHP from the calculation → The total uncertainty of $MHP^{PSS}$ the
calculated concentration of MHP was subtracted → the $MHP^{PSS}$ was subtracted

2. To be more explicit about my comment on CH3O2 and HO2 ratios and production, I suggest the following. The line "Similarly the CH3O2 to HO2 ratio can be deduced from Eq. 6." just before equation 6 should say something like, "Because individual peroxy radicals were not measured, the CH3O2 to HO2 ratio must be estimated from their production and loss terms. This ratio can be deduced as written in equation 6."
I agree with the authors that P(CH3O2) and P(HO2) do not need to be written explicitly when they are in equation 12.

We changed the text accordingly.

3. I was surprised to find out that HCHO had values below the detection limit of 30 pptv. In the AMA I would expect much higher mixing ratios because of the convective lofting. Perhaps the added sentences in section 3.3 refer to the NH background values. If so, I suggest saying, "HCHO in the *background NH* was below the detection limit of 30 pptv,"

Actually, HCHO in the UT was almost always below the 1σ-detection limit of 30 pptv. This can be due to the rather short lifetime of HCHO, which is a couple of hours and thus much shorter than the H2O2 lifetime.

4. My suggestion of putting some figures together is no longer needed now that the

scatter plots are multi-paneled based on the other reviewer's suggestion of additional plots.

Figure 3 caption should be revised with the update to the flight track colors.

We do not understand this request. The colors used in the flight track (Fig. 2) indicate altitude not origin, while the color bar in Fig. 3 stands for air mass origin.

Figure 5: I still see the triangles as more red than purple, and I'm looking at the computer screen. Since it is not exactly the same red as the monsoon category at the top, perhaps it is a dark red or purple-red combination. Below I pasted the figure and then added a text box with the colors MSWord thinks are purple, red, and dark red.

[Figure]

We will use a different pink color (for MHP$^{PSS}$ to avoid confusion with the red color used for the air mass origin.

Figure 6. Here the acetone mixing ratios are shown as lilac squares and not green circles as stated in the caption.

changed

Table 1. region → method

changed

Table 2. It still would be good to clarify if the values reported are from all flights or a subset or flights.

We state in the manuscript, that the values reported in Table 2 are from the upper troposphere deduced from all flights (e.g Fig.8 or Lines 289 and 328).

Answer to:
Thank you very much for your comments. We herewith answer point-by-point to your referee report:

1) Lack of consistent notation for what is calculated, inferred, and observed in the peroxides. The authors go back and forth between various notations in the text and figures. PSS-H2O2, H2O2 PSS, [H2O2]$_{PSS}$. Similarly UHP is referred to as UHP or PSS-UHP, and MHP is referred to as MHP or PSS-MHP. Observed H2O2 is referred to as H2O2 HYPHOP or [H2O2]$_{HYPHOP}$, or in situ H2O2 or H2O2. EMAP H2O2 is referred to as H2O2 EMAC or modelled H2O2 (Figure 3). This is highly confusing! Please pick one notation and be consistent throughout the manuscript, carefully going over the figure legends and captions. For example, on Figure 4, H2O2 should be labeled as H2O2 HYPHOP to be consistent with Figure 3. Also, MHP should be MHP PSS (or [MHP]$_{PSS}$). I suggest that throughout the text UHP be labeled as [UHP]$_{PSS}$ as it isn't measured. For the figures to be consistent with the text, H2O2 PSS should be labeled [H2O2]$_{PSS}$.

Following the suggestion and a similar one from a second referee we have changed the nomenclature throughout the text, tables, formulas and figures to:

$H_2O_2^{obs}$, $H_2O_2^{PSS}$, $H_2O_2^{EMAC}$, $ROOH^{obs}$, $ROOH^{EMAC}$, $MHP^{PSS}$, $MHP^{EMAC}$, $UHP^{PSS}$

2) While the authors have changed the manuscript in several areas to include a more quantitative comparison between PSS, EMAC and observations, there are still several areas that remain vague and qualitative, and could thus be improved (see in list of minor comments below).

3) In response to my concern about the authors' discussion of the deviation between EMAC and observed H2O2 and ROOH in terms of scavenging efficiencies, the authors added information in their reply that is not included in the revised manuscript. For example, the authors state that "CO in the upper troposphere is lower in EMAC due to weaker convective transport (possibly caused by the low resolution of the global model) as described in Tost et al. (2016) and Tomsche et al. (2019)." This is relevant to their study but not included in the manuscript, where the authors attribute the difference between observed and EMAC H2O2 solely to excessive scavenging. Also, when looking at Tomsche et al. (2019) (their Table 1), it seems that EMAC is overestimating CO in AMA but underestimating CH4, so I am confused. Is this an issue with emissions of CO and CH4? In their response, the authors also say that scavenging of all soluble species is turned off in their sensitivity study and that this is done globally. This is not clear from the text, which remained unchanged as "To investigate this assumption we performed a sensitivity study with EMAC excluding scavenging." The authors should update the text to be provide more detail to the reader. Finally the

authors do not address whether the EMAC very large underestimate in ROOH can be explained by excessive wet removal during transport. It seems that their sensitivity study should be able to answer this question in a straightforward way.

We apologize for a typo in our original reply to the reviewer. Instead of CO it should read $CH_4$. EMAC has a general tendency to overestimate CO in the UT, especially for the NH background, while it tends to underestimate $CH_4$ (Tosmche et al., 2019). The deviations in general are not significant, with the exception of $CH_4$ in the AMA (Table 1 in Tomsche et al., 2019). As discussed in Tomsche et al. 2019 the $CH_4$ mixing ratio in the AMA depends of the co-location of convection and underlying methane sources. The model resolution of 2.5° x 2.5° in not sufficient to resolve small scale variations in both convection and source distribution. The $H_2O_2$ mixing ratio over the Indian sub-continent is not expected to show large spatial variations, since latitudinal gradient are generally small (see e.g. Klippel et al., 2011). Therefore, we do not consider the model resolution as an alternative explanation for the deviation between $H_2O_2^{obs}$ and $H_2O_2^{EMAC}$.

As shown in Fig. 14 (middle and lower panel) EMAC tends to significantly underestimate MHP and higher organic hydroperoxides, much stronger than $H_2O_2$. The reasons for this behavior are unknown. But this seems not to be associated with too strong washout, since the sensitivity study indicates no significant enhancement for MHP with switched-off wet scavenging.

We will add the following sentence to the discussion of the sensitivity study:

Please note that large enhancements of MHP and larger hydroperoxides were not found in the sensitivity study with switched-off scavenging, indicating that the strong underestimation by the model of those species is not due to an overestimation of wet removal in convective clouds.

We also change the text on line 455: To investigate this assumption we performed a sensitivity study with the wet scavenging for all soluble species being switched-off globally.

Various other comments:

- Abstract: The deviation from steady-state is mentioned twice in the abstract but not quantified "In general, the observed concentrations are higher than steady-state calculations and EMAC simulations. " "…explaining strong deviations to steady-state calculations which only account for local photochemistry. " The authors should clearly state the magnitude of the deviation from PSS for H2O2.

We changed the text to:

In general, the observed concentrations are higher than steady-state calculations and EMAC simulations by a factor of 3 and 2, respectively.

- Abstract line 29: "strong deviations FROM steady-state calculations "

  changed

- Abstract line 30: The expression "Deviations to EMAC simulations" does not mean anything.

  Changed to: Deviations between $H_2O_2$ observations and EMAC simulations…

- Abstract line 30-33: "Deviations to EMAC simulations are most likely due to uncertainties in the scavenging efficiencies for individual hydroperoxides in deep convective transport to the upper troposphere, corroborated by a sensitivity study." And text lines 410-411 "Differences between observation and EMAC simulation could potentially arise due to uncertainties in the scavenging efficiency for H2O2, as the chemistry does not seem to be a dominant cause of uncertainty." From the response to the reviewers it seems that another explanation is the too weak convective transport in the model due to its relatively coarse resolution. The authors quoted an underestimate in CO. This should be noted in the abstract and text.

  We do not state that convection is too weak. Instead as mentioned above the low resolution of the model leads to an underestimation of species with local source in the inflow region of convection (e.g. CH4).

- Line 78-80. The authors should state what is actually measured in this paragraph. They only mention that the measurement of ROOH is unspecific without mentioning what is actually measured…

  We changed the sentence to: Since the measurements of the sum of all organic hydroperoxides do not differentiate between different species, we …

- Line 95. "The hydroperoxide data…" is very vague. It would be helpful to the reader if the authors would actually state what species are measured. Please clarify.

  We changed the sentence to: The hydroperoxide data ($H_2O_2^{obs}$ and total organic hydroperoxides $ROOH^{obs}$)…

- Lines 132-140. Given that the ROOH measurement is non-specific, I find it confusing that the LOD and uncertainties are given for MHP. Shouldn't they be given for ROOH, which is actually what is measured? Please clarify the text.

  Due to the fact, that the exact composition of $ROOH^{obs}$ is unknown, and the solubility of different ROOH species can be quite variable, a detection limit for $ROOH^{obs}$ cannot be given. Instead, we calculate an upper limit of the detection limit, assuming that all $ROOH^{obs}$ consists of MHP, the species with the smallest solubility.

  We changed the text to (page 5, line 136): and 9 – 52 pptv for ROOH (median 23 pptv) assuming that $ROOH^{obs}$ is composed of MHP only.

- Section 3.3 For clarity [MHP] in all the equations should be replaced with [MHP]$_{pss}$. This would make things consistent with section 3.4, where [H2O2]$_{pss}$ is used. Also, it is unclear if [H2O2] in equation (13) is the observed ([H2O2]$_{obs}$) or PSS value ([H2O2]$_{pss}$). Please clarify the text.

  Done

- Lines 254-255. This statement is misleading as it suggests that the EMAC simulations were done only on the OMO flight tracks. I assume that the authors mean to say that the EMAC results were extracted along the OMO flight tracks. Also, what is the vertical resolution of the simulation?

  We changed the text to:
  For this study EMAC simulations (T42L90, 2.8° x 2.8° horizontal resolution, 90 vertical levels to 0.01 hPa, time resolution 12 min) were sampled along the OMO flights tracks.

- Line 262. The meaning of "which found on cluster analysis" is unclear. Please reword.

  We changed the text to: …, based on the analysis of a cluster of trajectories.

- Figure 6. The regression coefficients listed in the figure legend (0.99, 0.98, 0.99) are not consistent with what is shown in the Figure and what is discussed in the text (line 313-314) stating that the correlation for H2O2 and ROOH are not as strong. Also please specify in the figure legend whether the values are r or r$^2$. Same comment for Figure 10, where the quoted correlations (0.96, 0.97, 0.96) seem too high. Also for Figure 10 is the black line for the entire dataset of only AMA? This should be specified in the figure caption.

  Due to an error in the fit routine, we overestimated R$^2$ throughout the manuscript. This error has been corrected.

- Lines 358-360. I already pointed out in my first review that the statement "The deviations from unity in the slope are within the combined uncertainties of measured and steady-state estimations of H2O2 (51%)" is quite wrong. The authors concurred by saying in their reply that 82% of the points are OUTSIDE the range of uncertainties, but haven't updated the text to correct their misleading statement.

  See above.  Changed to:
  The regression coefficient R2 is 0.26 with most of the $H_2O_2^{PSS}$ mixing ratios (75%) varying between 0 and 65 pptv with a median value of 15 pptv, while the $H_2O_2^{obs}$ extend over a larger range mainly between 10–210 pptv with a median of 150 pptv and thus 10 times higher than for steady-state, which can also be seen in the histograms in Figure 13.

- Section 4.3.2, lines 368-370. I agree with the second reviewer in that comparing the ranges of observed and modeled values is not informative. I suggest that the authors remove the discussion of ranges here as they have added the discussion of medians, which is a more relevant metric.

  removed

- Section 4.3.2. The authors go back and forth between referring to AMA and monsoon in the text, Table 2 and Figure 14. Please use one consistent notation.

  We changed the text using AMA consistently.

- Lines 372-374. "For the SH the model simulated H2O2 mixing ratios is four times higher than in the NH background (272 pptv), while the observations only show a median increase by 47 pptv to 211 pptv (Table 2)." This seems incorrect as the observed increase is 111 pptv. It would actually be more relevant to compare the relative increase in both (factor of 2 increase in observed H2O2 between NH and SH background.

  We changed the text to: For the SH the model simulated $H_2O_2^{EMAC}$ mixing ratios are four times higher than in the NH background (272 pptv), while the $H_2O_2^{obs}$ only show a median increase by roughly a factor of 2 (115 pptv to 211 pptv) (Table 2).

- Table 2 and Section 4.3.2. For the discussion of the comparisons of MHP and UHP, it would be more relevant to compare observed ROOH and EMAC MHP+EHP+PAA as the comparison between EMAC and PSS MHP is the comparison between two models, and the comparison for UHP is highly indirect. I suggest that the authors add a column for ROOH in Table 2 and discuss it in the text. The comparison between observed and EMAC ROOH is shown in Figure 17 and shows a very large underestimate of EMAC (factor of 5-10?). A similar suggesting was made by the other reviewer and not addressed by the authors.

  We added a sentence to Section 4.3.2: In general EMAC tends to strongly underestimate measured total hydroperoxide in all air masses by a factor of 5 -10.

- Section 4.3.3 Longitudinal gradient. The EMAC model is a factor of 5-10 lower than observed ROOH, but for H2O2 is it only a factor of ~2 lower compared to observations as Figures 16 and 17 show. The authors do not discuss potential reasons for the more extreme degree of disagreement in ROOH, which I suggest that they add to the text.
  As mentioned above, we added the following discussion to paragraph 4.4 at line 457: Please note that large enhancements of $MHP^{EMAC}$ and $ROOH^{EMAC}$ were not found in the sensitivity study with switched-off scavenging, indicating that the strong underestimation by the model of those species is not due to an overestimation of wet removal in convective clouds.

- Lines 455-457 and Figure 19. It would be useful to include a panel comparison modeled and observed ROOH in Figure 19. It seems that removing scavenging in EMAC results in between agreement for H2O2, this begs the question as to whether it also leads to improved agreement with ROOH. Please address this.

  See our comment above.

- Lines 470-475 "Steady-state calculations for H2O2 and MHP based on observed precursors yield much lower values, in particular in the AMA" Lower values relative to what? Also, this is misleading as MHP is not measured while only H2O2 is measured. Finally, the authors are again very qualitative, please give a quantitative comparison between observed and PSS H2O2.

  Steady-state calculations for $H_2O_2^{PSS}$ and $MHP^{PSS}$ based on observed precursors yield much lower values compared to $H_2O_2^{obs}$ and $MHP^{PSS}$ by roughly a factor of 3, in particular in the AMA, resulting in a large contribution of an unidentified organic hydroperoxide ($UHP^{PSS}$) in air masses affected by the AMA.

- Lines 477-480 "Convective injection of H2O2 (and MHP) into the upper troposphere over India most likely forms a pool of hydroperoxides in the upper troposphere that subsequently influences the western AMA, giving rise to a significant longitudinal gradient of H2O2 and MHP mixing ratios, with increasing values towards the center of the AMA. It is likely that at least a large part of UHP is due to additional MHP from an up-wind source." I suggest replacing MHP with ROOH as ROOH is the only quantity that is measured. Also, the conclusions should reflect the fact that EMAC significantly underestimates ROOH, but that the underestimate in H2O2 is not quite as large. The conclusion lacks any mention of the sensitivity study results.

  We changed the text to:

[revised manuscript text omitted]
{[\text{MHP}]^{\text{PSS}}}{[\text{H}_2\text{O}_2]^{\text{obs}}} = \frac{k_{\text{CH}_3\text{O}_2+\text{HO}_2} \cdot [\text{CH}_3\text{O}_2] \cdot [\text{HO}_2]}{k_{\text{HO}_2+\text{HO}_2} \cdot [\text{HO}_2]^2} \cdot \frac{k_{\text{H}_2\text{O}_2+\text{OH}} \cdot [\text{OH}] + J_{\text{H}_2\text{O}_2}}{k_{\text{MHP}+\text{OH}} \cdot [\text{OH}] + J_{\text{MHP}}},$$
(5)

Because individual peroxy radicals were not measured, the $\text{CH}_3\text{O}_2$ to $\text{HO}_2$ ratio must be estimated from their production and loss terms. This ratio can be deduced as written in Eq. 6.

$$\frac{[\text{CH}_3\text{O}_2]}{[\text{HO}_2]} = \frac{L(\text{HO}_2) \cdot P(\text{CH}_3\text{O}_2)}{P(\text{HO}_2) \cdot L(\text{CH}_3\text{O}_2)},$$
(6)

Dominant loss processes for $\text{HO}_2$ and $\text{CH}_3\text{O}_2$ are reactions with NO and the production of $\text{H}_2\text{O}_2$ and MHP, respectively, neglecting the production of peroxy nitrates due to low $\text{NO}_2$ concentrations in the UT (Eq. 7 and 8).

$$L(\text{HO}_2) = k_{\text{CH}_3\text{O}_2+\text{HO}_2} \cdot [\text{CH}_3\text{O}_2] \cdot [\text{HO}_2] + k_{\text{HO}_2+\text{NO}} \cdot [\text{HO}_2] \cdot [\text{NO}] + k_{\text{HO}_2+\text{HO}_2} \cdot [\text{HO}_2]^2,$$
(7)

$$L(\text{CH}_3\text{O}_2) = k_{\text{CH}_3\text{O}_2+\text{HO}_2} \cdot [\text{CH}_3\text{O}_2] \cdot [\text{HO}_2] + k_{\text{CH}_3\text{O}_2+\text{NO}} \cdot [\text{CH}_3\text{O}_2] \cdot [\text{NO}],$$
(8)

The first terms on the right side of both equations are identical. The second terms are dominated by the rate coefficients of the reactions with NO and the NO concentration. For the calculations of the rate coefficients the mean temperature of 259.18 K, the mean altitude of 10,992.8 m and the mean pressure of 22,932.9 Pa were used. The resulting values are shown in Eq. 9–11. As the relative humidity is very low in the upper troposphere the water dependence in eq. 11 was neglected.

$$k_{\text{HO}_2+\text{NO}} = 3.45 \cdot 10^{-12} \cdot \exp^{\frac{270}{T}} = 9.78 \cdot 10^{-12} \frac{\text{cm}^3}{\text{molecule} \cdot \text{s}},$$
(9)

$$k_{\text{CH}_3\text{O}_2+\text{NO}} = 2.3 \cdot 10^{-12} \cdot \exp^{\frac{360}{T}} = 9.22 \cdot 10^{-12} \frac{\text{cm}^3}{\text{molecule} \cdot \text{s}},$$
(10)

$$k_{\text{HO}_2+\text{HO}_2} = 2.2 \cdot 10^{-13} \cdot \exp^{\frac{600}{T}} + 1.9 \cdot 10^{-33} \cdot [\text{N}_2] \cdot \exp^{\frac{980}{T}} = 2.64 \cdot 10^{-12} \frac{\text{cm}^3}{\text{molecule} \cdot \text{s}},$$
(11)

This indicates that the reaction of $\text{HO}_2$ with NO is more than a factor of 3 faster than the self-reaction. The measured NO concentration is an order of magnitude larger than measured $\text{HO}_2$, so that reaction with NO is the dominant process for both $\text{HO}_2$ and $\text{CH}_3\text{O}_2$ resulting in similar loss rates for both radicals in the UT. Thus, the ratio of $\text{CH}_3\text{O}_2$ to $\text{HO}_2$ is dominated by their production rates (Eq. 12).

$$\frac{[\text{CH}_3\text{O}_2]}{[\text{HO}_2]} = \frac{P(\text{CH}_3\text{O}_2)}{P(\text{HO}_2)} = \frac{k_{\text{CH}_4+\text{OH}} \cdot [\text{CH}_4] \cdot [\text{OH}]}{k_{\text{CO}+\text{OH}} \cdot [\text{CO}] \cdot [\text{OH}]},$$
(12)

The combination of Eq. 5 and 12 yields in Eq. 13 which was used to calculate the $\text{MHP}^{\text{PSS}}$ concentrations out of the observed mixing ratios during OMO.

$$[\text{MHP}]^{\text{PSS}} = \frac{k_{\text{CH}_3\text{O}_2+\text{HO}_2}}{k_{\text{HO}_2+\text{HO}_2}} \cdot \frac{k_{\text{H}_2\text{O}_2+\text{OH}} \cdot [\text{OH}]^{\text{obs}} + J_{\text{H}_2\text{O}_2}^{\text{obs}}}{k_{\text{MHP}+\text{OH}} \cdot [\text{OH}]^{\text{obs}} + J_{\text{MHP}}^{\text{obs}}} \cdot \frac{k_{\text{CH}_4+\text{OH}} \cdot [\text{CH}_4]^{\text{obs}}}{k_{\text{CO}+\text{OH}} \cdot [\text{CO}]^{\text{obs}}} \cdot [\text{H}_2\text{O}_2]^{\text{obs}},$$
(13)

Please note that other sources of $\text{HO}_2$ and $\text{CH}_3\text{O}_2$, in particular the photolysis of formaldehyde (HCHO) and acetaldehyde, respectively have been neglected. This is justified by the generally low mixing ratios of these species at high altitudes. Measurements of HCHO with the TRISTAR instrument yielded values below the detection limit of 30 pptv, and although acetaldehyde was not measured, we assume that its mixing ratio is within a factor of two of those for HCHO.

The total uncertainty of $\text{MHP}^{\text{PSS}}$ from the calculation according to equation 13 can be deduced from error propagation taking into account uncertainties in $\text{OH}^{\text{obs}}$ (17.1%), $J_{\text{H2O2}}^{\text{obs}}$ (15%), $J_{\text{MHP}}^{\text{obs}}$ (25%), $\text{CH}_4^{\text{obs}}$ (0.275%), $\text{CO}^{\text{obs}}$ (5.1%), $\text{H}_2\text{O}_2^{\text{obs}}$ (25%) and rate constants, to be of the order of 45% (1σ).

To estimate the composition of the organic hydroperoxides the calculated concentration of $\text{MHP}^{\text{PSS}}$ was subtracted from the measured sum of all organic hydroperoxides $\text{ROOH}^{\text{obs}}$. This leads to a concentration of unidentified organic hydroperoxides ($\text{UHP}^{\text{PSS}}$) (Eq. 14).

$$[\text{UHP}]^{\text{PSS}} = [\text{ROOH}]^{\text{obs}} - [\text{MHP}]^{\text{PSS}},$$
(14)

[revised manuscript text omitted]

avg±sdev | 272
15–409
272±68 | 211
85–510
238±105 | 116
2–502
155±125 | 152
40–346
191±95 | 33
1–132
42±24 | 122
LOD–334
125±102 |

---

## Author Response (AR3)

Several comments by referee #2 have not or only very (too) briefly been addressed in the revised manuscript. They are cited below (re-ordered by topic). In addition, I have several additional comments that need to be addressed before the manuscript can be accepted for publication.

**Previous comments by Referee #2:**

- 3) […] In their response, the authors also say that scavenging of all soluble species is turned off in their sensitivity study and that this is done globally. This is not clear from the text, which remained unchanged as "To investigate this assumption we performed a sensitivity study with EMAC excluding scavenging." The authors should update the text to be provide more detail to the reader. Finally the authors do not address whether the EMAC very large underestimate in ROOH can be explained by excessive wet removal during transport. It seems that their sensitivity study should be able to answer this question in a straightforward way.

Authors' response:

*Please note that large enhancements of MHP and larger hydroperoxides were not found in the sensitivity study with switched-off scavenging, indicating that the strong underestimation by the model of those species is not due to an overestimation of wet removal in convective clouds. We also change the text on line 455:*

To investigate this assumption we performed a sensitivity study with the wet scavenging for all soluble species being switched-off globally.

- Lines 477-480 "Convective injection of $H_2O_2$ (and MHP) into the upper troposphere over India most likely forms a pool of hydroperoxides in the upper troposphere that subsequently influences the western AMA, giving rise to a significant longitudinal gradient of $H_2O_2$ and MHP mixing ratios, with increasing values towards the center of the AMA. It is likely that at least a large part of UHP is due to additional MHP from an up-wind source." I suggest replacing MHP with ROOH as ROOH is the only quantity that is measured. Also, the conclusions should reflect the fact that EMAC significantly underestimates ROOH, but that the underestimate in $H_2O_2$ is not quite as large. The conclusion lacks any mention of the sensitivity study results.

Authors' response:

We changed the text to:

Convective injection of $H_2O_2$ (and potentially MHP) into the upper troposphere over India most likely forms a pool of hydroperoxides in the upper troposphere that subsequently influences the western AMA, giving rise to a significant longitudinal gradient of $H_2O_2$ and ROOH mixing ratios, with increasing values towards the center of the AMA. It is likely that at least a large part of $UHP_{PSS}$ is due to additional MHP from an up-wind source. A sensitivity study using EMAC with no scavenging tends to reproduce the observed longitudinal gradients in $H_2O_2$, although it does not increase the level of ROOH. The reasons for this different behavior are unclear.

Editor comment 1: Both of these referee comments address the scavenging (or the lack of it) of hydroperoxides. Your response is very brief. Given that EMAC only considers three hydroperoxides, could it be that the properties relevant for scavenging (solubility) are not characteristic for the

majority of the hydroperoxides? Or if it is not scavenging, what other sources processes could occur?

As stated on page 12 line 382-383: "Data for UHP$^{EMAC}$ in the model are calculated from the sum of simulated ethyl hydroperoxide (EHP) and peroxyacetic acid, which are the only non-methyl organic hydroperoxides in the free troposphere according to the model." This means that those two species are the only ones that have non-zero mixing ratios in the free troposphere and not that EMAC only simulates those species. Therefore we change the text to:

Data for UHP$^{EMAC}$ in the model are calculated from the sum of simulated ethyl hydroperoxide (EHP) and peroxyacetic acid, which are the only non-methyl organic hydroperoxides in the free troposphere according to the model with non-zero mixing ratios.

For an answer to the questions of the two referees and the editor with respect to the role of scavenging, see our response to Editor comment 2.

**Previous comment by Referee #2:**

Abstract line 30-33: "Deviations to EMAC simulations are most likely due to uncertainties in the scavenging efficiencies for individual hydroperoxides in deep convective transport to the upper troposphere, corroborated by a sensitivity study." And text lines 410-411 "Differences between observation and EMAC simulation could potentially arise due to uncertainties in the scavenging efficiency for H2O2, as the chemistry does not seem to be a dominant cause of uncertainty." From the response to the reviewers it seems that another explanation is the too weak convective transport in the model due to its relatively coarse resolution. The authors quoted an underestimate in CO. This should be noted in the abstract and text.

Authors' response:

We do not state that convection is too weak. Instead as mentioned above the low resolution of the model leads to an underestimation of species with local source in the inflow region of convection (e.g. CH4).

Editor comment 2: Please add this information to the manuscript

We added the following paragraph at the end of the discussion (Section 4.4):

Please note that EMAC has a general tendency to overestimate CO in the UT, especially for the NH background, while it tends to underestimate CH$_4$ (Tomsche et al., 2019). The deviations in general are not significant, with the exception of CH$_4$ in the AMA (Table 1 in Tomsche et al., 2019). As discussed in Tomsche et al. 2019 the CH$_4$ mixing ratio in the AMA depends of the co-location of convection and underlying methane sources. The model resolution of 2.8° x 2.8° is not sufficient to resolve small-scale variations in both convection and CH$_4$ source distribution. The H$_2$O$_2$ mixing ratio over the Indian sub-continent is not expected to show large spatial variations, since latitudinal gradients are generally small (see e.g. Klippel et al., 2011). Therefore, we do not expect that the model resolution will have a strong influence on the deviation between H$_2$O$_2{}^{obs}$ and H$_2$O$_2{}^{EMAC}$. Another uncertainty arises from missing information on the absolute mixing ratios of H$_2$O$_2$, MHP and higher organic hydroperoxides in the inflow region of deep convection over India, since observations of these species in the boundary layer over India are not available. Note that the amount of hydrogenperoxide and organic hydroperoxides transported to the upper troposphere depends on the scavenging efficiency and on the mixing ratios of the individual species in the inflow region (Barth et al., 2016, Bozem et

al., 2017). Thus, an underestimation of hydroperoxides in the upper troposphere after convective injection can be either due to an underestimation of the scavenging efficiency for individual species, an underestimation of their mixing ratio in the inflow region or a combination of both, and might differ for individual hydroperoxides. Due to a lack of observations in the inflow and outflow region of convection over India this question cannot be resolved in this study.

**Previous comment by Referee #2:**

Lines 132-140. Given that the ROOH measurement is non-specific, I find it confusing that the LOD and uncertainties are given for MHP. Shouldn't they be given for ROOH, which is actually what is measured? Please clarify the text.

Authors' response:

Due to the fact, that the exact composition of $ROOH_{obs}$ is unknown, and the solubility of different ROOH species can be quite variable, a detection limit for $ROOH_{obs}$ cannot be given. Instead, we calculate an upper limit of the detection limit, assuming that all $ROOH_{obs}$ consists of MHP, the species with the smallest solubility.

We changed the text to (page 5, line 136): and 9 – 52 pptv for ROOH (median 23 pptv) assuming that $ROOH_{obs}$ is composed of MHP only.

Editor comment 3: Add this information on the solubility to the manuscript.

We changed the text to (page 5, line 138): and 9 – 52 pptv for $ROOH^{obs}$ (median 23 pptv) assuming that $ROOH^{obs}$ is composed of MHP only. Please note, that due to the fact, that the exact composition of $ROOH^{obs}$ is unknown, and the solubility of different ROOH species can be quite variable, a detection limit for $ROOH^{obs}$ cannot be given. Instead, we calculate an upper limit of the detection limit, assuming that all $ROOH^{obs}$ consists of MHP, the species with the smallest solubility.

**Previous comment by Referee #2:**

Lines 358-360. I already pointed out in my first review that the statement "The deviations from unity in the slope are within the combined uncertainties of measured and steady-state estimations of H2O2 (51%)" is quite wrong. The authors concurred by saying in their reply that 82% of the points are OUTSIDE the range of uncertainties, but haven't updated the text to correct their misleading statement.

Authors' response:

See above. Changed to:

The regression coefficient R2 is 0.26 with most of the $H_2O_{2PSS}$ mixing ratios (75%) varying between 0 and 65 pptv with a median value of 15 pptv, while the $H_2O_{2obs}$ extend over a larger range mainly between 10–210 pptv with a median of 150 pptv and thus 10 times higher than for steady-state, which can also be seen in the histograms in Figure 13.

Editor comment 4: Please add the information to the text that > 80% of all points are outside the range of uncertainty.

We changed the text to: The regression coefficient $R^2$ is 0.26. Most of the $H_2O_2^{PSS}$ mixing ratios (75%) vary between 0 and 65 pptv with a median value of 15 pptv, while the $H_2O_2^{obs}$ extend over a larger range mainly between 10–210 pptv with a median of 150 pptv, and thus 10 times higher than for steady-state, indicating that more than 80 % of all points in the correlation are outside the range of uncertainty. This can also be seen in the histograms in Figure 13.

**Previous comment by Referee #2:**

Table 2 and Section 4.3.2. […] I suggest that the authors add a column for ROOH in Table 2 and discuss it in the text. The comparison between observed and EMAC ROOH is shown in Figure 17 and shows a very large underestimate of EMAC (factor of 5-10?). A similar suggesting was made by the other reviewer and not addressed by the authors.

Authors' response:

We added a sentence to Section 4.3.2: In general EMAC tends to strongly underestimate measured total hydroperoxide in all air masses by a factor of 5 -10.

Editor comment 5: Given that both referees suggested to add a column to Table 2 reporting the ROOH values from EMAC and observations, your response is not sufficient. It seems that based on your discussion, you should be able to add these values.

We have added a column to Table 2 with values for $ROOH^{obs}$ and $ROOH^{EMAC}$.

**Previous comment by Referee #2:**

Lines 455-457 and Figure 19. It would be useful to include a panel comparison modeled and observed ROOH in Figure 19. It seems that removing scavenging in EMAC results in between agreement for H2O2, this begs the question as to whether it also leads to improved agreement with ROOH. Please address this.

Authors' response:

See our comment above.

Editor comment 6: Your response to the referee comment is too brief. I assume that the results for ROOH(EMAC) are not exactly identical with and without scavenging. I second the referee's suggestion to contrast the small predicted effect of ROOH scavenging to the much larger one by H2O2 by adding a panel for ROOH in Figure 19. Could you estimate which scavenging coefficient would be needed for ROOH to reach a better model/observation agreement?

As mentioned in section 4.3.2, EMAC underestimates ROOH by a factor 5 -10 in **ALL** airmasses and not only in the AMA. The EMAC sensitivity run without scavenging does not significantly improve the comparison to the observations. Actually in a panel showing $ROOH^{obs}$, $ROOH^{EMAC}$ and $ROOH^{EMAC}$(no scavenging), the latter two traces would be indistinguishable. Thus instead of following the referee's suggestion we added a statement at the end of the discussion (Sec. 4.4) that large enhancements of $MHP^{EMAC}$ and $ROOH^{EMAC}$ were not found and that thus the strong underestimation by the model of those species is not due to an overestimation of wet removal in convective clouds. Since "large enhancements" is rather unspecific, we have changed the text in line 465 to:

Please note that significant enhancements in $MHP^{EMAC}$ and $ROOH^{EMAC}$ were not found in the sensitivity study with switched-off scavenging, indicating that the strong underestimation by the model of these species is not due to an overestimation of wet removal in convective clouds. Instead, we found that EMAC underestimates ROOH in all air masses and not only in the AMA. The reasons for this underestimation are unknown. In a previous comparison of MHP observations and EMAC simulations over Europe for July 2007, Klippel et al. (2011) also reported a factor of 10 difference in the upper troposphere, while a comparison during the fall season (October 2006) yielded a rather good agreement (within a factor of 2).

Based on the sensitivity study with already zero scavenging it is not possible to further modify the scavenging coefficient for ROOH to reach a better model/observation agreement. The only way to solve this problem would be to enhance the mixing ratios of ROOH in the inflow region by up to a factor of 10 (depending on the scavenging efficiency) or include a source of ROOH in the droplets itself. However, this would be purely speculative and therefore we would prefer to leave it at the above statement.

**Previous comment by Referee #2:**

Lines 477-480 "Convective injection of $H_2O_2$ (and MHP) into the upper troposphere over India most likely forms a pool of hydroperoxides in the upper troposphere that subsequently influences the western AMA, giving rise to a significant longitudinal gradient of $H_2O_2$ and MHP mixing ratios, with increasing values towards the center of the AMA. It is likely that at least a large part of UHP is due to additional MHP from an up-wind source." I suggest replacing MHP with ROOH as ROOH is the only quantity that is measured. Also, the conclusions should reflect the fact that EMAC significantly underestimates ROOH, but that the underestimate in $H_2O_2$ is not quite as large. The conclusion lacks any mention of the sensitivity study results.

Authors' response:

We changed the text to:

Convective injection of $H_2O_2$ (and potentially MHP) into the upper troposphere over India most likely forms a pool of hydroperoxides in the upper troposphere that subsequently influences the western AMA, giving rise to a significant longitudinal gradient of $H_2O_2$ and ROOH mixing ratios, with increasing values towards the center of the AMA. It is likely that at least a large part of $UHP_{PSS}$ is due to additional MHP from an up-wind source. A sensitivity study using EMAC with no scavenging tends
to reproduce the observed longitudinal gradients in $H_2O_2$, although it does not increase the level of ROOH. The reasons for this different behavior are unclear.

Editor comment 7: Your response is unclear and confusing. The referee had suggested to replace MHP in this context by ROOH. Your response implies that the large underestimate of ROOH is exclusively due to MHP. Why do you exclude the possibility that other ROOH are underestimated?

We changed the text to: Convective injection of $H_2O_2$ (and ROOH) into the upper troposphere over India most likely forms a pool of hydroperoxides in the upper troposphere that subsequently influences the western AMA, giving rise to a significant longitudinal gradient of $H_2O_2$ and ROOH mixing ratios, with increasing values towards the center of the AMA. It is likely that next to an unidentified organic hydroperoxide (e.g. PAA) at least part of $UHP_{PSS}$ is due to additional MHP from an up-wind source. A sensitivity study using EMAC with no scavenging tends to reproduce the observed longitudinal gradients in $H_2O_2$, although it does not increase the level of ROOH. The reasons for this different behavior are unclear.

**Additional editor comments**

8) I appreciate that you added indices (PSS, EMAC, obs) to the species names. However, thy do not seem correct at all places and sometimes they are even confusing. Some examples are listed below. Please check the complete manuscript for their use.

- l. 23: 'We observed enhanced mixing ratios of […] MHP(PSS), UHP(PSS)' – is consistent with the previous sentence ('Observations are compared to photostationary calculations…')
We assume you mean inconsistent. We changed the text to:
This study focuses on in situ observations of hydrogen peroxide ($H2O2^{obs}$) and organic hydroperoxides ($ROOH^{obs}$), as well as their precursors and loss processes. Observations are compared to photostationary state calculations (PSS) of $H_2O_2^{PSS}$, and extended by a separation of $ROOH^{obs}$ into methyl hydroperoxide ($MHP^{PSS}$) and inferred unidentified hydroperoxide ($UHP^{PSS}$) mixing ratios using PSS calculations.

- l. 212 and Equation 5: Shouldn't it be H2O2(PSS) and not H2O2(obs)?
No, for the PSS calculation we only used *in situ* data to calculate $MHP^{PSS}$ and $UHP^{PSS}$.

9) l. 30 and l. 31: 'Deviations' is very qualitative. Please quantify the value and state whether there is a consistent trend (over- or underestimate) between model results and observations.
We changed the next to:
Underestimation of H2O2EMAC by appr. a factor of 2 in the NH and the AMA and overestimation in the SH (factor 1.3) are most likely due to uncertainties in the scavenging efficiencies for individual hydroperoxides in deep convective transport to the upper troposphere, corroborated by a sensitivity study.

10) l. 134 and l. 138: Clarify 'to be mainly MHP' or 'MHP only'?
We changed the text to:
The limits of detection (LOD) and precisions for H2O2 and MHP (assuming total ROOHobs to be only MHP), respectively, have been calculated for each flight from the reproducibility ($1\sigma$ standard deviation) of in-flight zero (650 values) and liquid calibration (100 values) measurements, taking into account the sensitivity, stripping and catalase efficiency.

11) l. 234/5: Should this read
'The combination of Eq. 5 and 12 **results** in Eq. 13 which was used to calculate the $MHP^{PSS}$ concentrations **based on** the observed mixing ratios during OMO.'?
We changed the text like suggested.

12) l. 244: 'To estimate the composition of the organic hydroperoxides…' should be replaced by 'To estimate the contribution of MHP to the total organic hydroperoxides…' (or similar) because you estimate does not yield any further information on the composition of ROOH.
We changed the text like suggested.

13) Equation 15: 1) The index obs should be before exponent, i.e. $[HO_2]^{obs\ 2}$
2) Specify the reaction denoted by $k_{OH}$ (i.e. similar to Eq.-13 etc $k_{OH+...}$
We changed the text like suggested.

14) l. 323-325: I do not understand this sentence. Why does the fact that you see a correlation between UHP)(PSS) and ROOH(obs) imply that this is 'mainly due to UHP(PSS)'? How does the correlation of MHP(PSS) look like?

The correlation for MHP looks mainly like the one of H2O2. We added the MHP correlation to fig 6 and added the correlation parameters to the sentence:

For H2O2obs, MHPPSS and ROOHobs the correlation is not that strong with slopes of -0.02±0.02 (ppbv/ppbv), -0.07±0.01 (ppbv/ppbv) and 0.13±0.03 (ppbv/ppbv) respectively and offsets of (0.21±0.02) ppbv. (0.12±0.01) ppbv and (0.11±0.03) ppbv (Figure 6).

15) l. 371/2:: The text here is confusing. Should it read
'For the SH the model simulated H2O2$^{EMAC}$ mixing ratios **(272 pptv)** are four times higher than in the NH background **(66 pptv),**`
We changed the text like suggested.

16) l. 373: Unless I misunderstand this sentence or the table, shouldn't it be '(**100** pptv to 211 pptv) (Table 2).'?
Yes, sorry for the confusion. We changed the text like suggested.

17) l. 374: 'measured' should be removed here
We changed the text like suggested.

18) l. 378 and 379: For both EMAC and observations, the NH background and AMA values show about a 10% difference. I suggest rewording the sentence in this regard rather than saying 'almost identical' versus 'small difference', respectively.
Similar as for H2O2EMAC, medians of MHPEMAC for NH background and AMA conditions are show very small differences (11 pptv and 13 pptv respectively, Table 2), while slightly higher differences were found for UHPEMAC in the AMA (64 pptv and 70 pptv, respectively).

19) l. 379-381: Please add the predicted values to this sentence so it is easier to follow the data in the table.
In the simulations, southern hemispheric MHPEMAC mixing ratios are almost ten times higher than NH background values (116 pptv and 11 pptv, respectively), compared to two to three times higher in the observations.

20) l. 388-390: These sentences do not read well. You mix relative differences ('factor of four') with absolute differences ('200 ppt'). Either use only the most meaningful difference or report both relative and absolute differences.
The smallest difference with 89 pptv was found for the SH (Table 2).

21) l. 400: should 'blue' be 'purple'?
We changed the text like suggested.

22) l. 409: Where can one see the 'rather large deviation of 150 pptv'? The average difference between the PSS values and observations in Figure 16 look less than that.
Since the steady-state calculations do not account for transport this can explain the rather large deviation of 144–164 pptv (between 51° and 57°) with the observations.

23) l. 414: 'Assuming that MHP is also enhanced in the outflow of deep convention…' – Is this assumption based on literature (if so, add reference(s)) or on observations in the current study?
A discussion of MHP enhancements based on literature can be found in our revised manuscript (l. 458-460). We added the two references given there (Barth et al., 2016; Mari et al., 2000) also on l.414:

Assuming that MHP is also enhanced in the outflow of deep convection (Mari et al., 2000; Barth et al., 2016)…

24) l. 426: Table 2 shows a value of 100 pptv, not 115 pptv.
We changed the value in the text to 100 ppt$_v$ after checking the data. (see also 16)

25) l. 456: Add 'EMAC' to H2O2.

Done.

26) l. 478 and l. 483: This is contradictory: You define UHP as all organic hydroperoxides except MHP. The sentence in l. 478 seems consistent with this definition 'a large contribution of an unidentified organic hydroperoxide (UHP$^{PSS}$)' whereas the later text 'a large part of UHP$^{PSS}$ is due to additional MHP from an up-wind source' is inconsistent. Please make sure that you use the definitions of ROOH, UHP, MHP consistently.

UHP is defined as all organic hydroperoxides except MHP$^{PSS}$ and thus MHP from locally oxidized CH$_4$. Transported MHP from convection is not excluded. We added in line 250:

[revised manuscript text omitted]